**The riddle of eastern tropical Pacific ocean oxygen levels : the role of the supply by intermediate depth waters.**

Olaf Duteil (oduteil@geomar.de)(1), Ivy Frenger(1), Julia Getzlaff(1)

(1) GEOMAR, Kiel, Germany

**Abstract**

Observed Oxygen Minimum Zones (OMZs) in the tropical Pacific ocean are located above intermediate depth waters (IDW) defined here as the 500 – 1500 m water layer. Typical climate models do not represent IDW properties and are characterized by a too deep reaching OMZ. We analyze here the role of the IDW on the misrepresentation of oxygen levels in a heterogeneous subset of ocean models characterized by a horizontal resolution ranging from 0.1° to 2.8°. First, we show that forcing the extra tropical boundaries (30°S/N) to observed oxygen values results in a significant increase of oxygen levels in the intermediate eastern tropical region. Second, the equatorial intermediate current system (EICS) is a key feature connecting the western and eastern part of the basin. Typical climate models lack in representing crucial aspects of this supply at intermediate depth, as the EICS is basically absent in models characterized by a resolution lower than 0.25°. These two aspects add up to a "cascade of biases", that hampers the correct representation of oxygen levels at intermediate depth in the eastern tropical Pacific Ocean and potentially future OMZs projections.

**1. Introduction**

Oxygen levels in the ocean are characterized by high values in the high latitudes and the subtropical gyres, while concentrations decrease to close to zero in the tropical oceans in the Oxygen Minimum Zones (OMZs). While OMZs are natural features, climate change is potentially responsible for their expansion (Breitburg et al., 2018), leading to a reshaping of the ecosystems and a potential loss of biodiversity.

Modelling oxygen levels is particularly challenging because of the complexity of the interactions between biological processes and physical transport (e.g Deutsch et al., 2014, Ito et al., 2013; Duteil et al., 2014a,b, 2018, Oschlies et al., 2017). Climate models tend to overestimate the volume of the OMZs (Cabre et al., 2015) and do not agree on the intensity and even sign of oxygen future evolution (Oschlies et al., 2017). In order to perform robust projections there is a need to better understand the processes at play that are responsible for the supply of oxygen to the OMZ. We focus here on the Pacific ocean, where large OMZs are located in a depth range from 100 to 900 m (Karstensen et al., 2008; Paulmier and Ruiz-Pino. 2009). Previous modelling studies have shown that the tropical OMZ extension is at least partly controlled by connections with

the subtropical ocean (Duteil et al., 2014). In addition, the role of the equatorial undercurrent (Shigemitsu et al., 2017; Duteil et al., 2018; Busecke et al., 2019), of the secondary Southern Subsurface Countercurrent (Montes et al, 2014), of the interior eddy activity (Frenger et al., 2018), have been previously highlighted. These studies focus on the mechanisms at play in the upper 500 m of the water column The oxygen content below the core of the OMZ however plays a significant role in setting the upper oxygen levels by diffusive (Duteil and Oschlies, 2009) or vertical advective (Duteil, 2019) processes. Here, we focus specifically on the mechanisms supplying oxygen toward the eastern tropical pacific ocean at intermediate depth (500 – 1500 m), below the OMZ core.

The water masses occupying this intermediate depth layer (500 – 1500 m) (Emery, 2003) subduct at high latitudes (Karstensen et al., 2008). Oxygen solubility increases with lower temperatures, thus waters formed in the Southern Ocean are characterized by high oxygen values. In particular, the Antarctic Intermediate Water (AAIW) (Molinelli, 1981) ventilates large areas of the lower thermocline of the Pacific Ocean (Sloyan and Rintoul., 2001) and is characterized by oxygen values larger than 300 mmol.m$^{-3}$ at subduction time (Russel and Dickson, 2003). The oxygenated core of the AAIW in the tropical Pacific is located at about 500-1200 m depth at 40°S (Russell and Dickson, 2003) and with this at a depth directly below the depth of the OMZs in the eastern Pacific; the Pacific AAIW mixes down to 2000 m depth with the oxygen poor Pacific Deep Water (PDW) as determined by the OMP (Optimum Multiparameter) analysis (Pardo et al., 2012; Carrasco et al., 2017). The oxygen rich (> 200 mmol.m$^{-3}$ at 40°S) AAIW spreads from its formation side in the Southern Ocean to the subtropical regions. The northern part of the Pacific basin is characterized by the North Pacific Intermediate Water (NPIW) (Talley, 1993) confined to the northern Pacific conversely to the AAIW, which spreads far northward as its signature reaches 15°N (Qu and Lindstrom., 2004). AAIW, NPIW and the upper part of the PDW are oxygenated water masses occupying the lower thermocline between 500 and 1500 m depth. In this study we do not specifically focus on the individual water masses, but rather on the water occupying the intermediate water depth (500 – 1500 m) (Emery, 2003) of the subtropical and tropical ocean. We will refer to the waters in this depth range as intermediate depth waters (IDW).

In the subtropics, the IDW (particularly the AAIW) circulates into the intermediate flow of the South Equatorial Current and the New Guinea Coastal Undercurrent (Qu and Lindstrom, 2004) where it retroflects in the zonal equatorial flows of the Southern Intermediate Countercurrent (SICC) and Northern Equatorial Intermediate Current (NEIC) within about ±2° off the equator (Zenk et al., 2005; Kawabe et al., 2010) (Fig 1). These currents are part of the Equatorial Intermediate Current System (EICS) constituted by a complex system of narrow jets extending below 500 m in the lower thermocline (Firing, 1987; Ascani et al., 2010; Marin et al. 2010; Cravatte et al., 2012, 2017;

Menesguen et al., 2019). While the existence of this complex jet system has been shown to exist in
particular using argo floats displacements (Cravatte et al., 2017) the spatial structure and variability
of the jets are still largely unknown. In addition, there is little knowledge about their role in
transporting properties such as oxygen.
The simulation of the supply of oxygen to the eastern tropical Pacific below the OMZ core is a
difficult task as it depends on the realistic simulation of the IDW properties (in particular the oxygen
content) and the IDW pathway (through the EICS). It is known that current climate models, in
particular CMIP5 (Coupled Model Intercomparison Project phase 5) models, have deficiencies in
correctly representing the IDW. In particular, the AAIW is too shallow and thin, with a limited
equatorward extension compared to observations (Sloyan and Kamenkovich, 2007; Sallee et al.,
2013; Meijers, 2014; Cabre et al., 2015; Zhu et al., 2018 for the south Atlantic ocean).
Discrepancies in the simulated properties of IDW compared to observations are due to a
combination of a range of errors in the climate models, including in the simulation of wind and
buoyancy forcing, an inadequate representation of subgrid-scale mixing processes in the Southern
Ocean, and midlatitude diapycnal mixing parameterizations (Sloyan and Kamakovich, 2007; Zhu et
al., 2018). In addition, the EICS is mostly lacking in coarse resolution models (Dietze and Loeptien,
2013; Getzlaff and Dietze, 2013). Higher resolution (0.25°, 1/12°) configurations partly resolve the
EICS but with smaller current speeds than observed (Eden and Dengler, 2008; Ascani et al., 2015).
The mechanisms forcing the EICS are complex and still under debate (see the review by
Menesguen et al., 2019).
In this study we focus on the impact of the subtropical IDW (and of the deficiencies in the
representation of its properties and transport) on the oxygen content in the eastern tropical Pacific
in a set of model simulations. Section 2 gives an overview of all models that we used as well as of
the sensitivity simulations. Next, we assess to which extent the subtropical IDW modulate (or drive)
the oxygen levels in the eastern tropical (20°S – 20°N; 160°W-coast) Pacific ocean, and determine
the role of I) the oxygen content of the IDW in the subtropical regions (section 3) and ii) on the
zonal recirculation of the oxygen by the EICS toward the eastern part of the basin (section 4). We
conclude in section 5.
**2. Description of models and experiments**
**2.1 Description of models**
We analyze the mean state of the oxygen fields, OMZ, EICS of the following model experiments
(see Table 1), which previously have been used in recent studies focusing on the understanding of
the tropical oxygen levels mean state or variability :

- The NEMO (Nucleus for European Modelling of the Ocean) model (Madec et al., 2017) has been
used throughout this study in different configurations. We first use a coarse resolution version (see
2.2). This configuration is known in the literature as ORCA2 (Madec et al., 2017) but we call it
NEMO2 in this study for clarity reasons. The resolution is 2°, refined meridionally to 0.5° in the
equatorial region. It possesses 31 vertical levels on the vertical (10 levels in the upper 100 m),
ranging from 10 m to 500 m at depth. Advection is performed using a third-order scheme.
Isopycnal diffusion is represented by a biharmonic scheme along isopycnal surfaces. The
parameterisation of Gent and McWilliams (1990) (hereafter GM) has been used to mimic the effect
of unresolved mesoscale eddies. The circulation model is coupled to a simple biogeochemical
model that comprises 6 compartments (phosphate, phytoplankton, zooplankton, particulate and
dissolved organic matter, oxygen). The same configuration has been used in Duteil et al., 2018;
Duteil, 2019. The simulation has been forced by climatological forcings based on the Coordinated
Reference Experiments (CORE) v2 reanalysis (Normal Year Forcing) (Large and Yeager, 2009)
and integrated for 1000 years. Initial fields (temperature, salinity, phosphate, oxygen) are provided
by the World Ocean Atlas 2018 (WOA) (Garcia et al, 2019; Locarnini et al., 2019)
Two other versions of NEMO have been used (see 2.2). The configuration ORCA05 (that we call
here NEMO05) is characterized by a spatial resolution of 0.5°. It possesses 46 levels on the
vertical, ranging from 6 to 250 m at depth (15 levels in the upper 100 m). Advection is performed
using a third-order scheme. Isopycnal diffusion is represented by a biharmonic scheme along
isopycnal surfaces. Effects of unresolved mesoscale eddies are parameterized following GM. In
the configuration TROPAC01 (that we call NEMO01 in the rest of this study), a 0.1° resolution two-
way AGRIF (Adaptive Grid Refinement In Fortran) has been embedded in the Pacific Ocean
between 49°S and 31°N into the global NEMO05 grid (similar to the configuration used in Czeschel
et al., 2011). Since the model is eddying in the nested region GM is not used. Both configurations
are forced by the same interannually varying atmospheric data given by the Coordinated Ocean–
Ice Reference Experiments (CORE) v2 reanalysis products over the period 1948–2007 (Large and
Yeager, 2009), starting from the same initial conditions. The initial fields for the physical variables
are given by the final state of a 60 year integration of NEMO01 (using 1948–2007 interannual
forcing and following an initial 80 year climatological spin-up at coarse resolution). The
interpretation of differences in the ventilation in the IDW is aided by the use of a passive tracer
(see 2.2.2)
- the UVIC (University of Victoria) model (e.g used in Getzlaff et al., 2016; Oschlies et al., 2017), an
earth System Model (ESM) that has a horizontal resolution of 1.8° latitude x 3.6° longitude. The
experiment has been integrated for 10000 years. The biogeochemical model is a NPZD-type
model of intermediate complexity that describes the full carbon cycle (see Keller et al., 2012 for a

detailed description). This model is forced by monthly climatological NCAR/NCEP wind stress fields.

- the GFDL (Geophysical Fluid Dynamics Laboratory) CM2-0 suite (Delworth et al., 2012; Griffies et al., 2015, Dufour et al, 2015): the suite is based on the GFDL global climate model and includes a fully coupled atmosphere with a resolution of approximately 50 km. It consists of three configurations that differ in their ocean horizontal resolutions: GFDL1 (original name : CM2-1deg) with a nominal 1° resolution, GFDL025 (original name : CM2.5) with a nominal 0.25° and GFDL01 with a nominal 0.1° resolution (original name : CM2.6) These configurations have been used in Frenger et al. (2018) and Busecke et al. (2019) for studies on ocean oxygen. At simulation year 48, the simplified ocean biogeochemistry model miniBLING has been coupled to the circulation model. It includes three prognostic tracers, phosphate, dissolved inorganic carbon and oxygen (Galbraith et al., 2015). Due to the high resolution of GFDL01, the integration time is limited. We here analyze simulation years 186 to 190.

All the models (NEMO2, UVIC, GFDL suite) are forced using preindustrial atmospheric pCO2 concentrations.

Differences in model resolution but also in atmosphere forcings or spinup duration strongly impact oxygen distribution (see Annex A). However, the heterogeneity of the configurations that we analyze permits to determine whether the simulated oxygen distributions display systematic biases / similar patterns.

The mean states of the oxygen distributions are discussed below in section 3.1 "IDW Oxygen levels in models".

**2.2 Sensitivity experiments**

In order to disentangle the different processes at play, we perform two different sets of sensitivity simulations, using the NEMO model engine. NEMO allows to test effects of increasing the ocean resolution and to integrate the model over a relatively long time span.

2.2.1 Forcing of oxygen to observed values in the subtropical regions

In the first set of experiments the focus is on the role of the lower thermocline oxygen content for the ventilation of the eastern equatorial Pacific. We use NEMO2, the oceanic component of the IPSL-CM5A (Mignot et al., 2013), that is part of CMIP5. NEMO2 shows mid-latitudes oxygen biases consistent with CMIP5 models. We compare three experiments :

- NEMO2-REF: the experiment is integrated from 1948 to 2007 starting from the spinup state described in 2.1.

- NEMO2-30S30N: the oxygen boundaries are forced to observed oxygen concentrations (WOA) at
the boundaries 30°N and 30°S in the whole water column: the mid-latitude oxygen levels in the
IDW are therefore correctly represented.
- NEMO2-30S30N1500M: same as NEMO2-30S30N; in addition oxygen is  forced to observed
concentrations below 1500m, mimicking a correct oxygen state of the deeper water masses (lower
part of the AAIW, upper part of the PDW)
With the above three experiments we focus on the transport of IDW oxygen levels to the tropical
ocean and the OMZs. The respiration rate (oxygen consumption) is identical in NEMO2-REF,
NEMO2-30S30N and NEMO2-30S30N1500M in order to avoid compensating effects between
supply and respiration that depend on biogeochemical parameterizations (e.g Duteil et al., 2012).
We aim to avoid such compensating effects to ease interpretation and be able to focus on the role
of the physical transport. The sensitivity of tropical IDW oxygen to subtropical and deep oxygen
levels is discussed in section 3.2
2.2.2 Conservative Tracer Release in oxygenated waters
In the second set of experiments, we assessed the effect of a resolution increase on the transport
of a conservative tracer. To do this, we used a 0.5° (NEMO05) and a higher resolution 0.1°
(NEMO01) configuration of the NEMO model engine (Table 1) to examine the transport of
oxygenated IDW from the subtropical regions into the oxygen deficient tropics. In these
experiments, we initialized the regions with climatological (WOA) oxygen levels greater than 150
$mmol.m^{-3}$ with a value of 1 (and 0 when oxygen was lower than 150 $mmol.m^{-3}$). In the model
simulations, the tracer is subject to the same physical processes as other physical and
biogeochemical tracers, i.e. advection and diffusion but it does not have any sources and sinks.
The experiments have been integrated for 60 years (1948 – 2007) using realistic atmospheric
forcing (COREv2).
In order to complement the tracer experiment we performed Lagrangian particle releases.
Lagrangian particles allow to trace the pathways of water parcels due to the resolved currents, and
to track the origin and fate of water parcels. The particles are advected offline with 5 days mean of
the NEMO05 and NEMO01 currents. The NEMO01 circulation fields have been interpolated to the
NEMO05 grid in order to allow a comparison of the large scale advective patterns between
NEMO01 and NEMO05. We do not take into account subgrid processes in NEMO05. We used the
ARIANE tool (Blanke and Raynaud, 1997). A particle release has been performed in the eastern
tropical OMZ at 100°W in the tropical region between 10°S – 10°N.The particles have been
released in the IDW (500 - 1500m) and integrated backward in time from 2007 to 1948 in order to
determine their pathways and their location of origin. The transport by the EICS is discussed in
section 4.2 (tracers levels and Lagrangian pathways).

**3. Intermediate water properties and oxygen content**
3.1. IDW Oxygen levels in models
The water masses  subducted in mid/high latitudes are highly oxygenated waters. The subducted
"oxygen tongue" (oxygen values up to 240 mmol.m$^{-3}$) located at IDW level is not reproduced in
most of the models part of CMIP5 (Fig 8 from Cabre et al., 2015, Fig 4 from Takano et al., 2018)
and in the models analyzed here (Fig 2a), with an underestimation of about 20-60 mmol.m$^{-3}$
(NEMO2, GFDL1, GFDL025, GFDL01). UVIC, a coarse resolution model, shows oxygenated
waters in the lower thermocline at mid latitudes (30°S-50°S). GFDL01, even though still biased low,
presents larger oxygen values than the coarser resolution models GFDL1, GFDL025 and NEMO2.
A possible explanation is a better representation of the water masses and in particular the AAIW in
eddy-resolving models (Lackhar et al., 2009).

The IDW oxygen maximum is apparent at 30°S throughout the lower thermocline (600 – 1000 m)
in observations (Fig 2b), consistent with the circulation of IDW with the gyre from the mid/high
latitude formation regions towards the northwest in subtropical latitudes (Sloyand and Rintoul.
2001), and followed by a deflection of the waters in the tropics towards the eastern basin (Qu et al.,
2004; Zenk et al., 2005). This oxygen peak is missing in all the models analyzed here.

Consistent with the low oxygen bias of models at subtropical latitudes (Fig 2b), models also feature
a bias in the tropical ocean (20°S-20°N) by 20 – 50 mmol.m$^{-3}$ (Fig 2a, Fig 2c) at intermediate
depths in the eastern part of the basin (similarly to CMIP5 models, as shown by Cabre et al.,
2015). The basin zonal average of the mean oxygen level in the lower thermocline layer (500 -
1500m) at 30°S and in the eastern part of the basin (average 20°S – 20°N, 160°W-coast; 500-1500
m) are positively correlated (Pearson correlation coefficient R=0.73) (Fig 2d, Annex A), suggesting
that the oxygen levels in the tropical pacific ocean are partly controlled by extra-tropical oxygen
concentrations at intermediate depths and the associated water masses.

The models presenting the poorest oxygenated water at 30°S display the largest volume of OMZs
(GFDL025 and GFDL1), though the negative correlation (Pearson correlation coefficient R=-0.52)
is less pronounced between the volume of the OMZs and the mean oxygen levels in the layer 500 -
1500 m at 30°S (Fig 2e).  A correlation, even weak, suggests a major role of the IDW in regulating
the OMZ volume. Reasons for this weaker correlation are due to the OMZs being a result of
several processes next to oxygen supply by IDW, e.g, vertical mixing with other water masses
(Duteil et al., 2011), isopycnal mixing in the upper thermocline (Gnanadesikan et al., 2013; Bahl et
al., 2019), supply by the upper thermocline circulation (Shigemitsu et al., 2017; Busecke et al.,
258  2019).
In order to better understand the role of IDW entering the subtropical domain from higher latitudes
for the oxygen levels in the eastern tropical Pacific Ocean, we perform sensitivity experiments (see
2.2.1) in the following.
3.2 Sensitivity of tropical IDW oxygen to subtropical and deep oxygen levels
3.2.1 Oxygen levels in the lower thermocline
The difference of the experiments NEMO2-30S30N – NEMO2-REF (average 1997-2007) (Fig 3c,d)
allows to quantify the effect of model biases of IDW at mid latitudes (30°N/30°S) on tropical oxygen
levels.
We first assess the oxygen concentration and density levels at 30°S and 30°N in both the World
Ocean Atlas (WOA) and the NEMO2-REF experiment. The deficiency in oxygen in NEMO2-REF is
clearly highlighted at 30°S, between 400 and 1500m. The density levels are well reproduced in
NEMO2-REF compared to WOA (Annex B).
As we force oxygen to observed levels at 30°S/°N (see 2.2.1), the difference between both
experiments shows a large anomaly in oxygen levels at 30°S (more than 50 mmol.m$^{-3}$) at IDW level
(500 – 1500 m) corresponding to the missing deep oxygen maximum. The northern negative
anomaly results from a deficient representation of the north Pacific OMZ, i.e., modeled oxygen is
too high for NPIW. The northern low and southern high anomalies spread towards the tropics at
intermediate depth. A fraction of the positive oxygen anomaly recirculates at upper thermocline
level due to a combination of upwelling and zonal advection by the tropical current system (for
instance the EUC at thermocline level is a major supplier of oxygen as shown in observations by
Stramma et al., 2010 and in ocean models by Duteil et al., 2014, Busecke et al., 2019).
The difference NEMO2-30S30N1500M – NEMO2-30S30N (Fig 3e,f) shows a deep positive
anomaly in oxygen, as oxygen levels are lower than in observations by 30-40 mmol.m$^{-3}$ in the
eastern tropical regions. This anomaly is partially transported into the IDW (500 - 1500 m). It shows
that a proper representation of the deep oxygen levels (> 1500 m) is important for a realistic
representation of the lower thermocline and OMZs. Causes of the oxygen bias of the deeper water
masses are beyond the scope of this study but may be associated with regional (tropical) issues,
such as an improper parameterization of respiration (e.g a too deep remineralisation) (Kriest et al.,
2010), or a misrepresentation of deeper water masses.

## 3.2.2 Oxygen budget and processes

To assess the processes that drive the oxygen content of the (sub)tropical lower thermocline, we analyzed the oxygen budget in NEMO2-REF and NEMO2-30S30N, NEMO30S30N1500M. The budget is computed as an average between 500 and 1500m and shown in Fig 3g and Fig.4.

The oxygen budget is :

$$\frac{\delta O_2}{\delta\,dt} = Adv_x + Adv_y + Adv_z + Diff_{Dia} + Diff_{Iso} + SMS$$

where $Adv_x, Adv_y, Adv_z$, are respectively the zonal, meridional and vertical advection terms, $Diff_{dia}$ and $Diff_{iso}$ are the diapycnal and isopycnal diffusion terms. SMS (Source Minus Sink) is the biogeochemical component (i.e below the euphotic zone this is only respiration)

In NEMO2-REF, the physical oxygen supply is balanced by the respiration. The oxygen supply in the model is divided into advection, i.e., oxygen transport associated with volume transport, and isopycnal diffusion, i.e. subgrid scale mixing processes that homogenize oxygen gradient. Diapycnal diffusion is comparatively small and can be neglected.

The supply of oxygen from the high latitudes toward the tropical interior ocean is constituted by several processes acting concomitantly. Below the subtropical gyre, the oxygen is transported from the south eastern to the northern western part of the gyre (Fig 4a and 4b). Downwelling from the oxygen-rich mixed layer supplies the interior of the subtropical gyre (Fig 4c). Isopycnal diffusion transfers oxygen from the oxygen-rich gyres to the poor oxygenated regions (Fig 4d). At the equator, the EICS transport westward oxygen-poor water originating in the eastern side of the basin (Fig 4a). The meridional advection term transports oxygen originating from the subtropics (Fig 4b) in the tropical regions, which is upwelled (Fig 4c).

Forcing oxygen levels in NEMO2-30S30N at 30°S and 30°N creates an imbalance between respiration (which remains identical in NEMO2-REF and NEMO2-30S30N) and supply. The oxygen anomaly generated at 30°S propagates equatorward. The positive anomaly originated from the southern boundary recirculates in the equatorial region. Isopycnal diffusion is a major process that transport the oxygen anomaly toward the equator (Fig 3g, Fig 4h), in particular from 30°S to the 5°S and 30°N to 10°N. Total advective transport plays an important role in the transport of the oxygen anomaly as well, especially in the equator region (Fig 4e and 4f) and and in the western boundary (Fig 4f). Meridional advection plays a large role close to the 30° boundaries as the oxygen is transported by the deeper part of the gyres. As the vertical gradient of oxygen decreases (the intermediate ocean being more oxygenated), the vertical supply from the upper ocean decreases in the south (increases in the north) subtropical gyre (Fig 4g). Comparatively the impact

on zonal term advection (Fig 4e) is small as the zonal oxygen gradient stays nearly identical in both experiments (the oxygen anomaly is almost longitude independent). The model does not display much increase in zonal recirculation at the equator as well, except in the western part of the basin due to the advection of the oxygen provided by the retroflection of the deep limb of the subtropical gyre. The increase of meridional transport (Fig 4f) is caused by the change in oxygen meridional gradient, mainly caused by isopycnal diffusion processes away from the western boundary.

In the experiment NEMO2-30S30N1500, in complement to the isopycnal propagation of the subtropical anomaly, the deep (> 1500 m) oxygen anomaly is upwelled in the eastern equatorial (500 – 1500 m) part of the basin (see Fig 3g). The transport due to advective terms strongly increases, mostly due to an increase in vertical advection. This is consistent with the analysis by Duteil (2019) who showed that vertical advection is the dominant process to supply oxygen from the lower to the upper thermocline in the equatorial eastern Pacific Ocean in a similar NEMO2 configuration.

This simple set of experiments already shows that in climate models oxygen in the lower thermocline (500 – 1500 m) tropical ocean are partially controlled by properties of IDW that enter the tropics from higher latitudes. This presumably also applies to other (biogeochemical) tracers. IDW oxygen propagates equatorward mostly by small scale isopycnal processes and the western boundary currents. Further, upwelling in the tropics from deeper ocean layers (Pacific Deep Water, partially mixed in the lower IDW) play an important role. We will examine more closely in the following the representation and the role of the EICS in supplying oxygen toward the eastern Pacific Ocean.

**4. Equatorial intermediate current system and oxygen transport**

4.1 Structure of the currents in the upper 2000 m in observations and models

The current structure of the models analyzed in this study (see section 2.1, Table 1) is shown in Fig 5. In the mixed layer, the broad westward drifting South and North Equatorial Currents (SEC, NEC) characterize the equatorial side of subtropical gyres. In the thermocline, the eastward flowing equatorial undercurrent (EUC), flanked by the westward flowing south and north counter currents are present in all models. This upper current structure is well reproduced (i.e the spatial structure and intensity are consistent with observations) across the different models (see 2.1 "Model analyzed") compared to observations. Previous studies already discussed the upper thermocline current structure in the GFDL models suite (Busecke et al., 2019), NEMO2 and NEMO05 (e.g Izumo, 2005, Lübbecke et al., 2008), UVIC (Loeptien and Dietze, 2013); the upper thermocline will not be further discussed in this study.

367

At intermediate depth, in the observations, a relatively strong (about 0.1 ms$^{-1}$) westward flowing Equatorial Intermediate Current (EIC) is present below the EUC at about 400-600 m depth (Marin et al., 2010). A complex structure of narrow and vertically alternating jets every 200 m, so-called Equatorial Deep Jets (EDJ), extends below the EIC till 2000 m (Firing, 1987; Cravatte et al., 2012). Laterally to the EIC, in the upper thermocline, the Low Latitude Subsurface Countercurrents (LLSC) are observed. They include the North and South Subsurface Counter Currents (NSCC and SSCC), located around 5°N/5°S, and a series of jets between 5°N/S and 15°N/S (in particular the Tsuchiya jets in the southern hemisphere, described by Rowe et al., 2000). Below the LLSCs, the Low Latitude Intermediate Currents (LLICs) include a series of westward and eastward zonal jets (500–1500-m depth range) alternating meridionally from 3°S to 3°N; the North and South Intermediate Countercurrents (NICC and SICC) flow eastward at 1.5°–2° on both flanks of the lower EIC. The North and South Equatorial Intermediate Currents (NEIC and SEIC) flow westward at about 3° (Firing, 1987). A detailed schematic view of the tropical intermediate circulation is shown in a recent review by Menesguen et al. (2019) and in Fig 1.

In coarse resolution models, the intermediate current system is not developed and sluggish (even missing in UVIC and GFDL1). NEMO2 and NEMO05 display an incomplete EICS as the LLSCs are not represented. High resolution models (GFDL025, GFDL01, NEMO01) display a more realistic picture, even if the mean velocity is still weaker than in observations (smaller than 5 cm.s$^{-1}$), where it reaches more than 10 cm$^{-1}$ at 1000 m (Ascani et al., 2010; Cravatte et al., 2017). An interesting feature is that the jets are broader and faster in NEMO01 than in GFDL01.  Possible causes include a different wind forcing, mixing strength or topographic features as all these processes play a role in forcing the intermediate jets (see the review by Menesguen et al., 2019). The intermediate currents are less coherent vertically in NEMO01 than in GFDL01, due to their large temporal variability in NEMO01. A strong seasonal and interannual variability of the EICS has been observed that displays varying amplitudes and somewhat positions of the main currents/jets (Firing, 1998; Gouriou et al., 2006: Cravatte et al., 2017). A clear observational picture of the EICS variability is however not yet available. Outside the tropics (in particular south of 15°S), the interior velocity pattern is similar in coarse and high resolution models, suggesting a similar equatorward current transport at intermediate depth in the subtropics, in for instance NEMO05 and NEMO01.

4.2 Transport by the EICS

4.2.1 Tracer spreading towards the eastern tropical Pacific

We released a conservative tracer in the subtropical domain in well oxygenated waters (waters where observed oxygen concentration is greater than 150 mmol.m$^{-3}$ - see 2.2.2) in a coarse (NEMO05) and a high resolution configuration (NEMO01). The tracer does not have sources or

sinks and is advected and mixed as any other model tracer and allows to assess the transport
pathway of tracer (such as oxygen) from oxygenated waters into the oxygen deficient eastern
tropical Pacific.
The importance of the ventilation by the oxygen rich waters, and in particular the IDW, is illustrated
by the tropical tracer concentration after 50 years (Fig 6a) of integration (mean 2002-2007).
Concentrations decrease from the release location to the northern part of the basin, where the
lowest values (below 0.1) are located in NEMO05 and NEMO01. The 0.1 isoline is however
located close to the equator in NEMO05 while it is found around 7°N in NEMO01. This feature is
associated with a pronounced tongue of high tracer concentration (> 0.2) between 5°N and 5°S in
NEMO1. Such a tongue is absent in NEMO05. The enhanced tracer concentration in the equatorial
region suggests a stronger zonal equatorial ventilation in NEMO01, consistent with a stronger
EICS (Figure 5)
The preferential pathways of transport are highlighted by the determination of the transit time it
takes for the tracer to spread from the oxygen rich regions to the tropical regions. We define a
threshold called t10% when the tracer reaches a concentration of 0.1 (Fig 6b) (similar to the
approach of SenGupta and England, 2007). t10% highlights a faster ventilation of the equatorial
regions in NEMO01 compared to NEMO05, as t10% displays a maximum value of 10 (western
part) to 30 years (eastern part) between 5°N/5°S in NEMO01 compared to 30 years to more than
50 years in NEMO05. The southern "shadow zone" is well individualized in NEMO01 compared to
NEMO05 as the oxygen levels are high in the equator in NEMO01, suggesting a strong transport
by the EICS. The value of t10% increases linearly at intermediate depth at 100°W in NEMO05 from
20°S to the equator, suggesting a slow isopycnal propagation (consistent with the experiments
performed using NEMO2 in part 3.2). Conversely, the tracer accumulation is faster in the equatorial
regions than in the mid-latitudes in NEMO01, suggesting a larger role of advective transport, which
is faster than the transport by isopycnal diffusive processes.
4.2.2  Equatorial IDW circulation
The analysis of the dispersion of Lagrangian particles (see 2.2.3) permits us to understand the
origin of the waters circulating in the eastern part of the basin at IDW level. A total of 26515
particles have been released in the area located at 100°W, 10°N-10°S, 500-1500 m. These
particles have been integrated backward in time in order to determine their origin and the
ventilation of the eastern tropical Pacific ocean (Fig 7).
After 5 years of backward integration we find that the particles originate from a well defined region,
which extends from 110°W and 80°W to NEMO05 (Fig 7a). This region extends westward till
150°W, as a result of the stronger currents in NEMO01 (Fig 7b). This larger dispersion and
westward origin of the particles is clearly visible after 10, 20 and 50 years of integration. In order to
quantify the dispersion of the particles, we define the Intermediate Eastern Pacific Ocean (IETP) as
the region 10°N-10°S, 500 – 1500 m, 160°W – coast. The particles originating outside of the IETP
in close to 5 % / 50 % of the cases in NEMO05 and 10 % / 60 % of the cases of NEMO01, after a
time scale of respectively 10 and 50 years. The Fig 7c shows a lag between NEMO01 and
NEMO05 : while 10 % of the particles originate outside the IETP after 10 years in NEMO01 the
same quantity is reached only after 20 years in NEMO05, suggesting a stronger transport in
NEMO01. However, after the time period of 20 years, the number of particles originating outside
the IETP does not grow faster any more in NEMO01 compared to NEMO05. A hypothesis is
enhanced recirculation in NEMO01: the same particles may recirculate several times in the
equatorial region due to alternating zonal jets in NEMO01.
The transport has been quantified based on this Lagrangian particles release (Fig 8). The volume
transport is higher in NEMO01 (up to 0.2 Sv) (Fig 8a) compared to NEMO05 (less than 0.1 Sv at
the equator) (Fig 8b). It also shows recirculating structures and alternating eastern and western
transport in NEMO01 (Fig 8c). These recirculating structures are absent in NEMO05 and foster the
dispersion of particles as shown above.
**5. Summary and conclusions**
IDW are constituted by waters masses  which are  subducted in the Southern Ocean and
transported equatorward to the tropics by isopycnal processes (Sloyan and Kamenkovich, 2007;
Sallee et al., 2013; Meijers, 2014) and the western boundary currents. At lower latitudes they
recirculate into the lower thermocline of the tropical regions at 500 - 1500 m and into the EICS
(Zenk et al., 2005; Marin et al., 2010; Cravatte et al., 2012; 2017; Ascani et al., 2015;  Menesguen
et al., 2019) (see schema Fig 1). We show here that the representation of this ventilation pathway
is important to take into account when assessing tropical oxygen levels and the extent of the OMZ
in coupled biogeochemical circulation or climate models. Particularly, we highlight two critical, yet
typical, biases that hamper the correct representation of the tropical oxygen levels.
5.1 Subtropical IDW properties and tropical oxygen
First, the current generation of climate models, such as the CMIP5 models, show large deficiencies
in simulating IDW. Along with an unrealistic representation of IDW properties when the waters
enter the subtropics, the models also lack the observed prominent oxygen maximum associated
with IDW. Restoring oxygen levels to observed concentrations at 30°S/30°N and at 1500 m depth
in a coarse resolution model, comparable to CMIP5 climate models in terms of resolution and
oxygen bias, shows a significant impact on the lower thermocline (500 – 1500 m) oxygen levels: a
positive anomaly of 60 mmol.m$^{-3}$ at midlatitudes translates into an oxygen increase by 10 mmol-m$^{-3}$
in tropical regions after 50 years of integration.

The equatorward transport of the anomaly in the subtropics is largely due to the isopycnal subgrid
scale mixing processes away from the western boundaries, as shown by the NEMO2 budget
analysis. It suggests that mesoscale activity plays a major role in transporting IDW equatorward. In
addition subsurface eddies may transport oxygen westward from the eastern Pacific ocean toward
the mid-Pacific ocean region (Frenger et al., 2018, see their Fig 2).

5.2 Transport at IDW level and Equatorial Intermediate Current System
Second, the Equatorial Intermediate Current System (EICS) is not represented in coarse
resolution models and only poorly represented in high resolution ocean circulation models (0.25°
and 0.1°), as its strength remains too weak by a factor of two (consistent with previous studies, e.g
Ascani et al., 2015). The EICS transports the IDW that occupies the lower thermocline (500 – 1500
m depth) and the recirculation of the IDW in the tropical ocean, as suggested by the observational
study of Zenk et al. (2005), and shown in our study.

We investigated the impact of the EICS on the oxygen supply with tracer release experiments: the
concentration of a conservative tracer that originates from the subtropical ocean, is, after 50 years,
30 % higher in the eastern equatorial (5°N-5S) Pacific in an ocean model with 0.1° resolution,
compared to an ocean model with 0.5 ° resolution. As the oxygen gradient along the equator is
similar to the gradient of the conservative tracer, we assume a similar enhancement of oxygen
supply by 30 % in the eastern equatorial Pacific at the same time scale. This means, if we account
for oxygen consumption due to respiration (about 1 mmol.m$^{-3}$.yr$^{-1}$ between 5°N-5°S, see section
3.2), that the better resolved EICS in the higher resolution ocean leads roughly to higher
intermediate oxygen levels of 15 - 30 mmol-m$^{-3}$ compared to the lower resolution ocean experiment
in a timescale of 50 years. Consistently, the 0.1°-ocean GFDL01 model displays oxygen
concentrations larger by about 30 mmol.m$^{-3}$ in the eastern equatorial lower thermocline (500-1500
m) compared to the 1°-ocean GFDL1 configuration (with higher subtropical oxygen concentrations
of IWM of 15 mmol.m$^{-3}$ in GFDL01 at 30°S)

We would like to highlight two potential implications of our finding of the important role of the EICS
for the Pacific eastern tropical oxygen supply: i) First, we have shown that the intermediate current
system EICS is important for the connection between the western and eastern Pacific Ocean at a
decadal / multidecadal time scale. This suggests that the EICS modulates the mean state and the
variability of the tropical oxygen in the lower thermocline, and subsequently the whole water
column by upwelling of deep waters. ii) Second, we have found an enhancement of the
connections between the equatorial deep ocean (> 2000 m) and the lower thermocline if the
resolution of a model is enhanced. This result is consistent with the studies of Brandt et al. (2011,
2012), who suggested, based on observational data and on an idealized model, that Equatorial
Deep Jets as part of the EICS (see Fig 1b) propagate their energy upward and impact the upper
ocean properties of the ocean, including their oxygen content. Taken this into account, we
hypothesize that the Pacific Deep Water has a larger role than previously thought in modulating the
intermediate and upper ocean properties.

A pragmatic approach to account for the missing EICS is to increase diffusion anisotropically, with
increased zonal mixing in the tropics (Getzlaff and Dietze, 2013). This parameterization mimics a
more vigorous EICS and improves the simulated shape of the OMZ in climate models (see Annex
C). However, the prominent bias of IDW in climate models, and therefore of the water masses
entering the EICS is not accounted for with this parameterization. Furthermore such a
parameterization improves the mean state but does not reproduce the variability of the EICS.

5.3 Implication for biogeochemical cycles
The IDW are an important important supplier of oxygen to the tropical oceans, but also of nutrients
(Palter et al., 2010) as well as anthropogenic carbon (e.g Kathiwala et al., 2012), which
accumulates in mode and intermediate waters of the Southern Ocean (Sabine et al., 2004;
Resplandy et al., 2013). The mechanisms that we discussed here may therefore play a role in
ocean carbon climate feedbacks on time scales of decades to a century.

This study shows that there is a need to look with greater care into IDW properties to understand
the tropical oxygen distribution in models, in particular in CMIP class models. As shown by
Kwiatkowski et al. (2020), CMIP6 models (typical horizontal resolution of 1°) do not agree on the
future change in tropical oxygen levels (mean 100 – 600m, their Fig 2). This may partly originate in
a misrepresentation of the properties of the IDW in the different models and the strength of the
connection between western and eastern Pacific Ocean. Simple analyses, similar to our Fig 2
(oxygen levels at 30°S and oxygen levels in the eastern tropical Pacific) and Fig 9 (Mean Kinetic
Energy at intermediate depth) may give some insight into the mechanisms at play. In addition,
analyses of experiments performed in the context of the High Resolution Model Intercomparison
Project (resolution greater than 0.25°) (Haarsma et al., 2016), part of CMIP6, will give a more
complete insight on whether a significant Equatorial Intermediate Current System develops at
higher resolution. While HighResMIP are not coupled with a biogeochemical module, velocity fields
are available at a monthly resolution, which allows to perform "offline" tracer or Lagrangian particle
experiments.

Finally, this study suggests that changes of the properties of the IDW may contribute to the still
partly unexplained deoxygenation of 5 mmol.m$^{-3}$ / decade occurring in the lower thermocline of the
equatorial eastern Pacific Ocean (Schmidtko et al., 2017; Oschlies et al., 2018). In addition to an
oxygen decrease in tropical regions, Schmidtko et al. (2017) showed a decrease of oxygen levels
by 2-5 mmol.m$^{-3}$ in the regions of formations of AAIW. Based on repeated cruise observations,
Panassa et al. (2018) highlighted an increase of the apparent oxygen utilization in the core of the
AAIW, together with a 5 % increase in nutrient concentrations from 1990 to 2014. The transport of
this modified AAIW, poorer in oxygen and richer in nutrients, toward the low latitudes both by small
scale processes (section 3) and at the equator by the EICS (section 4), may explain a significant
part of the occurring deoxygenation in the equatorial ocean. In addition to changes in the AAIW
properties, little is known about the variability and long term trend of the strength of the EICS, an
oceanic "bridge" between the western and the eastern part of the basin. After our first steps toward
assessing the role of extratropical oxygen characteristics and the zonal transport of waters at
intermediate depths for tropical oxygen concentration, a possible way forward to further assess this
cascade of biases could be to perform idealized model experiments in high resolution
configurations, aiming to assess both the effect of the observed change in the AAIW properties and
of a potential change of EICS strength on oxygen levels.
**Data and code availability**
The code for the Nucleus for European Modeling of the Ocean (NEMO) is available at:
https://www.nemo-ocean.eu/. The code for the University of Victoria (UVIC) model is available
at :http://terra.seos.uvic.ca/model/. The Lagrangian particles ARIANE code is available at
http://stockage.univ-brest.fr/~grima/Ariane/. The Coordinated Ocean-ice Reference Experiments
(COREv2) dataset is available at: https://data1.gfdl.noaa.gov/nomads/forms/core/COREv2.html.
The experiments data is available on request.
**Authors contributions**
OD conceived the study, performed the NEMO model and ARIANE experiments. OD, IF and JG
analyzed the data, discussed the results and wrote the manuscript.
**Competing interest**
The authors declare that they have no conflict of interest.

**Acknowledgments**
This work is a contribution of the SFB754 "Climate-Biogeochemistry Interactions in the Tropical
Ocean", supported by the Deutsche Forschungsgemeinschaft (DFG). The NEMO simulations were
performed at the North German Supercomputing Alliance (HLRN). We would like to thank Markus
Scheinert (research unit "Ocean Dynamics", GEOMAR) for his technical support in compiling the
NEMO code and for providing the high resolution NEMO input files. We would like to thank GFDL
for producing the CM2-0 suite that involved a substantial commitment of computational resources
and  data storage. J.G acknowledges support by the project "Reduced Complexity Models"
(supported by the Helmholtz Association of German Research Centres (HGF) – grant no. ZT-I-
0010). I.F. acknowledges the German Federal Ministry of Education and Research (BMBF) project
CUSCO (grant no. 03F0813A). O.D acknowledges the German Research Foundation (DFG) (grant
no. 434479332)

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

 **Figures and Table**

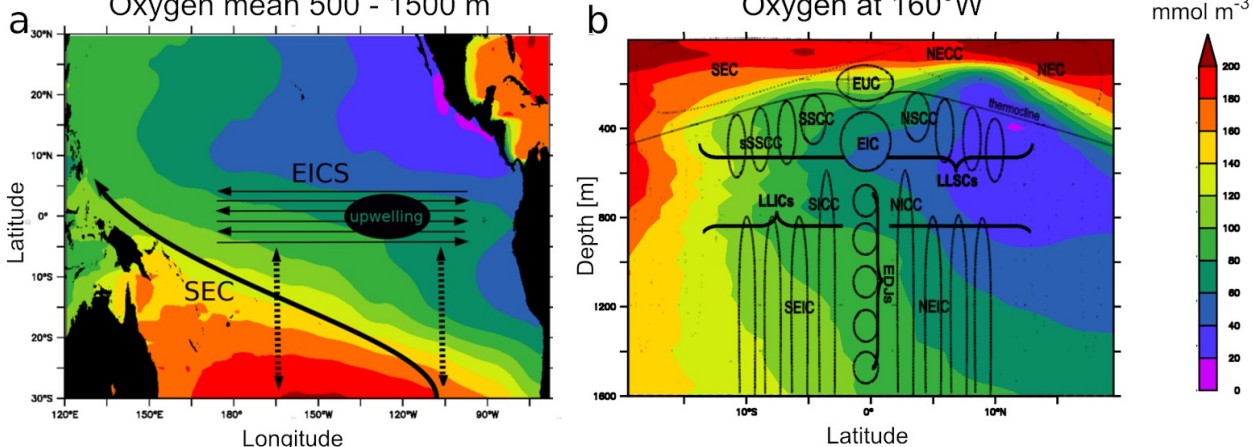

Figure 1 : a- schema summarizing the intermediate water masses (IWM)  pathway from the subtropics into the equatorial regions. EICS : Equatorial Intermediate Current System. SEC : South Equatorial Current (Kawabe et al., 2008). Dashed line : isopycnal diffusive processes. Observed (World Ocean Atlas) oxygen levels (mmol.m$^{-3}$) in the lower thermocline (mean 500-1500m) are represented in color. b - schema (adapted from Menesguen et al., 2019) illustrating the complexity of the EICS, extending below the thermocline till more than 2000 m depth (see section 4.1 for a detailed description). Observed (World Ocean Atlas) oxygen levels at 160°W are represented in color. SEC : South Equatorial Current. N/SEC : North/South Equatorial Current. NECC: North Equatorial Counter Current. EUC : Equatorial Undercurrent. EIC : Equatorial Intermediate Current. N/SSCC : North / South Subsurface Counter Current. LLSC : Low Latitude Subsurface Currents. LLIC : Low Latitudes Intermediate Currents. N/SEIC : North / South Equatorial Intermediate Current. N/SICC : North / South Intermediate Current. EDJ : Equatorial Deep Jets.

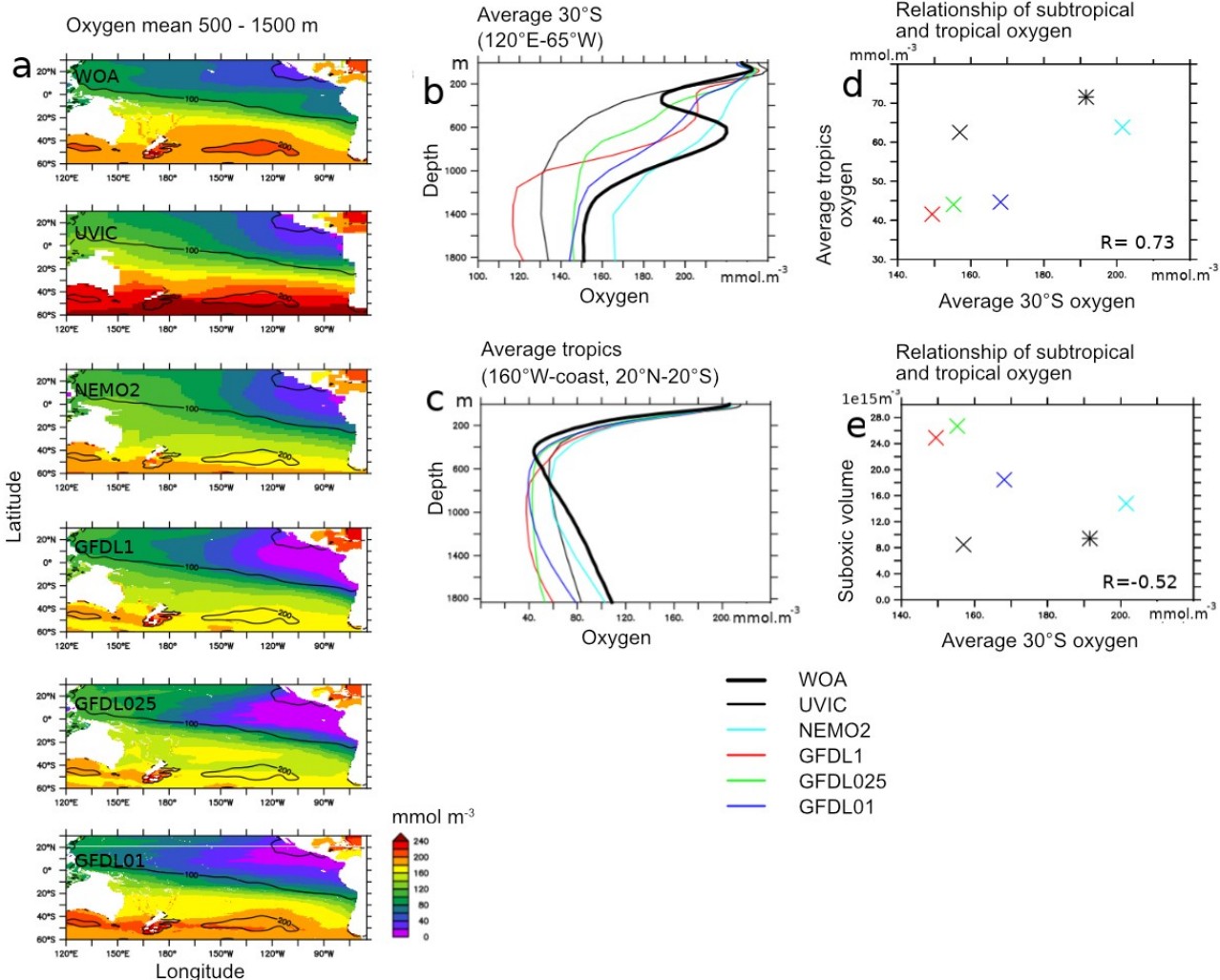

Figure 2 : a- oxygen levels (mmol.m⁻³) in observations (World Ocean Atlas - WOA) (mean 500 – 1500 m) and models (UVIC, NEMO2, GFDL1, GFDL025, GFDL01). Contours correspond to WOA values. b: average "30°S" (120°E-65°W, 30°S) c : average "tropics" (160°W-coast, 20°N-20°S). d: average "30°S" vs "tropics". e: average "30°S" vs volume of tropical suboxic ocean (oxygen lower than 20 mmol.m⁻³) regions (1e15m3). b-e : UVIC : black, NEMO2 : cyan, GFDL1 : red, GFDL025, green; GFDL01 : blue, WOA: bold line (b,c) and star (d,e).

881

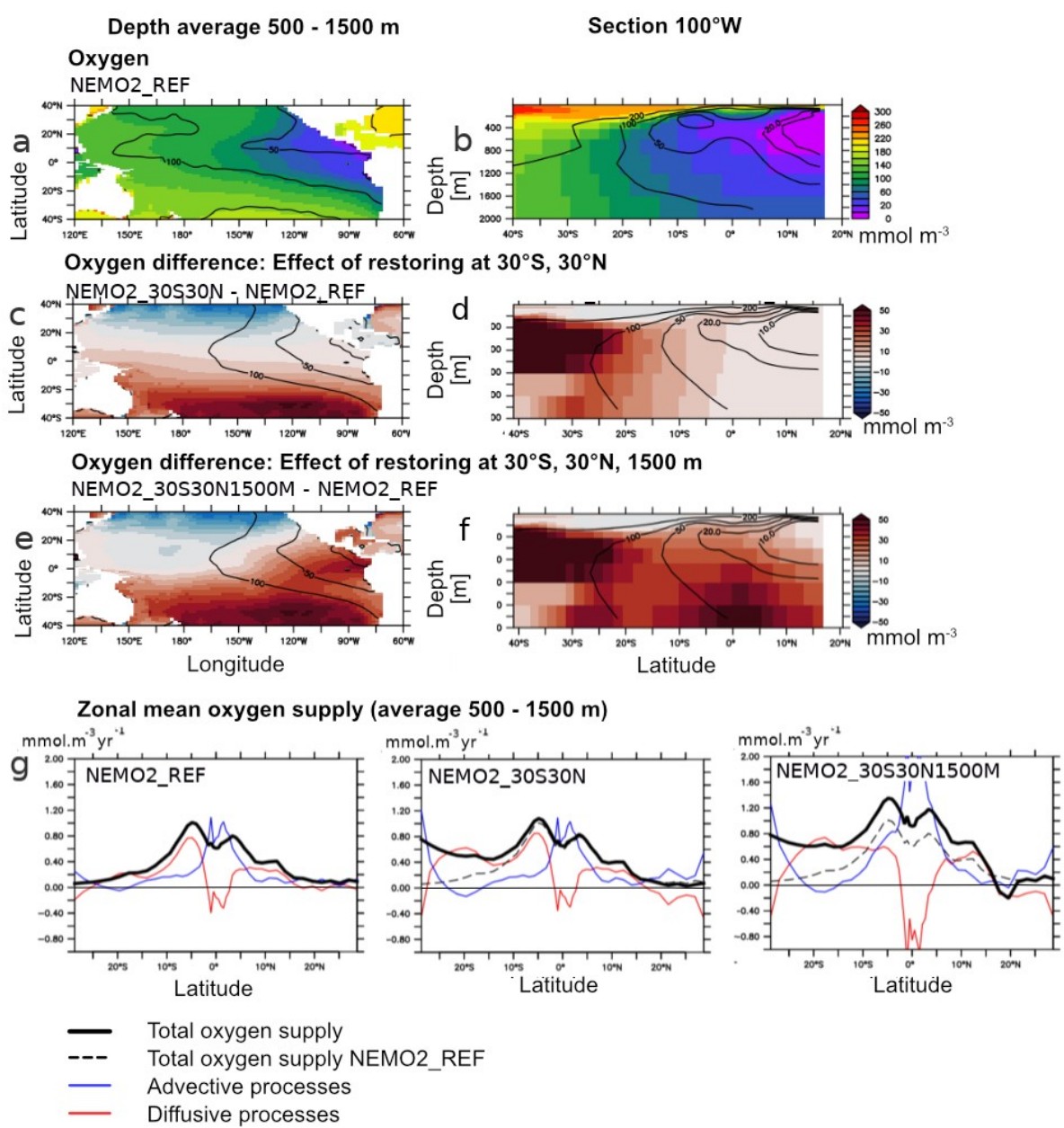

Figure 3 : a,b: Oxygen (mmol.m⁻³) in the experiments NEMO2_REF (color) and World Ocean Atlas (contour) (a- average 500-1500 m, b- 100°W). c,d: Oxygen (mmol.m⁻³) difference (c- average 500 – 1500m, d- 100°W) between the experiments NEMO2_30S30N minus NEMO2_REF. e,f : Oxygen (mmol.m⁻³) difference (e- average 500-1500m, f- 100°W) between the experiments NEMO2_30S30N1500M minus NEMO2_REF. g- basin zonal average (average 500 - 1500 m) of the oxygen total supply (bold) (mmol.m⁻³.year⁻¹), advective processes (blue) and isopycnal diffusion (red) in NEMO2_REF, NEMO2_30S30N, NEMO2_30S30N1500M. The dashed line is the oxygen total supply in NEMO2_REF.

890

891

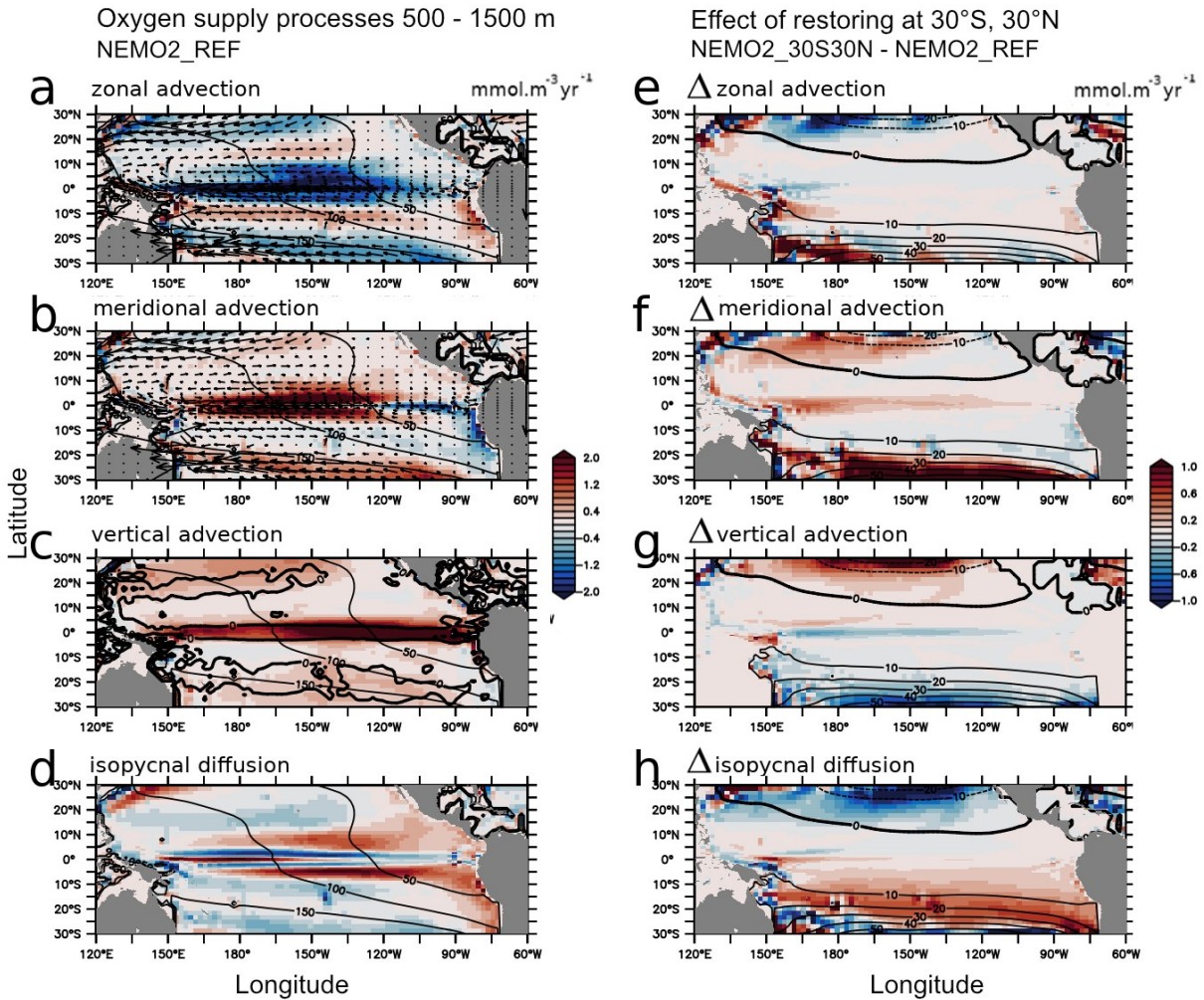

Figure 4 : a-d Oxygen supply processes (mmol.m$^{-3}$.year$^{-1}$ – average 500 - 1500m) in NEMO2_REF : a -zonal advection, b -meridional advection, c- vertical advection, d- isopycnal diffusion. The mean meridional and zonal currents are displayed as vectors (meridional, zonal advection). The mean vertical current (0 isoline) is represented as bold contour (vertical advection). Oxygen levels (mmol-m.$^{-3}$) are displayed in black contour. e-h: Difference in oxygen supply processes (mmol.m$^{-3}$.year$^{-1}$ – average 500-1500m) between NEMO2_30S30N and NEMO2_REF : e- zonal advection, f- meridional advection, g- vertical advection, h- isopycnal diffusion. The NEMO2_30S30N – NEMO2_REF oxygen anomaly (mmol.m$^{-3}$) is displayed in contour.

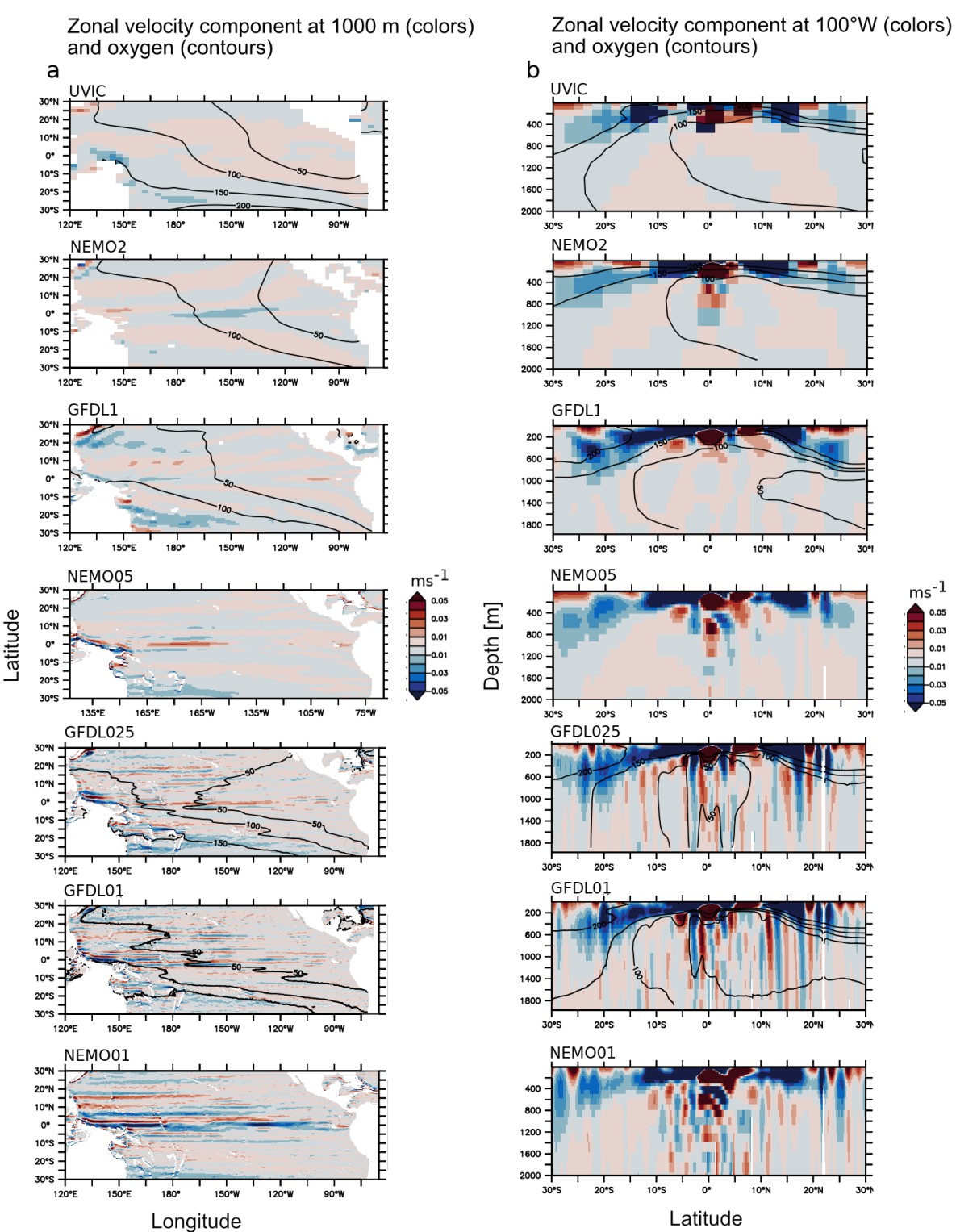

905

Figure 5 : mean currents velocity (ms⁻¹) at a- 1000 m depth  b- 100°W in UVIC, NEMO2, NEMO05,
GFDL025, GFDL01, NEMO01. The mean oxygen levels (mmol.m⁻³) (when coupled circulation-
biogeochemical experiments have been performed – see Table 1) are displayed in contour.

909

910

911

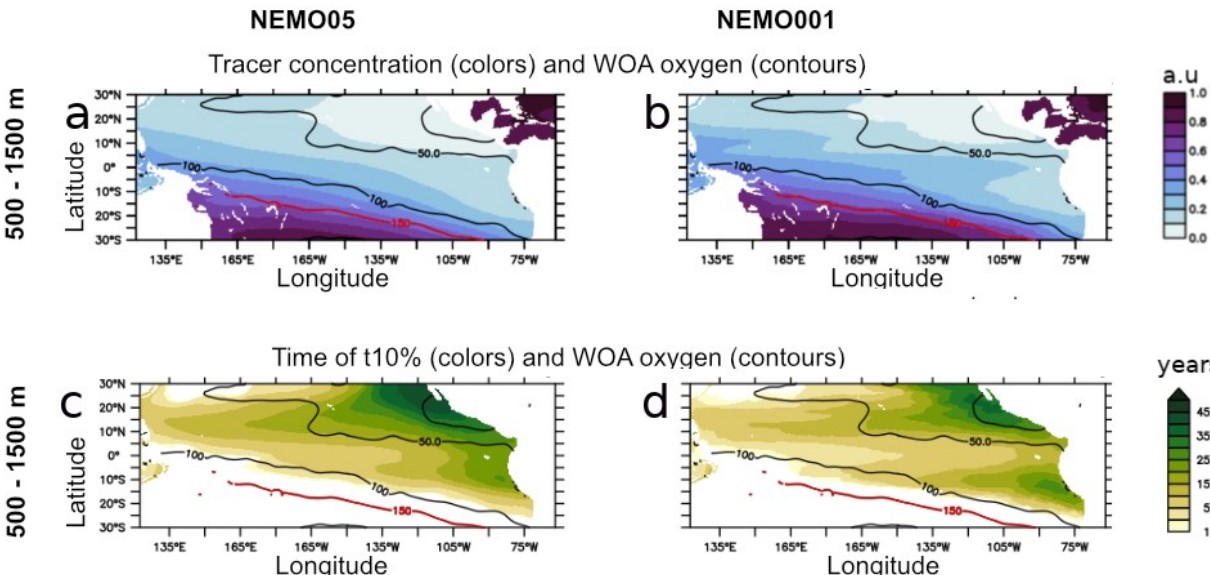

Figure 6: mean 500 – 1500 m tracer concentration (arbitrary unit) after 60 years integration in a-
NEMO05 and b - NEMO01. Time (years) at which the released tracer reaches the concentration
0.1 (t10%) in c- NEMO05 and d- NEMO01: The WOA oxygen levels (mean 500 – 1500 m) are
displayed in contour. The red contour is the WOA 150 mmol.m$^{-3}$ oxygen isoline, used to initialize
the tracer level.

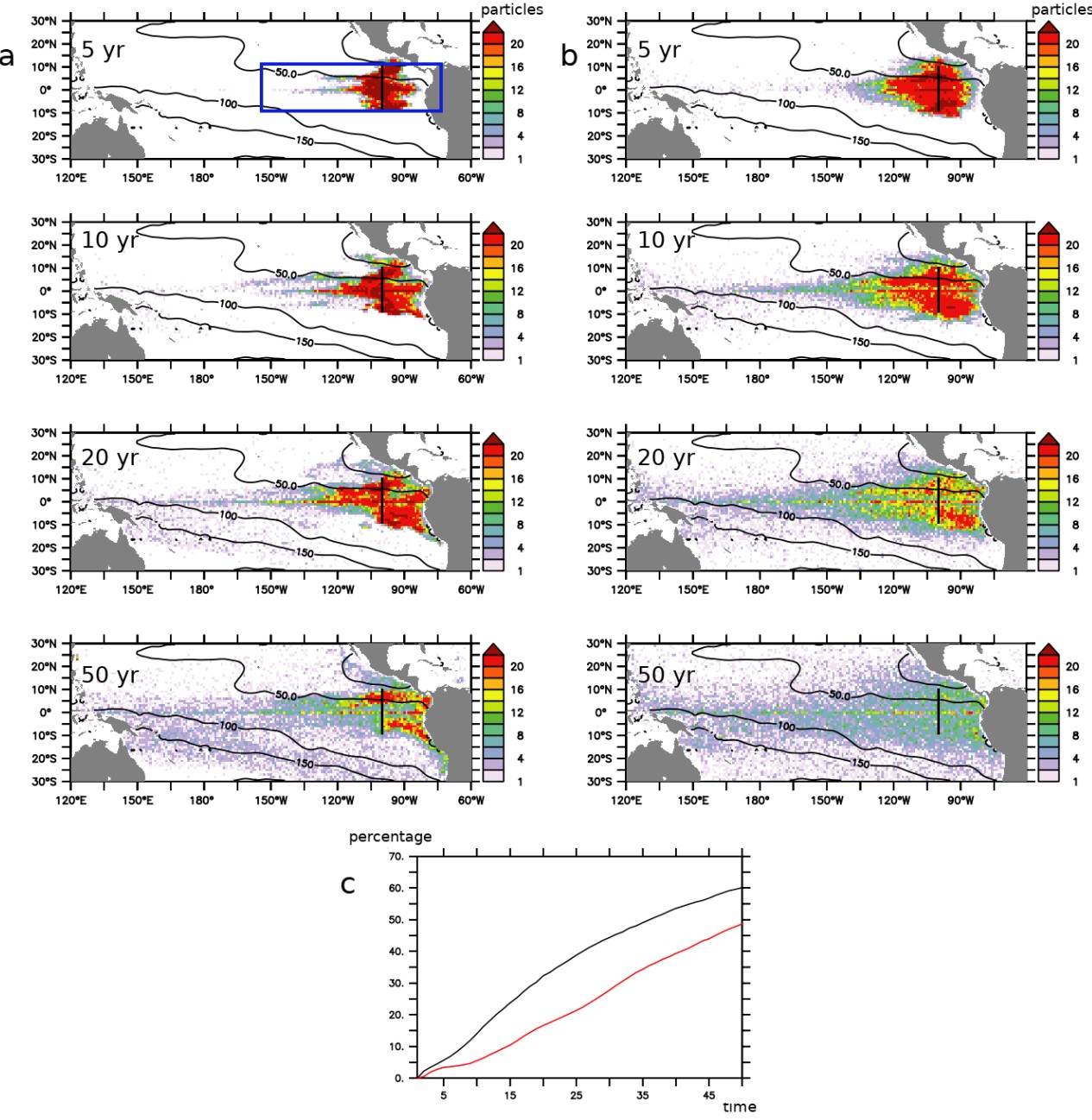

Figure 7 : Density (number of particles in a 1°x1° box) distribution of the location of released
Lagrangian particles (backward integration in years) in a - NEMO05 and b- NEMO01. The release
location is identified in bold and is located at 100°W/10°N-10S/500-1500 m depth (black line). The
number of particles have been integrated vertically. The observed mean (500 – 1500 m) oxygen
levels (WOA) are displayed in contour. The blue contour represents the Intermediate Eastern
Tropical Pacific basin (IETP). c – percentage of particles originating outside the Intermediate
Eastern Tropical Pacific (IETP) basin (160°W, 10°N-10°S, 500-1500 m) in NEMO05 (red) and
NEMO01 (black) over time (years)






























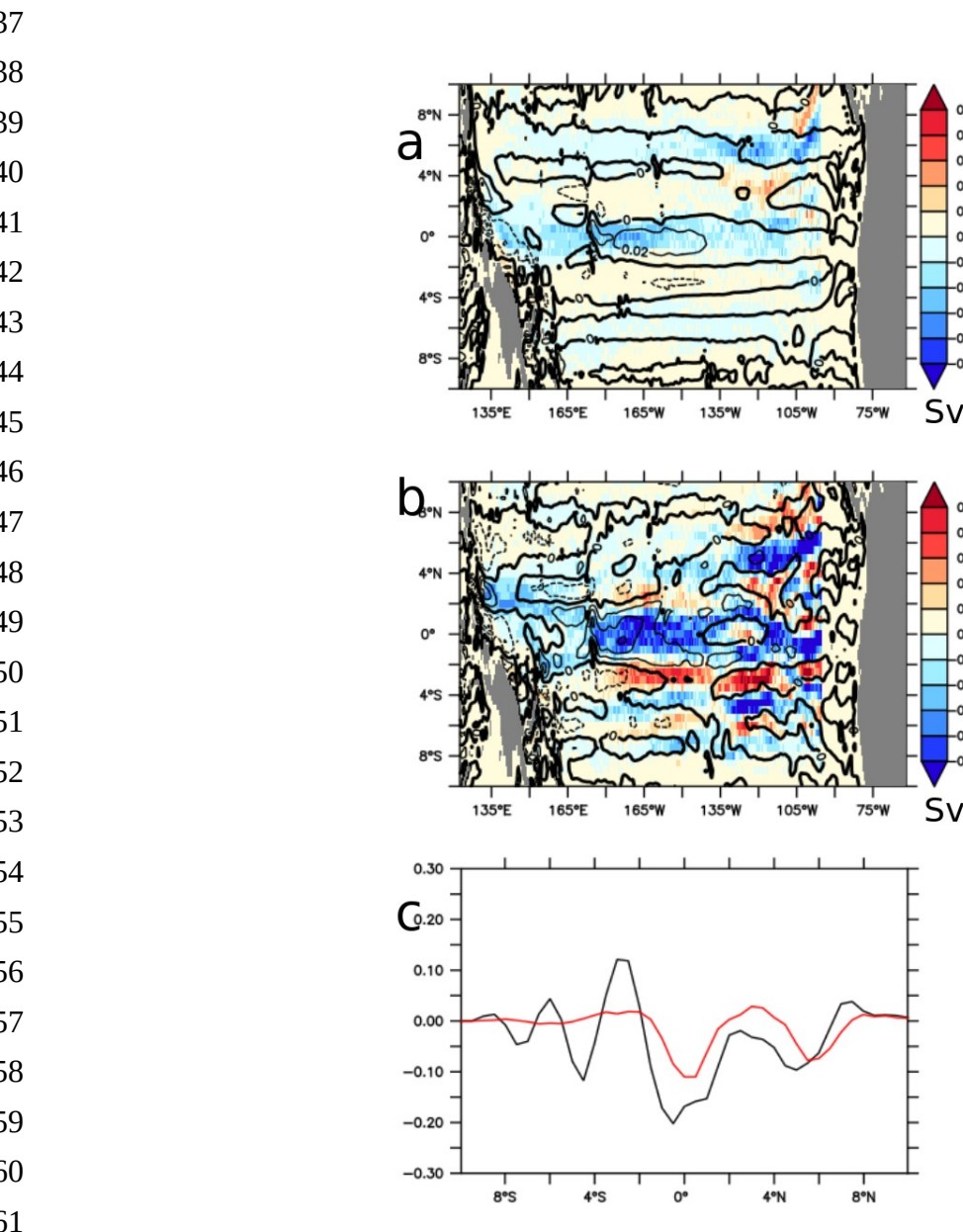

Figure 8 : mean transport (Sv) in a- NEMO05 and -b NEMO01 derived from the release of particles at 100°W, 10°N-10°S, 500-1500m (backward integration). The mean zonal velocity (ms$^{-1}$) is represented in contour. c- zonally integrated transport (Sv) derived from the release of particles at 100°W, 10°N-10°S, 500-1500m in NEMO05 (red) and NEMO01 (black)







Table 1 :

| Model | Resolution | Atmosphere | Integration (years) | BGC | Model Reference (circulation) | Model Reference (BGC) |
|---|---|---|---|---|---|---|
| **Mean state comparison** | | | | | | |
| UVIC | 2.8° | Coupled (temperature, humidity) Forced (NCEP/ NCAR wind stress) | 10000 | UVIC-BGC | Weaver et al., 2001 | Keller et al., 2012 |
| NEMO2 | 2° (0.5 eq) | Forced COREv2 "normal year" | 1000 | NPZD-O2 | Madec et al., 2015 | Kriest et al, 2010 Duteil et al., 2014 |
| GFDL1 | 1° | Coupled | 190 | BLING | Delworth et al, 2012, Griffies et al, 2015 | Galbraith et al., 2015 |
| GFDL025 | 0.25 ° | Coupled | 190 | BLING | | |
| GFDL01 | 0.1° | Coupled | 190 | BLING | | |
| | | | | | | |
| **Process oriented experiments** | | | | | | |
| Model | Resolution | Atmosphere | Integration (years) | BGC | Characteristics | |
| NEMO2 -REF -30N30S -30N30S1500M (section 2.2.1) | 2° (0.5 eq) | Forced COREv2 1948-2007 | 60 | NPZD-O2 | - control experiment - O2 restoring to WOA at 30°N/30°S - O2 restoring to WOA at 30°N/30°S/1500m | |
| NEMO05 (section 2.2.2) | 0.5° | Forced COREv2 1948 - 2007 | 60 | Tracer release | - Tracer initialized to 1 (O2 WOA > 150 mmol.m-3) or 0 (O2 WOA < 150 mmol-m-3) | |
| NEMO01 (section 2.2.2) | 0.1° | Forced COREv2 1948 – 2007 | 60 | Tracer release | | |








**Annex A**

The differences in oxygen levels between the "models groups" (GFDL suite, UVIC, NEMO2) are partly related to differences in the atmospheric fields employed and the integration time (see 2).

1. Wind forcing

Zonal wind mean stress typically varies by 5 to 20 % between the different wind products (Chauduri et al., 2013). To test this impact, we performed an experiment using the UVIC model using 2 different wind products (NCEP and COREv2 – Large and Yeager, 2009) (Figure A1). While the shape of the OMZ shows slight differences, the volume of the OMZ and the mean oxygen levels in the tropical regions and in the mid latitudes are similar. Consistent with the Figure 2, higher oxygen levels at 30°S lead to higher oxygen levels in the tropical ocean and to a smaller OMZ volume (Figure A2)

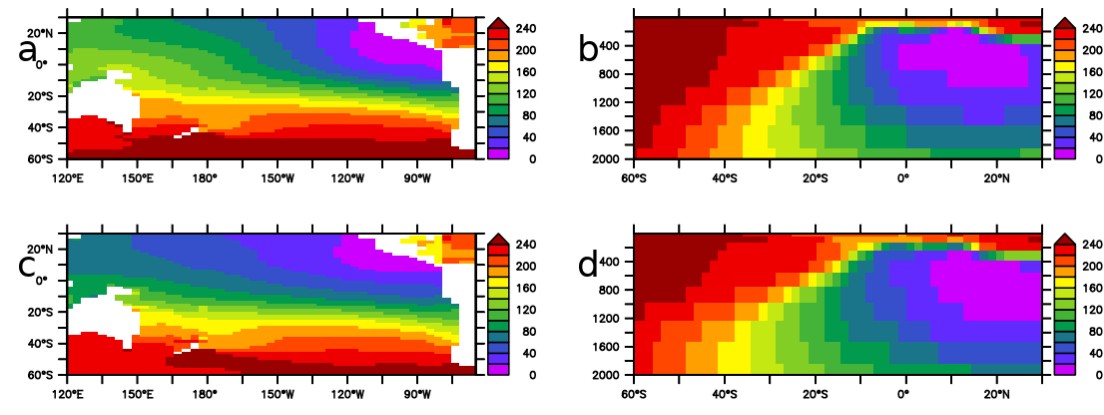

Figure A1 : Oxygen levels in UVIC (10000 years integration) a- mean 500-1500 m forcing NCEP. b-section 120°W forcing NCEP. c- mean 500-1500 m forcing COREv2, d- section 120°W forcing COREv2.

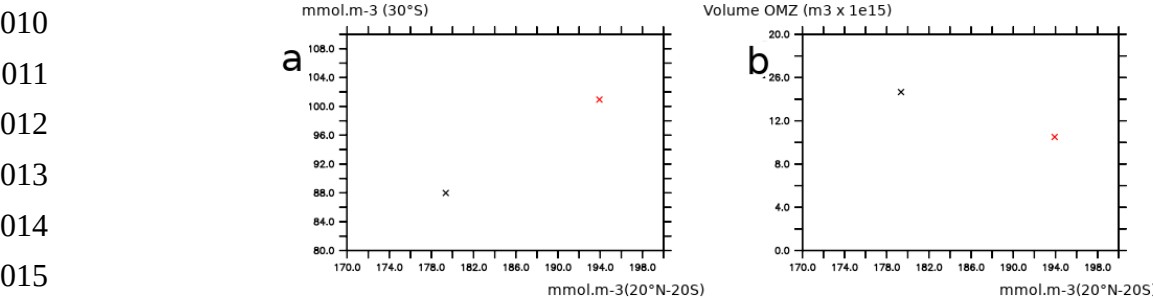

Figure A2 : a - Oxygen levels in UVIC (10000 years integration) at 30°S (zonal mean in the Pacific Ocean from surface to 2000 m depth) and in the tropical regions (20°S-20°N, averaged over the

whole Pacific Ocean). b - Oxygen levels in UVIC (10000 years integration) at 30°S (zonal mean in
the Pacific Ocean, from surface to 2000 m depth) and volume of the OMZ in the Pacific Ocean.
The configuration forced by COREv2 is shown in black, the configuration forced by NCEP is shown
in red.

2. Spinup state
In complement, the spinup state of the model also impacts the oxygen levels as the deep ocean
needs thousands of years to be in equilibrium. It may explain why UVIC (integrated for 10000
years) is characterized by much larger oxygen levels than the GFDL model suite (integrated for
190 years). As an example, the Figure A3 shows the evolution of oxygen levels during spinup in
NEMO2. Larger oxygen levels at 30°S (e.g after 1000 years of integration) are characterized by a
smaller OMZ volume (which is consistent with Fig 2) (Figure A4)

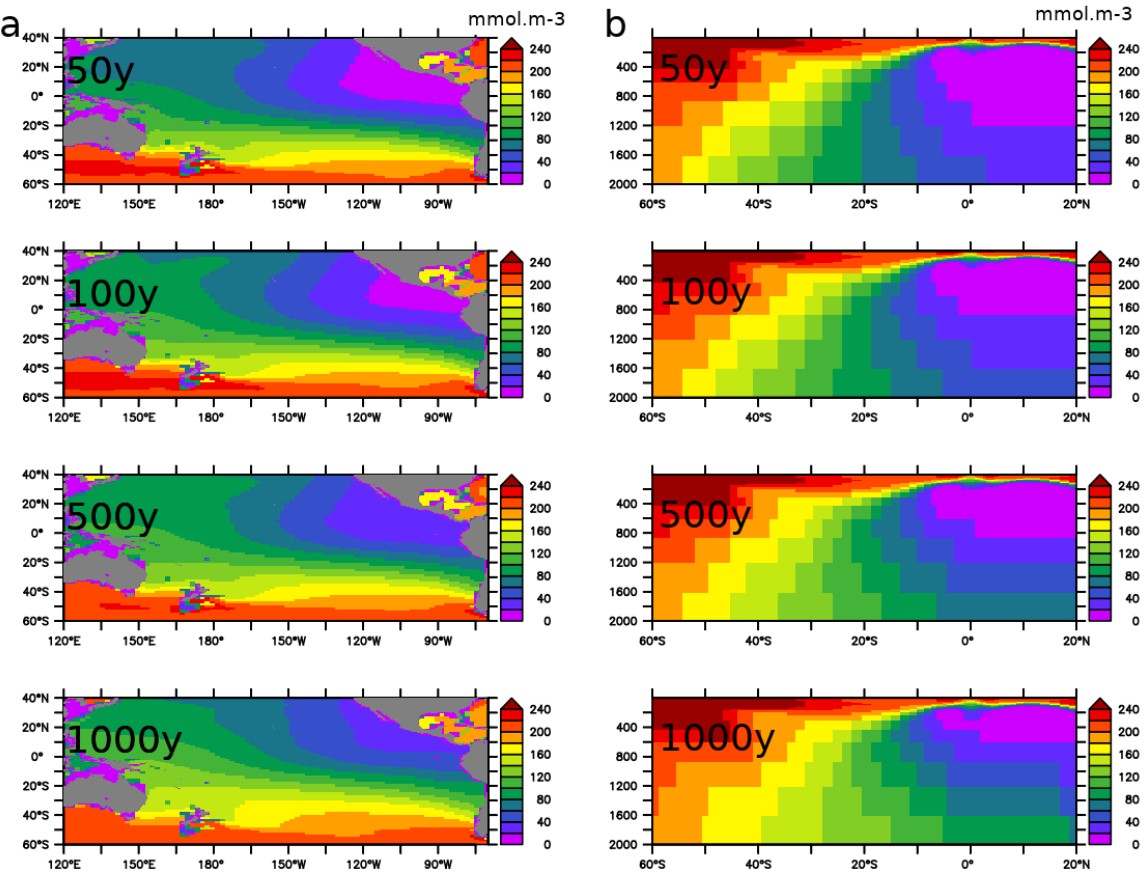
Figure A3 : oxygen levels at a - intermediate depth (average 500 – 2000 m) and b - 120°W in
NEMO2 after 50, 100,500 and 1000 years integration

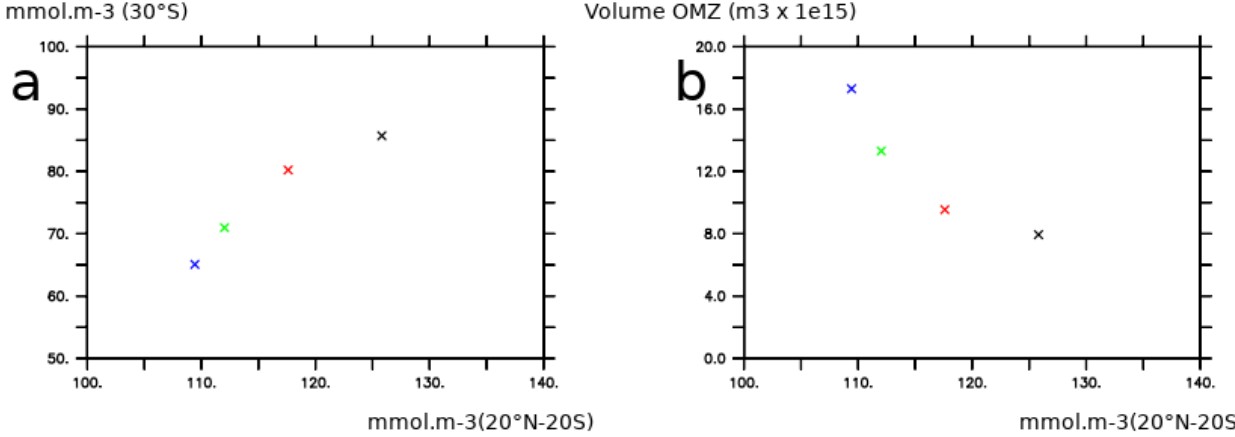

Figure A4 : a - Oxygen levels in NEMO2 at 30°S (zonal mean in the Pacific Ocean from surface to
2000 m depth) and in the tropical regions (20°S-20°N, averaged over the whole Pacific Ocean from
surface to 2000 m depth). b - Oxygen levels in NEMO2 at 30°S (zonal mean in the Pacific Ocean
from surface to 2000 m depth) and volume of the OMZ in the Pacific Ocean. The color of the cross
depends of the integration duration (black : 50 years, red : 100 years, green : 500 years, blue 1000
years).



















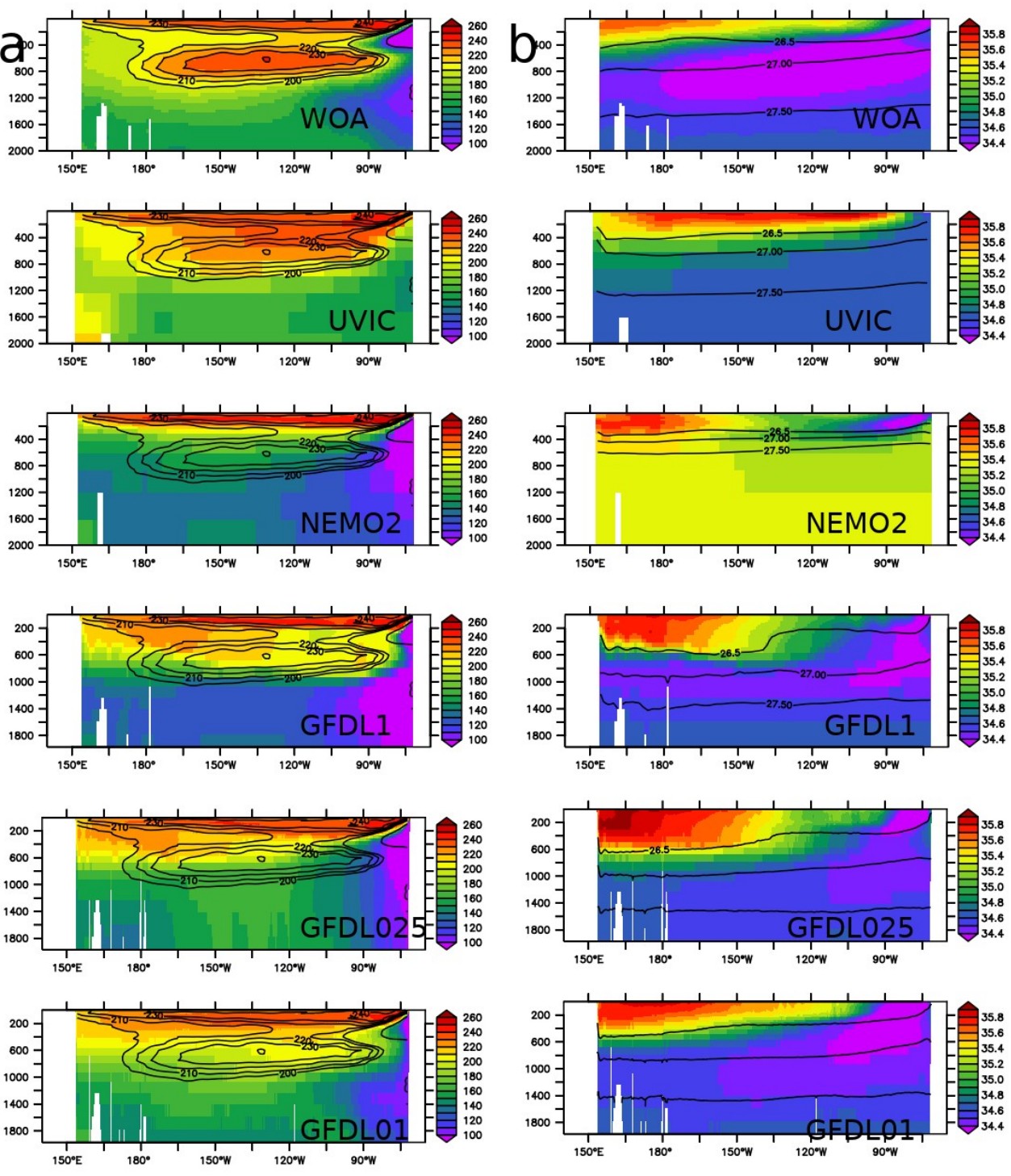


Figure A5 :  a - oxygen levels (mmol.m-3) in observations and models at 30°S. The WOA oxygen
levels are displayed in contour. b- salinity in observations and models at 30°S. The density
anomaly (26.5, 27, 27.5) is displayed in contour.

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

in WOA and NEMO2 in the subtropical and tropical ocean.


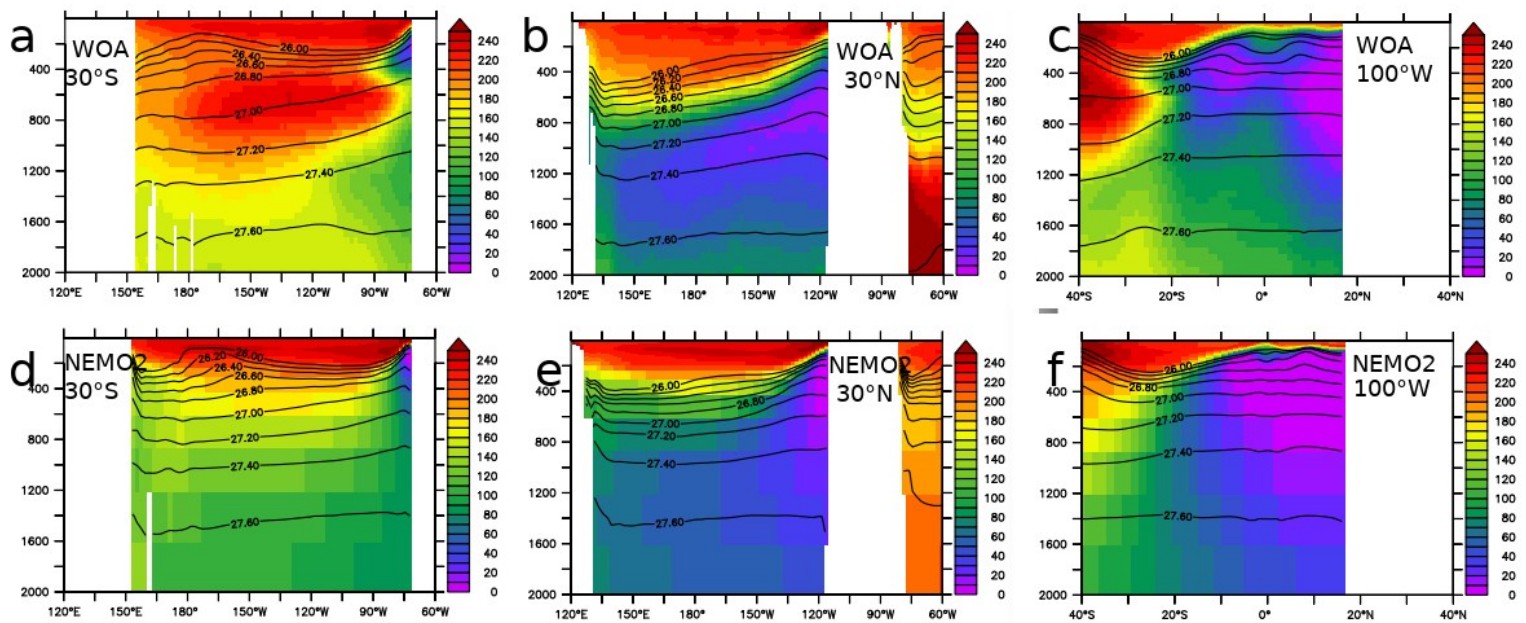

Fig B1 :  oxygen levels (mmol.m$^{-3}$) (color) and density levels (contour) at 30°S, 30N and 100°W in
the WOA dataset (a,b,c) and NEMO2-REF experiment (d,e,f)



**Annex C**

The experiments discussed in 4.2 were not coupled with biogeochemical cycles for computational cost reasons. In order to assess the robustness of our findings (EICS plays a large role in setting tropical oxygen levels), we next analyze equatorial oxygen in a set of climate models similar to CMIP models. To this end we use the GFDL model suite, characterized by a resolution increase (GFDL1, GFDL025 and GFDL01 - see Table 1).

The striking difference between GFDL01 and GFDL025 / GFDL1 are the high oxygen levels in the eastern part of the ocean below 1000 m in GFDL01 compared to GFDL025/GFDL1 (Fig 2). The oxygen levels show weaker zonal gradient in GFDL01, consistent with the tracer experiment that we performed in 4.2. and a more ventilated intermediate equatorial ocean. High values of mean kinetic energy are associated with higher oxygen values (Fig C1). This is particularly clear in GFDL01 at around 1500 m depth, where strong values of MKE are present and form the "bottom" of the low oxygen volume (oxygen lower than 50 mmol.m-3). Conversely GFDL025 and GFDL1 do not present high MKE values below 1000 m in the eastern part of the basin; the low oxygen volume extends till depths greater than 2000 m. It suggests that intermediate currents participate in the ventilation of the eastern tropical ocean and thus in limiting the vertical extension of the OMZ.

Oxygen levels do not increase linearly with the currents strength, i.e while currents strength increase in GFDL1, GFDL025 and GFDL01, oxygen levels are relatively similar in GFDL1 and GFDL025 (see Fig 5 and Fig C1). The relatively small net balance between large fluxes of respiration and oxygen supply (Duteil et al., 2014) may be responsible for this behavior. If the supply is slightly higher compared to the consumption by respiration, it will lead to an increase of oxygen concentration. If it is slightly lower, the oxygen levels will decrease. A small difference in supply (e.g slightly weaker currents) may therefore lead to a large difference in oxygen levels when integrated over decades. For this reason, the impact of the EICS is more visible below 1000 m as the respiration decreases following a power-law with depth (Martin et al., 1987) and is therefore easier to offset even by a moderate oxygen supply.

Resolving explicitly the EICS results in a similar oxygen distribution to what Getzlaff and Dietze (2013) (GD13) achieved with a simple EICS parameterization (Fig C1a): to compensate for the "missing" EICS in UVIC, a coarse resolution model, they enhanced anisotropically the lateral diffusivity in the equatorial region. The oxygen levels from UVIC GD13 are shown in blue contours on top of the UVIC oxygen distribution (black) in Fig C1. Implementing this approach tends to homogenize oxygen levels zonally, with an increase of the mean levels by 30-50 mmol.m-3 in the eastern basin and a decrease of oxygen concentrations in the western basin. While this approach

may be useful to better represent the oxygen mean state, it however does not take into account the
potential variability and future evolution of the EICS.

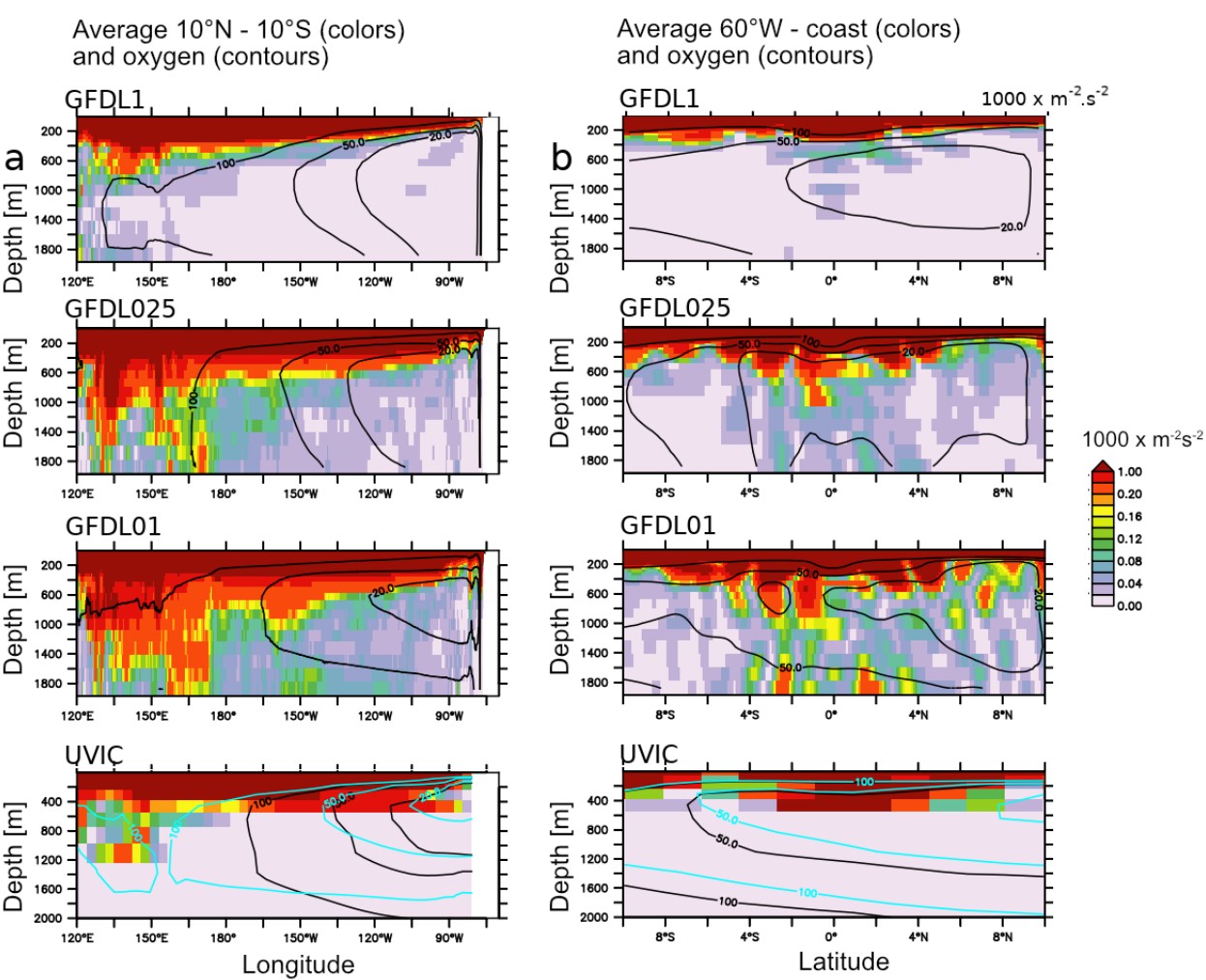




Figure C1 : a - Mean Kinetic Energy (m2.s-2 x 1000) (average 10°N-10°S) in GFDL01, GFDL025,
GFDL01, UVIC, b - similar to a. but average 160°W- coast. Oxygen levels (mmol.m-3) are
displayed in black contour. The blue contour corresponds to UVIC GD13 (Getzlaff and Dietze,
2013, including an anisotropical increase of lateral diffusion at the equator)
