# Peer review of "Intermediate water masses, a major supplier of oxygen for the eastern tropical Pacific ocean"

_Ocean Science, 2020_

## Referee Comment (RC1) · Anonymous Referee #1 · 30 Apr 2020

**1 Main Objective of this Study and General Comments**

This study investigates the impact of intermediate water masses (IMW) and it's pathway and supply along Equatorial Intermediate Current System (EISC) on dissolved oxygen content in the Pacific Oxygen Minimum Zone (OMZ) (in the eastern tropical Pacific ocean). The authors utilized a suite of simulations to address these questions.

The manuscript consists of i) mean state diagnostics and evaluations from suite of models (NEMO (ocean stand-alone simulation), UVIC (coupled, energy moisture balance model, forced wind stress), GFDL (coupled)) and ii) sensitivity simulations (or transient simulations over 60 years) (oxygen restoring, conservative tracer release, and Lagrangian tracking of tracers)elucidating the role of subtropical IMW on dissolved

oxygen supply (through EICS) in eastern tropical Pacific ocean.

Despite the limitations (or discrepancies) in simulating properties of IWM in the current climate models, the authors did a nice set of simulations tackling how bias in IMW and EICS could impact on dissolved oxygen (and possibly impact on projections of OMZs due to climate change). This could provide insights on improving ocean bio-geochemistry in ESMs and I think the work contains interesting and important results. However, I have several comments and some sections and figure presentations should be revisited before publication. Therefore, I suggest a major revision. I state specific comments below and hope this helps to improve the manuscript.

**2 Major Comments**

[1] The heterogeneous subset of models (simulations) will be an advantage exploring model and resolution dependencies (as author stated in L116 - 118) on IMW characteristics and tracers (here dissolved oxygen) but also makes the results difficult to interpret to some extent. I still think the results will have impacts from not only the differences in model structures and resolutions, but also the forcing (forcing dataset, prescribed vs. coupled) and model integration time (spinup states) (some specific comment on forcing dataset is stated below). I would like to ask authors to discuss further on these points since for example, the wind and buoyancy forcing bias could be one of the reasons introducing errors in climate (and ocean) models as stated in the introduction.

[2] Regarding to sensitivity of tropical IWM oxygen to subtropical and deep dissolved oxygen levels, the authors refer AAIW, NPIW (and the upper part of the PDW) as IWM in this study. I was wondering what will be the relative contributions of each water masses to dissolved oxygen supply, ventilation in the eastern tropical Pacific ocean (particularly North (NPIW) vs South (AAIW)). My impression is that AAIW could be more dominant
(e.g. Talley, 2013) but I would like to know what sensitivity simulations indicates. At least, I think it is possible to obtain insights from the Lagrangian tracking diagnostics (or if possible, conducting additional restoring simulations with 30°S boundary only for example).

[*Reference*] Talley, Lynne D., (2011), Descriptive Physical Oceanography: An Introduction, Academic Press.

[3] The core of the study is based on a suite of sensitivity simulations from NEMO (NEMO2). In the first reading, I struggled a bit on connecting aim and each sensitivity experiments. The dissolved oxygen restoring simulations aim on investigating sensitivity of tropical IWM oxygen to subtropical and deep dissolved oxygen levels (as stated in section 3.2) and the conservative tracer release simulations are more dedicated to investigate spreading of tracers towards the eastern tropical Pacific (transport by the EICS, as stated in section 4.1).

While the standard structure of the manuscript is to introduce overall data and methods in the beginning, (section 2), I suggest to move some of the objective and details of sensitivity experiments to each corresponding sections (referring to sections 3.2 and 4.1) so it is much easier to follow the aim bridging to sensitivity experiments (I think it is still fine to keep brief general descriptions in section 2 including Table 1). Alternatively, the methods section could be revised to include additional descriptions connecting to corresponding result sections. I will leave this decision to the authors regarding to the structure of the paper but I think the flow could be improved.

[4] Another major issue is the figures. Figure labels and captions are not easy to interpret (and in some part, the authors are referring to figure does not appear, *e.g.L267, Fig.4i*). Therefore, I suggest the authors to carefully revisit all the figures and add necessary caption, labels for better presentation. For example, for time series plots (*e.g.Fig.2, 3g - i, 8*), the difference in color (models, configurations etc.) should also be informed in the label (not just in figure captions) because it is not easy to follow.

OSD
Similar issues for multiple maps (such as Fig.5), it will be reader friendly to label maps with "zonal advection", "meridional advection" etc. Also, some of the model names (labels) are not obvious because those are overlaid on color shading (*e.g.* Fig.9). I put few more specific suggestions below and hope this helps to point out the difficulties I am referring to.

[4.1] *Fig.*1 caption, (L762 - 763) oxygen levels (mean 500 - 1500m) at 160W, I think color shading in *b*) is not vertical mean (because it is depth-latitude section). Also, is dissolved oxygen in *Fig.*1 from observations such as World Ocean Atlas? [4.2] Regarding to *Fig.*4, I have several suggestions to improve figure presentation. I am still a bit confused what is in color shading and contours. For example, in *L*789, it states the vertical current as contour in *c*) but the contours do not look like vertical current values. Also the continent shading in *g*) is missing (no gray shaded). Similar confusion occurred to me in other panels and I suggest to revisit and clearly state what is presented in color shading and contours for each panels with units. Also, why did you only present the results from NEMO2-30DEG (not including NEMO2-30DEG1500M or NEMO2-30DEG)?

[4.3] Add information labels for Fig.7a) - c) the first release, and d) - e) the second release, respectively.

[4.4] Add information labels (like figure title) for Fig.9, zonal sections and meridional sections, respectively.

**3 Minor Comments**

[1] I am curious whether CORE v2 climatological forcing (used for NEMO) and NCEP/NCAR climatological forcing (wind stress, used for UVIC) makes a difference in
spinup states. As far as I know, CORE v2 forcing is based on NCEP/NCAR reanalysis but it has several corrections and adjustments in the forcing and difference between the two could lead to different results, particularly after long-term spinup. Do authors think this is a minor thing?

[2] Are all the GFDL model simulations integrated for the same period following high-resolution (GFDL01) for comparison (I assumed 200 years from Busecke et. al., 2018) or the low-resolution configurations are integrated for longer durations?

[3] Because of the high resolutions configurations for GFDL01, the integration time is limited but does this impact on IWM (and upper part of PDW) characteristics and tracers (i.e. insufficient spinup, drift in certain properties etc.)? Upper ocean could be quasi-equilibrated (say few hundred meters) but I am wondering about mid  $\sim$  deep ocean you are more focusing on in this study.

[4] Regarding to dissolved oxygen restoring, are the boundaries (and depth interface at 1500m) all in the Pacific ocean only (e.g. thinking of for example, 30°N and 30°S zonal walls and 1500m layer in the entire Pacific ocean) or globally? Also, how strong (i.e. timescale) is the restoring in these simulations?

[5] Regarding to the respiration rate (in L144), did you set all the simulations respiration rate (similar to fixing oxygen utilization rate I would assume) to NEMO2-REF?

[6] I am a bit confused by the locations of particle release and IETP/IWTP regions you were referring to (L363 - 383, Fig.7 and 8). While the the locations of particle release is in sections (shown as black bold lines (or dot) in Fig.7), I thought the IETP/IWTP are basins in specific rectangles and this is different from the locations of particle release (it contains of course) if I understand correctly. If that is the case, I suggest to revise the main text and Figure to include these information more explicitly (I think adding boxes in Fig.7 could help and you can refer to that interpreting Fig.8).

[7] Just for clarification: do ocean stand-alone simulations (i.e. NEMO and UVIC)
also use preindustrial  $pCO_2$  for spinup (related to mean state diagnostics)?

[8] In section 2.1, Table 1, and part of the main text: The author mix use the NEMO and NEMO2 through the manuscript and I have got a bit confused. Since all the simulations use NEMO2, you should make the terminology consistent through the text after introducing (or just NEMO, I will leave this to the authors).

[9] For Table 1, I would suggest to include model integration time information.

3.1 Line Specific Comments

[L70] Cabre et I., : should be Cabre "et al.,"

[L85] eastern tropical (20°S-20°N): I think you should add longitude information since you mentioned "eastern" tropical Pacific.

[L104] (see Keller Keller 2012 for ... : delete "Keller" (duplicates).

[L124] more than 50 years: suggest to change to "60 years" (the same as the statement in latter section, L160).

[L167] 5 daily means: I think "5-day mean" is more common.

[L262 - 263] Where is the information (figure) of total advective term? Fig. 4g is the vertical advection term difference and I could not find specific information on total term in the figure (although it is possible to infer from all the terms).

[L301] Tsuchuya jets: should be "Tsuchiya jets".

OSD

---

## Referee Comment (RC2) · Anonymous Referee #2 · 5 May 2020

This paper highlights the role of intermediate waters as the O2 supply pathway for the waters of oxygen minimum zones primarily focusing on the Pacific basin. This study consists of three model simulation with different source code, resolution and biogeochemical parameterizations. In general current generation of earth system models tend to have difficulties representing this mode of oxygen supply, thus overestimating the size of low-oxygen waters. Here are main conclusions; (1) the O2 concentration of these water masses in the subtropics is biased in models. If restoring is used to correct the model bias in O2 entering into the subtropics, the tropical O2 representation improves significantly. (2) the ocean jets and eddies play major role for the O2 transport of intermediate water, as supported by the runs with different model resolutions. Coarse resolution models must rely on parameterization for this process. (3) Due to

tropical upwelling, the biases in the deep and intermediate water can impact on the entire upper ocean water column.

I think these points are not really surprising, but the authors have done a detailed, systematic analysis of oxygen responses to model resolution and source water properties to support these conclusions. In my view, this paper is publishable perhaps with a few minor revisions. Below are my technical comments. Main text has several typos. It will benefit from a careful proofreading.

Fig 2b. If I'm reading this figure correctly, it is remarkable that not a single model can capture the peak of O2 at about 800m. I think this feature should be pointed out more clearly in the main text at about page 6. The caption does not indicate which line is WOA. I think it is obvious that the observation is the thick black line, but it needs to be spelled out in the caption.

Fig 3 and main text in page 7. I really like this figure and the discussion in the main text, up to panel f. Then I'm confused. The figure caption says the panels g, h, i are zonal mean tendencies of O2. The main text talks about something different about deep O2. It doesn't even mention how these tendencies are calculated. This probably means there is some version inconsistency between Figure 3 and the main text. This obviously needs a revision.

L284 and in some other places; What is meant by the "upper layer"? I interpreted as the surface, but please be more specific (such as the surface or sigma-theta level or a z-level).

The text related to Fig 4 is confusing, if I read it correctly, the net advective transport divergence is not affected but is not shown (L262-263). Is the change in O2 concentration entirely caused by the eddy parameterization part of the transport? In my opinion, this type of budget analysis may be more interesting if it is applied to contrast the low- and high-resolution runs and separate the mean flow and (resolved or parameterized) eddy contribution.

---

## Author Comment (AC1) · 7 Sep 2020

We thank the referee #1 for her/his review. We annex our reply as a .pdf file.

Please also note the supplement to this comment:
https://os.copernicus.org/preprints/os-2020-17/os-2020-17-AC1-supplement.pdf
* * *

---

## Author Response (AR1)

**Intermediate water masses, a major supplier of oxygen for the eastern tropical Pacific ocean" by Olaf Duteil et al.**

**A. Reply to Referee #1**

**Main Objective of this Study and General Comments**

This study investigates the impact of intermediate water masses (IMW) and it's pathway and supply along Equatorial Intermediate Current System (EISC) on dissolved oxygen content in the Pacific Oxygen Minimum Zone (OMZ) (in the eastern tropical Pacific ocean). The authors utilized a suite of simulations to address these questions.The manuscript consists of) mean state diagnostics and evaluations from suite of models (NEMO (ocean stand-alone simulation), UVIC (coupled, energy moisture balance model, forced wind stress), GFDL (coupled) and ii) sensitivity simulations (or transient simulations over 60 years) (oxygen restoring, conservative tracer release, and Lagrangian tracking of tracers) elucidating the role of subtropical IMW on dissolved oxygen supply (through EICS) in eastern tropical Pacific ocean. Despite the limitations (or discrepancies) in simulating properties of IWM in the current climate models, the authors did a nice set of simulations tackling how bias in IMW and EICS could impact on dissolved oxygen (and possibly impact on projections of OMZs due to climate change). This could provide insights on improving ocean bio-geochemistry in ESMs and I think the work contains interesting and important results.

We thank the reviewer for her/his positive evaluation.

However, I have several comments and some sections and figure presentations should be revisited before publication. Therefore, I suggest a major revision. I state specific comments below and hope this helps to improve the manuscript.

**Major Comments**

[1] The heterogeneous subset of models (simulations) will be an advantage exploring model and resolution dependencies (as author stated in L116−118) on IMW characteristics and tracers (here dissolved oxygen) but also makes the results difficult to interpret to some extent. I still think the results will have impacts from not only the differences in model structures and resolutions, but also the forcing (forcing dataset, prescribed vs. coupled) and model integration time (spinup states) (some specific comment on forcing dataset is stated below). I would like to ask authors to discuss further on these points since for example, the wind and buoyancy forcing bias could be one of the reasons introducing errors in climate (and ocean) models as stated in the introduction.

We agree with the reviewer that extracting information from a heterogeneous subset of simulations is not straightforward and needs a specific conceptual reasoning, that we clarify in a first step. In a second step, we reply specifically to the comments of the reviewer.

1. Conceptual reasoning

We compare the oxygen levels in a set of models characterized by different resolutions, integration time scale, forcings, etc.. Despite all these differences, we found common behaviours (part 3.1): the properties of the intermediate waters are poorly represented in all simulations that we analyzed and we found a correlation between oxygen levels in intermediate waters and oxygen levels in tropical regions (part 3.1 of the ms).

It suggests that intermediate waters affect oxygen levels and OMZ volume in tropical regions. We test this hypothesis using a "what if ?" experiment : "If the oxygen levels are realistic south of 30°S and/or below 1500m does it have an impact on OMZs ?". These sensitivity simulations are performed using a single model framework: same resolution, same forcings, same integration time. (part 3.2)

Another second hypothesis that we investigate is "do the intermediate circulation and associated jets play a large role in setting oxygen levels in the equator region ?". To reply to this question, we performed a set of sensitivity simulations using again a single model framework: same integration time, same forcings, but different spatial resolution. (part 4.2).

In addition (part 4.3) we compare the oxygen levels in a climate model suite: similar model framework, same integration time, different ocean resolution.

In summary, we investigate the mechanisms impacting tropical oxygen levels at intermediate depths in a very heterogeneous set of models, by performing dedicated sensitivity simulations that are easy to interpret.

2. Reviewer comment on the heterogeneity of the models and model set-ups that makes it difficult to pinpoint causes for differences of the simulations.
- Atmospheric forcing

We agree that the atmospheric forcing data play a large role in setting ocean properties. Differences in wind stress between reanalyses data are of the order of 5-20 % (zonal mean wind stress), as shown by the figure below (Chauduri et al., 2013)

Chaudhuri, Ayan & Ponte, Rui & Forget, Gael & Heimbach, Patrick. (2013). A Comparison of Atmospheric Reanalysis Surface Products over the Ocean and Implications for Uncertainties in Air-Sea Boundary Forcing. Journal of Climate. 26. 153-170. 10.1175/JCLI-D-12-00090.1.

[Figure]

Figure 1: Zonally averaged profiles of zonal wind stress from 1999–2006 for ERA-Interim, JRA-25, NCEP1, CORE2, and QuikSCAT (Chauduri et al., 2013).

Large differences exist especially in the eastern tropical Pacific Ocean where the wind is weak. The Figure 2 below shows the relative difference in wind speed between NCEP and CORE (Large and Yeager, 2009), i.e., it shows that winds of the different products in the eastern tropical Pacific differ by up to 50%.

[Figure]

Figure 2: Global distributions of the multiplicative speed applied to NCEP wind vectors to obtain CORE wind vectors (Large and Yeager, 2009)

Large, W.G., Yeager, S.G. /2009). The global climatology of an interannually varying air–sea flux data set. Clim Dyn 33, 341–364. 10.1007/s00382-008-0441-3

To test this impact, we performed an experiment using the UVIC model using 2 different wind products (NCEP and COREv2 – Large and Yeager, 2009) (Figure A1). While the shape of the OMZ shows slight differences, the volume of the OMZ and the mean oxygen levels in the tropical regions and in the mid latitudes are similar. Consistent with the Figure 2, higher oxygen levels at 30°S lead to higher oxygen levels in the tropical ocean and to a smaller OMZ volume (Figure 3)

[Figure]

Figure 3 : Oxygen levels in UVIC (10000 years integration) a- mean 500-1500 m forcing NCEP. b- section 120°W forcing NCEP. c- mean 500-1500 m forcing COREv2, d- section 120°W forcing COREv2.

[Figure]

Figure 4 : a - Oxygen levels in UVIC (10000 years integration) at 30°S (zonal mean in the Pacific Ocean from surface to 2000 m depth) and in the tropical regions (20°S-20°N, averaged over the whole Pacific Ocean). b - Oxygen levels in UVIC (10000 years integration) at 30°S (zonal mean in the Pacific Ocean, from surface to 2000 m depth) and volume of the OMZ in the Pacific Ocean. The configuration forced by COREv2 is shown in black, the configuration forced by NCEP is shown in red.

**Coupled ocean atmosphere experiments**

Coupled ocean-atmosphere experiments introduce further discrepancies compared to the use of realistic atmospheric forcings. However, the mean surface velocity is similar in the suite of GFDL models (especially GFDL01 and GFDL025) that we analyzed, suggesting that the effect of atmospheric forcing is likely not dominant when comparing this subset of models (part 4.3).

[Figure]

Figure 5: ocean zonal surface velocity (ms-1) in GFDL01, GFDL025 and GFDL1

**Model integration time**

In complement, the spinup state of the model also impacts the oxygen levels as the deep ocean needs thousands of years to be in equilibrium. It may explain why UVIC (integrated for 10000 years) is characterized by much larger oxygen levels than the GFDL model suite (integrated for 190 years). As an example, the Figure 6 shows the evolution of oxygen levels during spinup in NEMO2. Larger oxygen levels at 30°S (e.g after 1000 years of integration) are characterized by a smaller OMZ volume (which is consistent with Fig 2) (Figure 7)

[Figure]

Figure 6 : oxygen levels at a - intermediate depth (average 500 – 2000 m) and b - 120°W in NEMO2 after 50, 100,500 and 1000 years integration

[Figure]

Figure 7 : a - Oxygen levels in NEMO2 at 30°S (zonal mean in the Pacific Ocean from surface to 2000 m depth) and in the tropical regions (20°S-20°N, averaged over the whole Pacific Ocean from surface to 2000 m depth). b - Oxygen levels in NEMO2 at 30°S (zonal mean in the Pacific Ocean from surface to 2000 m depth) and volume of the OMZ in the Pacific Ocean. The color of the cross depends of the integration duration (black : 50 years, red : 100 years, green : 500 years, blue 1000 years).

3. Conclusion

The differences induced by the different forcings and integration time have (not surprisingly) an impact on water masses and oxygen levels. Despite the heterogeneity of our simulations, our results nevertheless suggest a strong coupling between subtropical and tropical oxygen content and justify our questioning and the experiments performed in the part 3 and 4 of this study (see 1. Conceptual reasoning)

[2] Regarding to sensitivity of tropical IWM oxygen to subtropical and deep dissolved oxygen levels, the authors refer AAIW, NPIW (and the upper part of the PDW) as IWM in this study. I was wondering what will be the relative contributions of each water masses to dissolved oxygen supply, ventilation in the eastern tropical Pacific ocean (particularly North (NPIW) vs South (AAIW)). My impression is that AAIW could be more dominant (e.g. Talley, 2013) but I would like to know what sensitivity simulations indicates. At least, I think it is possible to obtain insights from the Lagrangian tracking diagnostics (or if possible, conducting additional restoring simulations with 30◦S boundary only for example).[Reference] Talley, Lynne D., (2011), Descriptive Physical Oceanography: An Introduction, Academic Press.

We perform a complementary experiment using NEMO2 where the oxygen levels are forced (see Minor Comment 4) to WOA solely south of 30°S (experiment NEMO2_30S. The experiment where oxygen is restored both to the south and to the north. NEMO2_DEG30 has been renamed NEMO2_30S30N). It shows clearly that AAIW has a dominant impact in setting tropical Pacific Ocean intermediate oxygen levels and the OMZs volume. This is not surprising as AAIW recirculates till about 20°N and NPIW has a much smaller, regional extension (Talley, 2011)

[Figure]

Figure 8: NEMO2-30S minus NEMO2-REF and NEMO2_30S30N minus NEMO2_REF (average 500-2000 m).

[3] The core of the study is based on a suite of sensitivity simulations from NEMO(NEMO2). In the first reading, I struggled a bit on connecting aim and each sensitivity experiments. The dissolved oxygen restoring simulations aim on investigating sensitivity of tropical IWM oxygen to subtropical and deep dissolved oxygen levels (as stated in section 3.2) and the conservative tracer release simulations are more dedicated to investigate spreading of tracers towards the eastern tropical Pacific (transport by the EICS, as stated in section 4.1). While the standard structure of the manuscript is to introduce overall data and methods in the beginning, (section 2), I suggest to move some of the objective and details of sensitivity experiments to each corresponding sections (referring to sections 3.2 and 4.1) so it is much easier to follow the aim bridging to sensitivity experiments (I think it is still fine to keep brief general descriptions in section 2 including Table 1). Alternatively, the methods section could be revised to include additional descriptions connecting to corresponding result sections. I will leave this decision to the authors regarding to the structure of the paper but I think the flow could be improved.

We decided to keep the original, classical, structure but agree that the methodology section need to be better connected with the results / discussions. We therefore added the following lines :

"L126: The mean state of the oxygen distributions is discussed below in section 3.1 "IWM Oxygen levels in models""

"L159 : The sensitivity of tropical IWM oxygen to subtropical and deep oxygen levels is discussed in section 3.2"

"L187 : The transport by the EICS is discussed in section 4.2 (tracers levels and Lagrangian pathways)."

[4] Another major issue is the figures. Figure labels and captions are not easy to interpret (and in some part, the authors are referring to figure does not appear,e.g.L267, Fig.4i). Therefore, I suggest the authors to carefully revisit all the figures and add necessary caption, labels for better presentation. For example, for time series plots (e.g.Fig. 2,3g−i,8), the difference in color (models, configurations etc.) should also be informed in the label (not just in figure captions) because it is not easy to follow.

The figures/labels/captions are revised in the final version of the ms. See the new set of figures at the end of the reply. When necessary, the figure / subpanel number has been modified in the text to match the new set of figures.

Similar issues for multiple maps (such as Fig.5), it will be reader friendly to label maps with "zonal advection", "meridional advection" etc.

The transport terms (Fig 4) are labeled in the final version of the ms. See the new set of figures at the end of the reply.

Also, some of the model names(labels) are not obvious because those are overlaid on color shading (e.g. Fig.9).

The names are labeled in a more obvious way in the final version of the ms. See the new set of figures at the end of the reply.

I put few more specific suggestions below and hope this helps to point out the difficulties I am referring to.

Thanks to the reviewer for these suggestions. We have rechecked all captions to make sure that they are correctly describing the panels.

[4.1] Fig.1caption, (L762−763) oxygen levels (mean 500 - 1500m) at 160W, I think color shading in b) is not vertical mean (because it is depth-latitude section). Also, is dissolved oxygen in Fig.1from observations such as World Ocean Atlas?

A new caption has been written.

[4.2] Regarding to Fig.4, I have several suggestions to improve figure presentation. I am still a bit confused what is in color shading and contours. For example, in L789,it states the vertical current as contour in c) but the contours do not look like vertical current values. Also the continent shading in g) is missing (no gray shaded). Similar confusion occurred to me in other panels and I suggest to revisit and clearly state what is presented in color shading and contours for each panels with units.

The Figure 4 has been revisited (missing shading of the continent, captions, legend).  See the new set of figures at the end of the reply.

Also, why did you only present the results from NEMO2-30DEG (not including NEMO2-30DEG1500M or NEMO2-30DEG1500M minus NEMO2-30DEG)?

The experiment NEMO2-30DEG has been renamed NEMO2_30S30N for clarity reasons (see above comment). We show in Fig 4 both the transport terms of NEMO2_30S30N and of NEMO2_30S30N minus NEMO2_REF.  We do not show NEMO2_30S30N1500M as from Figure 3i it becomes clear that the processes transferring oxygen from the deeper layer toward the intermediate ocean are vertical advective processes. This is now stated explicitly in the new version of the ms.

[4.3] Add information labels for Fig.7a)−c)the first release, and d)−e)the second release, respectively.

Information labels have been added and the figure revisited. See the new set of figures at the end of the reply.

[4.4 ]Add information labels (like figure title) for Fig.9, zonal sections and meridional sections, respectively.

Information labels have been added and the figure revisited. See the new set of figures at the end of the reply.

Minor Comments

[1] I am curious whether CORE v2 climatological forcing (used for NEMO) and NCEP/NCAR climatological forcing (wind stress, used for UVIC) makes a difference in paper spinup states. As far as I know, CORE v2 forcing is based on NCEP/NCAR reanalysis but it has several corrections and adjustments in the forcing and difference between the two could lead to different results, particularly after long-term spinup. Do authors think this is a minor thing ?

The different climatological  forcings have indeed a significant impact (see Figure 3 of our response). However we think that differences in resolution play a larger role by resolving additional processes (in particular deep equatorial jets)

[2] Are all the GFDL model simulations integrated for the same period following high-resolution (GFDL01) for comparison (I assumed 200 years from Busecke et. al.,2018) or the low-resolution configurations are integrated for longer durations ?

All configurations have been integrated for 190 years (more precisely 48 years physics only + 142 years biogeochemical cycles), including the lower resolution version. This is now clearly stated in the new Table 1.

[3] Because of the high resolutions configurations for GFDL01, the integration time is limited but does this impact on IWM (and upper part of PDW) characteristics and tracers (i.e. insufficient spinup, drift in certain properties etc.)? Upper ocean could be quasi-equilibrated (say few hundred meters) but I am wondering about mid~deep ocean you are more focusing on in this study.

We agree with the reviewer, the model spin-up has a large impact on ocean properties. The mid-depth (500 – 1500 m) ocean is not fully equilibrated after 100/200 years. However, the experiments part 3.2 : "If the oxygen levels are realistic south of 30°S and/or below 1500m does it have an impact on OMZs ?" and 4.2/4.3 "do the intermediate circulation and associated jets play a large role in setting oxygen levels in the equator region ?" (see 1 - Conceptual reasoning) clearly show that a timescale of 100 - 200 years is sufficient to investigate the connectivity between mid-latitude / tropical regions, as well as the role of the intermediate current system in controlling oxygen (and more generally tracers) concentration. Even if a short integration timescale does not allow to characterize the steady state and the relative importance of all the processes at play, it permits nevertheless to assess the importance of specific processes (especially that the experiments, e.g the GFDL suite of models, have been integrated for the same duration (190 years).

[4] Regarding to dissolved oxygen restoring, are the boundaries (and depth inter-face at 1500m) all in the Pacific ocean only (e.g. thinking of for example, 30∘N and 30∘S zonal walls and 1500m layer in the entire Pacific ocean) or globally ? Also, how strong (i.e. timescale) is the restoring in these simulations ?

The term "restoring" is maybe inadequate and has been replaced by "forcing" L132, 139 and 142. of the manuscript including corrections as the oxygen levels are forced to the WOA monthly climatology. The latitude where the forcing is applied has been set globally (however as it is a "forcing", it does not make any difference if it were applied solely in the Pacific Ocean).

[5] Regarding to the respiration rate (in L144), did you set all the simulations respiration rate (similar to fixing oxygen utilization rate I would assume) to NEMO2-REF?

Respiration rates (as all other biogeochemical fluxes) are the same in all the experiments, as stated L147 : " The respiration rate (oxygen consumption) is identical in NEMO2-REF, NEMO2-30S30N and NEMO2-30S30N1500M". Solely the oxygen concentrations are forced by WOA values at 30N/30S/1500m depth. Forcing phosphate levels would complicate the picture (as stated L150-151), as the resulting differences of productivity and respiration would counteract the difference of advection of modified oxygen concentrations. Quantifying the sensitivity of respiration to a change in nutrients is an important aspect, but is outside the scope of this study which focuses on the transport of oxygen by intermediate water masses. Furthermore our Figure 2 (correlation oxygen content at 30°S and in tropical regions) suggests that differences in ocean circulation are dominant compared to differences in biology in the simulations that we consider.

[6] I am a bit confused by the locations of particle release and IETP/IWTP regions you were referring to (L363−383, Fig.7 and 8). While the the locations of particle release is in sections (shown as black bold lines (or dot) in Fig.7), I thought the IETP/IWTP are basins in specific rectangles and this is different from the locations of particle release (it contains of course) if I understand correctly. If that is the case, I suggest to revise the main text and Figure to include these information more explicitly (I think adding boxes in Fig.7 could help and you can refer to that interpreting Fig.8).

A new Figure 8a has been added, which shows the IETP/IWTP regions and the release locations R1 and R2. We state more explicitly in the text:

L389 : "The release location R1 is the eastern tropical Pacific (100°W, 5°N-5°S, 1000 m depth ) . R1 is included in the larger Intermediate Eastern Tropical Pacific (IETP) ocean region (160°W – coast / 10°N-10°S / 200 – 2000 m )"

L405 : "The location of the second release R2 (160°E, 5°N-5°S, 1000 m depth) is  included  in the Intermediate Western Tropical Pacific (IWTP) ocean region (160°W – coast / 10°N-10°S / 200 – 2000 m) (Fig 7b)"

[7] Just for clarification: do ocean stand-alone simulations (i.e. NEMO and UVIC) paper also use preindustrial pCO2for spinup (related to mean state diagnostics)?
Preindustrial pCO2 is used. This is now stated in the text L117.

[8] In section 2.1, Table 1, and part of the main text: The author mix use the NEMO and NEMO2 through the manuscript and I have got a bit confused. Since all the simulations use NEMO2, you should make the terminology consistent through the text after introducing (or just NEMO, I will leave this to the authors).

Three versions at 2°,0.5° and 0.1° horizontal resolution of the general NEMO model engine (Madec et al., 2017) are used : NEMO2 (with biogeochemical cycles), NEMO05, NEMO01 (physics only). We refer specifically to these versions in the text.

L132 : we refer to generally to the NEMO model engine and state that explicitly to avoid confusion : "we perform two different sets of sensitivity simulations using the general NEMO model engine"

[9] For Table 1, I would suggest to include model integration time information.

The model integration time has been added in the Table 1 (see last section of this document)

3.1 Line Specific Comments

[L70]Cabre et l., : should be Cabre "et al.,"

This is corrected in the final version of the ms

[L85]eastern tropical (20∘S-20∘N): I think you should add longitude information since you mentioned "eastern" tropical Pacific.

We added "east of 160°W" in the final version of the ms

[L104](see Keller Keller 2012 for ... : delete "Keller" (duplicates).

This is corrected in the final version of the ms

[L124]more than 50 years: suggest to change to "60 years" (the same as the statement in latter section, L160).

This is corrected in the final version of the ms

[L167]5 daily means: I think "5-day mean" is more common.

This is corrected in the final version of the ms

[L262−263]Where is the information (figure) of total advective term? Fig. 4g is the vertical advection term difference and I could not find specific information on total term in the figure (although it is possible to infer from all the terms).

The objective of the Figure 4 is to better explain the differences between the model experiments (Fig 3g). As the patterns are mostly zonal, we did not show in Fig 4 the total term (the zonal mean of the total term is already displayed in Fig. 3g).

[L301]Tsuchuya jets: should be "Tsuchiya jets".
This is corrected in the final version of the ms

**B. Reply to Referee #2**

This paper highlights the role of intermediate waters as the O2 supply pathway for the waters of oxygen minimum zones primarily focusing on the Pacific basin. This study consists of three model simulation with different source code, resolution and biogeochemical parameterizations. In general current generation of earth system models tend to have difficulties representing this mode of oxygen supply, thus overestimating the size of low-oxygen waters.

Here are main conclusions; (1) the O2 concentration of these water masses in the subtropics is biased in models. If restoring is used to correct the model bias in O2 entering into the subtropics, the tropical O2 representation improves significantly.

(2) the ocean jets and eddies play major role for the O2 transport of intermediate water, as supported by the runs with different model resolutions.Coarse resolution models must rely on parameterization for this process.

(3) Due to tropical upwelling, the biases in the deep and intermediate water can impact on the entire upper ocean water column.

I think these points are not really surprising, but the authors have done a detailed, systematic analysis of oxygen responses to model resolution and source water properties to support these conclusions. In my view, this paper is publishable perhaps with a few minor revisions.

We thank the author for her/his positive feedback.

Below are my technical comments. Main text has several typos. It will benefit from a careful proofreading.

The final version of the ms has been carefully proofread.

Fig 2b. If I'm reading this figure correctly, it is remarkable that not a single model can capture the peak of O2 at about 800m. I think this feature should be pointed out more clearly in the main text at about page 6. The caption does not indicate which line is WOA. I think it is obvious that the observation is the thick black line, but it needs to be spelled out in the caption.

The "missing" O2 peak is indeed a remarkable feature in the models. We point that out more clearly in the new version of the ms.

We added to the paragraph L209-213 : "The IWM oxygen maximum is apparent at 30°S throughout the lower thermocline (600 – 1000 m) in observations (Fig 2b), consistent with the circulation of IWM with the gyre from the mid/high latitude formation regions towards the northwest in subtropical latitudes, and followed by a deflection of the waters in the tropics towards the eastern basin", the sentence L213 : " This oxygen peak is missing in all the models analyzed here".
The figure 2 has been updated and is reproduced at the end of the reply.

Fig 3 and main text in page 7. I really like this figure and the discussion in the main text, up to panel f. Then I'm confused. The figure caption says the panels g, h, i are zonal mean tendencies of O2. The main text talks about something different about deep O2. It doesn't even mention how these tendencies are calculated. This probably means there is some version inconsistency between Figure 3 and the main text. This obviously needs a revision.
The text L229 (page 7)  to which the reviewer refers reads : "The difference NEMO2-30DEG1500M – NEMO2-30DEG (Fig 3f-h) shows a deep positive anomaly in oxygen, as oxygen levels are lower than in observations by 30-40 mmol.m-3 in the eastern tropical regions". The reference to Fig 3 f-h is wrong. It has been corrected in the new version of the ms.

We added a brief description of the budget terms L265 :
"The oxygen budget is :

$$\frac{dO_2}{dt} = Adv_x + Adv_y + Adv_z + Diff_{Dia} + Diff_{Iso} + SMS$$

where $Adv_x, Adv_y, Adv_z$, are respectively the zonal, meridional and vertical advection terms, $Diff_{dia}$ and $Diff_{iso}$ are the diapycnal and isopycnal diffusion. SMS (Source Minus Sink) is the biogeochemical component (respiration at depth, below the euphotic zone)

L284 and in some other places; What is meant by the "upper layer"? I interpreted as the surface, but please be more specific (such as the surface or sigma-theta level or z-level).
The upper layer corresponds to the mixed layer. This is clearly specified in the new version of the ms.

The text related to Fig 4 is confusing, if I read it correctly, the net advective transport divergence is not affected but is not shown (L262-263). Is the change in O2 concentration entirely caused by the eddy parameterization part of the transport? In my opinion this type of budget analysis may be more interesting if it is applied to contrast the low-and high-resolution runs and separate the mean flow and (resolved or parameterized) eddy contribution.

We show below the total advective transport in NEMO2_REF and its anomaly (NEMO2_30S30N minus NEMO2_REF) (Figure 7)

[Figure]

Figure 7: left : total advection term in NEMO2_REF. Right : difference in the total advection term between NEMO2_30S30N – NEMO2_REF

The Fig 7 right panel in our response letter shows clearly that the total advection terms are similar in NEMO2_30S30N and NEMO2_REF at the equator. In contrast, the differences are large in the gyres as the anomaly is advected by the strong westward currents. In the tropics, most of the anomaly is due to isopycnal mixing (or "eddy parameterization" transport as stated by the reviewer), see Fig 4b in the new version of the ms. This is maybe not surprising as the intermediate currents are weak in NEMO2 (coarse resolution). Higher resolutions models will likely be characterized by the imprint the of zonal jets. We agree with the reviewer, a similar experiment but performed at high resolution would be very useful to quantify precisely the impact of these jets. Unfortunately a high resolution eddy resolving simulation coupled with biogeochemical cycle was not available due to computational expenses (which is the reason why we compare coarse and high resolution simulations coupled to a single passive tracer in part 4 of this ms).

**C. Updated Figures and Table**

[Figure]

Figure 1 : a- schema summarizing the intermediate water masses (IWM) pathway from the subtropics into the equatorial regions. EICS : Equatorial Intermediate Current System. SEC : South Equatorial Current. Dashed line : isopycnal diffusive processes. Observed (World Ocean Atlas) oxygen levels (mmol.m$^{-3}$) in the lower thermocline (mean 500-1500m) are represented in color. b - schema (adapted from Menesguen et al., 2019) illustrating the complexity of the EICS, extending below the thermocline till more than 2000 m depth (see section 4.1 for a detailed description). Observed (World Ocean Atlas) oxygen levels at 160°W are represented in color.

[Figure]

Figure 2 : a- oxygen levels (mmol.m$^{-3}$)  in observations (World Ocean Atlas - WOA) (mean 500 – 1500 m) and models (UVIC, NEMO2, GFDL1, GFDL025, GFDL01). Contours correspond to WOA values. b: average "30°S" (120°E-65°W, 30°S) c : average "tropics" (160°W-coast, 20°N-20°S). d: average "30°S" vs "tropics". e: average "30°S" vs volume of tropical suboxic ocean (oxygen lower than 20 mmol.m$^{-3}$) regions (1e15m3). b-e : UVIC : black, NEMO2 : cyan, GFDL1 : red, GFDL025, green; GFDL01 : blue, WOA: bold line (b,c) and star (d,e).

[Figure]

Figure 3 : a,b: Oxygen (mmol.m$^{-3}$) in the experiments NEMO2_REF (color) and World Ocean Atlas (contour) (a- average 500-1500 m, b- 100°W). c,d: Oxygen (mmol.m$^{-3}$) difference (c- average 500 – 1500m, d- 100°W) between the experiments NEMO2_30S30N minus NEMO2_REF. e,f : Oxygen (mmol.m$^{-3}$) difference (e- average 500-1500m, f- 100°W) between the experiments NEMO2_30S30N1500M minus NEMO2_REF. g- basin zonal average (average 500 - 1500 m) of the oxygen total supply (bold) (mmol.m$^{-3}$.year$^{-1}$), advective processes (blue) and isopycnal diffusion (red) in NEMO2_REF, NEMO2_30S30N, NEMO2_30S30N1500M. The dashed line is the oxygen total supply in NEMO2_REF.

[Figure]

Figure 4 : a- Oxygen supply processes (mmol.m$^{-3}$.year$^{-1}$ – average 500 - 1500m) in NEMO2_REF : zonal advection, meridional advection, vertical advection, isopycnal diffusion. The mean meridional and zonal currents are displayed as vectors (meridional, zonal advection). The mean vertical current (0 isoline) is represented as bold contour (vertical advection). Oxygen levels (mmol-m.$^{-3}$) are displayed in black contour. b- Difference in oxygen supply processes (mmol.m$^{-3}$.year$^{-1}$ – average 500-1500m) between NEMO2_30S30N and NEMO2_REF : zonal advection, meridional advection, vertical advection, isopycnal diffusion. The NEMO2_30S30N – NEMO2_REF oxygen anomaly (mmol.m$^{-3}$) is displayed in contour.

Zonal velocity component at 1000 m (colors) and oxygen (contours)

Zonal velocity component at 100°W (colors) and oxygen (contours)

[revised manuscript text omitted]

---

## Author Response (AR2)

The Reviewers comments (Reviewer 1 and Reviewer 3) are reproduced below in red. We reply in black. The line number corresponds to the line in the track-change file

Reviewer 1
First of all, my apologies in the previous review comments that I made typos in acronyms, "IWM" (Intermediate Water Mass) by IMW. Hope the comments was clear enough to explain the points. I thank the authors for conducting a suite of additional simulations addressing the impact of the atmospheric forcing, model integration time, and oxygen forcing simulations.

The authors have done substantial amount of additional simulations and revision and I think the additional results (figures in response and supplementary materials) addressed the questions from my comments. They showed that the impact of atmospheric forcing and spin-up state exists but does not affect the question and conclusion at least in this study (i.e. strong coupling between subtropics and tropics).

The message in the summary session is organized and I think the results support the conclusions.
We thank the reviewer for her/his positive evaluation.

I still have few comments that I would like to ask the authors to address before publications. The comments are mainly about the presentation and clarification of the paper and most of them are relatively minor comments.

1. I thank the authors for stating "conceptual reasoning" in the response. I understand why the authors choose to utilize a suite of models and I think it is useful to combine a subset of the models and a single model sensitivity simulations to dig into further mechanisms. The basic presentation style of this paper is to present analysis from a subset of models (UVIC, NEMO2, and GFDL models) to explore the representations of IWM, EICS, and dissolved oxygen levels and  then show sensitivity simulations (NEMO2) to explore mechanisms, which I think is good.
We thank the reviewer for her/his comment.

The only part I have got stuck a bit is the section 4.3, which comes back to the result from the subset of models. The authors stated in the comment "investigating mechanisms in a heterogeneous set of models by performing dedicated sensitivity simulations" but the analysis and discussion I see is the analysis of MKE (Mean Kinetic Energy) and dissolved oxygen (i.e. more of the analysis rather than sensitivity simulations).
The MKE discussion itself is good but I also had to look into the caption in Figure 9 carefully to see the results from UVIC GD13 (and this was not completely clear from just reading the section 4.3).
I am not sure what the authors meant by "performing dedicated sensitivity simulations" from a heterogeneous set of models (because I only see sensitivity simulations from NEMO2) and I kind of see a slight jump from the previous sections.

We indeed stated in our reply to the reviewer "In summary, we investigate the mechanisms impacting tropical oxygen levels at intermediate depths in a very heterogeneous set of models, by performing dedicated sensitivity simulations that are easy to interpret". This sentence does however not appear in the manuscript. Rather than applying specifically to part 4.3, this sentence was meant to summarize briefly our conceptual thinking. We perform sensitivity experiments using NEMO2 (part 3), NEMO05 and NEMO01 (part 4). The above sentence could be rephrased: "In summary, we analyze the tropical oxygen distribution at intermediate depth in a very heterogeneous set of models. In order to better understand the mechanisms at play, we use a single model framework to perform additional sensitivity experiments."
Section 4.3 has been rewritten for clarity reasons (see below)

As the authors stated in L413 in a short statement, a suite of tracer release and Lagrangian tracking simulations could help interpreting the subset of the models so I suggest to include more statement on this part (perhaps including MKE and dissolved oxygen from NEMO2 sensitivity simulations help along with Figure 9?) Also, please make it more clear that you are showing the

results from UVIC GD13 (perhaps pointing out the specific panel and mention blue contours, label all the panels alphabetically or just point out "UVIC" panel on the left column). I think the material presented is nice, it is more about the presentation and discussion to make a nice flow from sensitivity simulations back to a subset of models. I also think including results from UVIC GD13 is useful for discussion so it will be good to keep it in (it was just a bit unclear in the text and figure captions).

The part 4.3 has been completely rewritten. We took in account reviewers's comments (in particular regarding the clarity of the text)

"The experiments discussed in 4.2 were not coupled with biogeochemical cycles for computational cost reasons. In order to assess the robustness of our findings (EICS plays a large role in setting tropical oxygen levels), we next analyze equatorial oxygen in a set of climate models similar to CMIP models. To this end we use the GFDL model suite, characterized by a resolution increase (GFDL1, GFDL025 and GFDL01 - see Table 1).

The striking difference between GFDL01 and GFDL025 / GFDL1 are the high oxygen levels in the eastern part of the ocean below 1000 m in GFDL01 compared to GFDL025/GFDL1 (Fig 2). The oxygen levels show weaker zonal gradient in GFDL01, consistent with the tracer experiment that we performed in 4.2. and a more ventilated intermediate equatorial ocean. High values of mean kinetic energy are associated with higher oxygen values (Fig 9). This is particularly clear in GFDL01 at around 1500 m depth, where strong values of MKE are present and form the "bottom" of the low oxygen volume (oxygen lower than 50 mmol.m-3). Conversely GFDL025 and GFDL1 do not present high MKE values below 1000 m in the eastern part of the basin; the low oxygen volume extends till depths greater than 2000 m. It suggests that intermediate currents participate in the ventilation of the eastern tropical ocean and thus in limiting the vertical extension of the OMZ.

Oxygen levels do not increase linearly with the currents strength, i.e while currents strength increase in GFDL1, GFDL025 and GFDL01, oxygen levels are relatively similar in GFDL1 and GFDL025 (see Fig 5 and Fig 9). The relatively small net balance between large fluxes of respiration and oxygen supply (Duteil et al., 2014) may be responsible for this behavior. If the supply is slightly higher compared to the consumption by respiration, it will lead to an increase of oxygen concentration. If it is slightly lower, the oxygen levels will decrease. A small difference in supply (e.g slightly weaker currents) may therefore lead to a large difference in oxygen levels when integrated over decades. For this reason, the impact of the EICS is more visible below 1000 m as the respiration decreases following a power-law with depth (Martin et al., 1987) and is therefore easier to offset even by a moderate oxygen supply.

Resolving explicitly the EICS results in a similar oxygen distribution to what Getzlaff and Dietze (2013) (GD13) achieved with a simple EICS parameterization (Fig 9a): to compensate for the "missing" EICS in UVIC, a coarse resolution model, they enhanced anisotropically the lateral diffusivity in the equatorial region. The oxygen levels from UVIC GD13 are shown in blue contours

on top of the UVIC oxygen distribution (black) in Fig 9. Implementing this approach tends to homogenize oxygen levels zonally, with an increase of the mean levels by 30-50 mmol.m$^{-3}$ in the eastern basin and a decrease of oxygen concentrations in the western basin. While this approach may be useful to better represent the oxygen mean state, it however does not take in account the potential variability and future evolution of the EICS. "

We chose not to include NEMO2 as the message here is specifically on the role of the intermediate current system and its impact on intermediate oxygen levels. The intermediate currents system is not represented in NEMO2. Instead we focus on two model subsets to address the question how changing the resolution or including a parameterization affects the equatorial oxygen transport in climate models.

As stated below by the reviewer, an issue in inter-model comparison are the compensating effects between oxygen supply / respiration. As a result, when physics are deficient, one could tune the biogeochemistry to achieve a realistic field (Duteil et al., 2012). This effect is limited when comparing solely GFDL1, GFDL025 and GFDL01 models as the biogeochemical model is the same. Similarly comparing UVIC and UVIC-GD13 highlight the role of the increased transport at depth. It is however not straightforward to compare NEMO2 and the GFDL models suite in this context. NEMO2 is characterized by a weak MKE but a relatively well oxygenated bottom layer (despite low oxygen levels at 30°S), pointing out an important role of biology to maintain the strong OMZ around 90°W.

[Figure]

Fig : a - Mean Kinetic Energy (m2.s-2 x 1000) (average 10°N-10°S) in NEMO2, b - similar to a. but average 160°W- coast. Oxygen levels (mmol.m-3) are displayed in black contour.

2.I would like to ask the authors to include a short statement (or suggestions) for analyzing CMIP class multi-models in summary section. I think the authors have done nice analysis on heterogeneous set of models (with the aid of sensitivity simulations exploring further mechanisms) and the results can possibly point out what we should analyze to explore the multi-model characteristics (or bias) understanding tropical OMZs. I understand the additional difficulties digging in multi-models since the models include biological effect (compensation authors discussed) but do you think analysis like in Figure 2d or 9 supports on understanding IWM and EICS impacts (or if authors have other suggestions or ideas from their simulations I would be interested to know).

We added the following paragraph L546 :
"This study shows that there is a need to look with greater care into IDW properties to understand the tropical oxygen distribution in models, in particular in CMIP class models. As shown by Kwiatkowski et al. (2020), CMIP6 models (typical horizontal resolution of 1°) do not agree on the future change in tropical oxygen levels (mean 100 – 600m, their Fig 2). This may partly originate in a misrepresentation of the properties of the IDW in the different models and the strength of the connection between western and eastern Pacific Ocean. Simple analyses, similar to our Fig 2 (oxygen levels at 30°S and oxygen levels in the eastern tropical Pacific) and Fig 9 (Mean Kinetic

Energy at intermediate depth) may give some insight into the mechanisms at play. In addition, analyses of experiments performed in the context of the High Resolution Model Intercomparison Project (resolution greater than 0.25°) (Haarsma et al., 2016), part of CMIP6, will give a more complete insight on whether a significant Equatorial Intermediate Current System develops at higher resolution. While HighResMIP are not coupled with a biogeochemical module, velocity fields are available at a monthly resolution, which allows to perform "offline" tracer or Lagrangian particle experiments"

Haarsma, R. J., Roberts, M. J., Vidale, P. L., Senior, C. A., Bellucci, A., Bao, Q., Chang, P., Corti, S., Fučkar, N. S., Guemas, V., von Hardenberg, J., Hazeleger, W., Kodama, C., Koenigk, T., Leung, L. R., Lu, J., Luo, J.-J., Mao, J., Mizielinski, M. S., Mizuta, R., Nobre, P., Satoh, M., Scoccimarro, E., Semmler, T., Small, J., and von Storch, J.-S.(2016). High Resolution Model Intercomparison Project (HighResMIPv1.0)forCMIP6, Geosci. Model Dev., 9, 4185–4208, https://doi.org/10.5194/gmd-9-4185-2016

Kwiatkowski, L., Torres, O., Bopp, L., Aumont, O., Chamberlain, M., Christian, J. R., Dunne, J. P., Gehlen, M., Ilyina, T., John, J. G., Lenton, A., Li, H., Lovenduski, N. S., Orr, J. C., Palmieri, J., Santana-Falcón, Y., Schwinger, J., Séférian, R., Stock, C. A., Tagliabue, A., Takano, Y., Tjiputra, J., Toyama, K., Tsujino, H., Watanabe, M., Yamamoto, A., Yool, A., and Ziehn, T.: Twenty-first century ocean warming, acidification, deoxygenation, and upper-ocean nutrient and primary production decline from CMIP6 model projections, Biogeosciences, 17, 3439–3470, https://doi.org/10.5194/bg-17-3439-2020, 2020.

3. I would like to ask for clarifications on "oxygen forcing" instead of "oxygen restoring". Actually the new term "forcing" raise additional question (at least to me). Does this mean the authors experimental design is NOT "restoring" but just replacing the oxygen values, for example at 30N and 30S to observed values (and always fixed to observed values during the model integration)? What will be the difference? The forcing sounds to me like the atmospheric forcing (such as wind stress) changing with time but I assume that the dissolved oxygen forcing at the boundaries does not change with time (just fixed to the observed climatology) correct?

Indeed we replace the oxygen values at 30°S and 30°N which is why we label this experiment "forcing" than "restoring". We use a time varying "forcing" (= we use oxygen values from monthly mean climatological observations at 30°S/N) to reproduce the basic aspects of the seasonal cycle. We use this strategy as the goal of our experiment is to test what would be the impact on tropical oxygen of an "observed" oxygen boundary. In a typical "restoring" approach, a term is added to the prognostic equation to match (or push toward) the observational values. The resulting tracer concentration is not necessarily equal to the concentration toward which it is restored (depending on the strength of the restoring). A "forcing" is actually an extreme case of "restoring". The section is now called "2.2.1 Forcing of oxygen to observed values in the subtropical regions".

4. L98: Is the CORE-II climatological forcing "normal year" forcing or you constructed climatology based on CORE-II forcing from 1948-2007? Also, I think this does not impact the result but is there a reason why you used CORE-II forcing from 1948-2007 instead of 2009 (since CORE-II extend to 2009)?
The CORE-II climatological forcing that we use is the Normal Year Forcing. This is now stated explicitly (L132)

5. What is the initial condition of sensitivity simulations (60 years simulations) by NEMO2? Do all these sensitivity simulations start from spun-up simulation from the mean state comparison (i.e. 1000 years integration from NEMO2)?

Yes, all the sensitivity simulations start from the spinup state (otherwise the simulations would not be exploitable due to the strong drift in oxygen levels in the first hundred of years of integration). This now stated explicitly in the text.

6. L39: Is it semicolon here? I would use period and separate the sentences but I will leave this to the author (including checking with native speakers).
We agree and separate the sentences.

7. L126: "eNEMO" should be "NEMO"?
Yes.

Reviewer 3

General comment : we would like to thank the reviewer for her / his comments, which helped us to improve the manuscript. Thanks to these comments, we realized that the scope of our manuscript was not well defined. Rather than a study focusing on the characterization of the subducting water masses, we aim here to understand the oxygen distribution in the tropical ocean and the role of the intermediate depth waters (defined here as the 500 - 1500 m) in modulating this distribution. We choose here a depth range rather than a density criterion as the strong attenuation of respiration with depth introduces difficulties when comparing different density layers. The layer 500 - 1500 m is now called "Intermediate Depth Water" instead of "Intermediate Water Masses".
Our focus on tropical oxygen levels was not clear from the previous version of the manuscript and we clarified this point (introduction and along the text), in particular for readers originating from another community than the "oxygen community" (this manuscript is part of a joint special issue "Ocean deoxygenation: drivers and consequences – past, present and future" but its scope should be of course clear for any reader outside of this community).

The authors present a variety of analyses of the controls on O2 at intermediate levels in the Equatorial Pacific. Despite some interesting results, the manuscript was remarkably scattered and unfocused, and thereby difficult to read. Part of the problem stems from an inadequate analysis of water masses, and part of the problem stems from a confusing set of experiments with a disparate set of models.

The authors really need to state somewhere the abstract conclusions something like "biases in the modeled O2 concentrations in intermediate layers in the Equatorial Pacific (which we carefully define in density space) obviously reflect issues with the formation and fate of intermediate waters within the interior. This study uses A/B methods to disentangle the causes of unsatisfactory IWM O2 in the tropics, and arrives at the conclusion that the problem is X% formation, Y% fate/ventilation, and Z% bad biogeochemical modeling. The limitations of our analysis are C, and we recommend that D be used to look at this future work". Otherwise this somewhat sprawling mess of a manuscript won't be comprehensible and won't contribute to broader community efforts.

We want to make clearer that this study focuses on better understanding the biases in oxygen levels in the tropical Pacific Ocean and reformulate the abstract as well as the title of the manuscript
"Title : The riddle of eastern tropical Pacific ocean oxygen levels : the role of the supply by intermediate depth waters"

"Abstract : Observed Oxygen Minimum Zones (OMZs) in the tropical Pacific ocean are located above intermediate depth waters (IDW). Typical climate models do not represent IDW properties and are characterized by a too deep reaching OMZ. We test here the role of the IDW on the misrepresentation of oxygen levels in a heterogeneous subset of ocean models characterized by a horizontal resolution ranging from 0.1° to 2.8°. First, we show that forcing the extra tropical boundaries (30°S/N) to observed oxygen values results in a significant increase of oxygen levels in the intermediate eastern tropical region. Second, the equatorial intermediate current system (EICS) is a key feature connecting the western and eastern part of the basin. Typical climate models lack in representing crucial aspects of this supply at intermediate depth, as the EICS is basically absent in models characterized by a resolution lower than 0.25°. These two aspects add up to a "cascade of biases", that hampers the correct representation of oxygen levels at intermediate depth in the eastern tropical Pacific Ocean and potentially future OMZs projections."

Otherwise this somewhat sprawling mess of a manuscript won't be comprehensible and won't contribute to broader community efforts.
Most of models display 2 biases : 1- not enough oxygen in the extra tropical region (an incorrect quantity is transported toward the eastern Pacific Ocean) 2- bad representation (absence) of the currents at intermediate depth (incorrect transport). These biases "cascade" as an incorrect quantity is incorrectly transported, leading to a large underestimation of oxygen levels in the tropical Pacific Ocean. This result is new and useful to the modeling community, especially the large community focusing on oxygen minimum zones (OMZ).

Our ms gives specifically 2 directions for future work : 1- improving the representation of the water masses subducting in the southern ocean is fundamental to represent correctly tropical OMZs, 2- the intermediate current system is basically "missing" in most of ocean models and there is a need to quantify more precisely its impact on biogeochemical cycles (especially when performing future projections). We have outlined these implications in the discussion.

The most important problem with the study is that it approached intermediate water masses (IWMs) with very little consideration of the associated water masses. Defining IWM as waters spanning 500m-1500m is scientifically flawed, and if the authors insist on having their analysis focused on that horizon then the words "intermediate water masses" should not appear in the title. There are a number of places where this problem arises.

We agree with the reviewer if the goal of this study were to assess the formation processes of the (generally speaking) "intermediate water masses" in high latitudes. However, we focus here on the region 30°S-30°N, far away from the regions where intermediate water masses subduct. The isopycnals are mostly flat meridionally in the latitude band 30°N-30°S. We show here that at 30°S models show too little oxygen at intermediate depth (500 – 1500 m). Identifying precisely the water masses based on density characteristics is not the scope of this study. We want to make clearer that the focus of this manuscript is to understand the tropical oxygen bias in ocean models in this intermediate depth range, rather than to quantify precisely the water masses composition, which is why we have replaced intermediate water masses (IWM) throughout the text by intermediate depth waters (IDW).

Our decision to focus on a depth range is not least due to that there is no unique way to define intermediate water masses based on density in a model environment. In a model intercomparison (CMIP3), Downes et al. (2010) used the density of the Potential Vorticity minimum to characterize the core of the SAMW and the Salinity minimum to characterize the core of the AAIW. As a result the SAMW and AAIW present density ranges varying in between models. Sallee et al. (2013) used a similar approach when comparing CMIP5 models. Lower and upper boundary of the water masses are determined by using an arbitrary density range. The arbitrary density range is either fixed (+/-0.03 kg.m-3 as in Downes et al., 2010) or adjusted (Sallee et al., 2013 state "manually adjusted to best capture the five water masses in each model analyzed"). Kwon (2013) used a different approach and solely define SAMW based on potential vorticity and state "the definitions of Subtropical Mode Water and Antarctic Intermediate Water (AAIW) are not strict in that they are defined here as two bordering water classes that are lighter and heavier than SAMW ". All these studies focus on the Southern Ocean (Southern of 30°S) and water mass formation processes.

Using this kind of methodology (note that there are several definitions of the intermediate water) in our study to understand oxygen distribution in the region 30°S-30°N may complicate the picture as the density structure of the models differs from observations and between each other, i.e the density of AAIW salinity minimum will be different.

Furthermore the oxygen content of a water parcel is very sensitive to its depth due to vertical mixing from the surface ocean (Duteil and Oschlies, 2009) we prefer to use a depth range rather than a density threshold that will vary in depth between the models.

We agree with the reviewer, the depth horizon 500 – 1500 m encompass several different "intermediate" water masses : see the table by Emery, 2003. Emery (1986, 2003) pragmatically

separated the ocean into 3 depth horizons: upper waters (0 - 500 m), intermediate waters (500 – 1500 m), deep waters (> 1500 m). In our study we use this basic classification. We do not think that it is fundamentally "scientifically flawed" as the goal of this study is 1- to highlight the sensitivity of tropical oxygen levels to subtropical oxygen concentration (30°S boundary). 2- to assess the role of the equatorial deep jets on oxygen levels. We do not focus on water mass formation and fate.

| Layer | Atlantic Ocean | Indian Ocean | Pacific Ocean |
|---|---|---|---|
| Upper waters (0–500 m) | Atlantic Subarctic Upper Water (ASUW) (0.0–4.0°C, 34.0–35.0‰) Western North Atlantic Central Water (WNACW) (7.0–20.0°C, 35.0–36.7‰) Eastern North Atlantic Central Water (ENACW) (8.0–18.0°C, 35.2–36.7‰) South Atlantic Central Water (SACW) (5.0–18.0°C, 34.3–35.8‰) | Bengal Bay Water (BBW) (25.0–29°C, 28.0–35.0‰) Arabian Sea Water (ASW) (24.0–30.0°C, 35.5–36.8‰) Indian Equatorial Water (IEW) (8.0–23.0°C, 34.6–35.0‰) Indonesian Upper Water (IUW) (8.0–23.0°C, 34.4–35.0‰) South Indian Central Water (SICW) (8.0–25.0°C, 34.6–35.8‰) | Pacific Subarctic Upper Water (PSUW) (3.0–15.0°C, 32.6–33.6‰) Western North Pacific Central Water (WNPCW) (10.0–22.0°C, 34.2–35.2‰) Eastern North Pacific Central Water (ENPCW) (12.0–20.0°C, 34.2–35.0‰) Eastern North Pacific Transition Water (ENPTW) (11.0–20.0°C, 33.8–34.3‰) Pacific Equatorial Water (PEW) (7.0–23.0°C, 34.5–36.0‰) Western South Pacific Central Water (WSPCW) (6.0–22.0°C, 34.5–35.8‰) Eastern South Pacific Central Water (ESPCW) (8.0–24.0°C, 34.4–36.4‰) Eastern South Pacific Transition Water (ESPTW) (14.0–20.0°C, 34.6–35.2‰) |
| Intermediate waters (500–1500 m) | Western Atlantic Subarctic Intermediate Water (WASIW) (3.0–9.0°C, 34.0–35.1‰) Eastern Atlantic Subarctic Intermediate Water (EASIW) (3.0–9.0°C, 34.4–35.3‰) Antarctic Intermediate Water (AAIW) (2–6°C, 33.8–34.8‰) Mediterranean Water (MW) (2.6–11.0°C, 35.0–36.2‰) Arctic Intermediate Water (AIW) (−1.5–3.0°C, 34.7–34.9‰) | Antarctic Intermediate Water (AAIW) (2–10°C, 33.8–34.8‰) Indonesian Intermediate Water (IIW) (3.5–5.5°C, 34.6–34.7‰) Red Sea–Persian Gulf Intermediate Water (RSPGIW) (5–14°C, 34.8–35.4‰) | Pacific Subarctic Intermediate Water (PSIW) (5.0–12.0°C, 33.8–34.3‰) California Intermediate Water (CIW) (10.0–12.0°C, 33.9–34.4‰) Eastern South Pacific Intermediate Water (ESPIW) (10.0–12.0°C, 34.0–34.4‰) Antarctic Intermediate Water (AAIW) (2–10°C, 33.8–34.5‰) |
| Deep and abyssal waters (1500 m-bottom) | North Atlantic Deep Water (NADW) (1.5–4.0°C, 34.8–35.0‰) Antarctic Bottom Water (AABW) (−0.9–1.7°C, 34.64–34.72‰) Arctic Bottom Water (ABW) (−1.8 to −10.5°C, 34.88–34.94‰) | Circumpolar Deep Water (CDW) (1.0–2.0°C, 34.62–34.73‰) | Circumpolar Deep Water (CDW) (0.1–2.0°C, 34.62–34.73‰) |
| | | *Circumpolar Surface Waters* | Subantarctic Surface Water (SASW) (3.2–15.0°C, 34.0–35.5‰) Antarctic Surface Water (AASW) (−1.0–1.0°C, 34.0–34.6‰) |

Table 1 from Emery (2003).

Downes, S. M., N. L. Bindoff, and S. R. Rintoul (2010), Changes in the subduction of Southern Ocean water masses at the end of the 21st century in eight IPCC models, *J. Climate*, 23, 6526–6541, doi:10.1175/2010JCLI3620.1.

Emery, W.J and J. Meincke. 1986. Global water masses: summary and review. Oceanol. Acta, 9, 383-391.

Emery, W. J. 2003. Water types and water masses. In: Encyclopedia of Atmospheric Sciences. 2nd ed. (eds. J.R. Holton, J.A. Curry and J.A. Pyle). Elsevier, Atlanta, GA, pp. 1556–1567

Kwon, E.Y (2013), Temporal variability of transformation, formation, and subduction rates of upper Southern Ocean waters, *J. Geophys. Res. Oceans*, 118, 6285– 6302, doi:10.1002/2013JC008823.

Sallée, J.□B., Shuckburgh, E., Bruneau, N., Meijers, A. J. S., Bracegirdle, T. J., Wang, Z., and Roy, T. (2013), Assessment of Southern Ocean water mass circulation and characteristics in CMIP5 models: Historical bias and forcing response, *J. Geophys. Res. Oceans*, 118, 1830– 1844 doi:10.1002/jgrc.20135.

One confusing case was for the point made in Fig. 2b, namely the panel showing the observed and modeled vertical distribution of O2 at 30S in the Pacific.

The reviewer does not state explicitly the issue but we may understand her/his concern. Indeed, the reviewer may think that we average zonally waters with very different characteristics (density, T,S, O2) at 30°S. Consequently, her/his confidence in the meaning of a zonal mean is low. The figure below shows the zonal oxygen level at 30°S in the Pacific Ocean for all the models.

[Figure]

Figure : left column : oxygen levels in observations and models at 30°S. The WOA oxygen levels are displayed in contour. Right column : salinity in observations and models at 30°S. The density anomaly (26.5, 27, 27.5) is displayed in contour (now Figure A5)

Based on this figure, discussing the Fig 2b and a zonally averaged quantity does not present any major difficulty. Generally speaking, zonally averaging water properties in depth coordinates in intermodel comparisons efforts is very common (e.g Cabre et al., 2015 for oxygen).

First and foremost, the authors really need to be clear about water masses.

We choose here a very pragmatic depth horizon and state that explicitly.

I believe that the O2 subsurface maximum is located near the boundary of SAMW and AAIW, rather than squarely in AAIW densities. This should be checked by averaging across the basin first in density at 30S.

We do not agree with the reviewer, the O2 subsurface maximum is clearly located below sigma 26.8 (often used as a density criterion to define AAIW in observations, e.g Karstensen et al., 2008) in the WOA close to the salinity minimum, characteristic of the core of the AAIW. This is consistent with Russell and Dickson, 2003.

[Figure]

Fig : oxygen levels (mmol.m-3) in observations at 30°S in the Pacific Ocean. The density anomaly levels 26.8 and 27.4 (kg.m-3) are displayed in black contour. Salinity levels lower than 34.5 are displayed in blue contour.

Karstensen, J., Stramma, L., and Visbeck, M.(2008). Oxygen minimum zones in the eastern tropical Atlantic and Pacific Oceans, Prog.Oceanogr., 77, 331–350, doi:10.1016/j.pocean.2007.05.009

Russell, J. L., & Dickson, A. G. (2003). Variability in oxygen and nutrients in South Pacific Antarctic Intermediate Water. Global Biogeochemical Cycles, 17(2), doi:10.1029/2000gb001317

Second, with the text in lines 209-210, it wasn't clear what the authors were saying about the "large role" of IWM. Are they referring to the "formation process" that sets the O2 content of IWM waters? Or something else?

The sentence L209 – 210 (now L238-239 in the ms version including corrections) is "The basin zonal average of the mean oxygen level in the lower thermocline (layer 500 - 1500) m at 30°S and in the eastern part of the basin (average 20°S – 20°N, 160°W-coast; 500-1500 m) are positively correlated (Pearson correlation coefficient R=0.73) (Fig 2d, Appendix A), suggesting a large role of the IWM in controlling the oxygen levels in the tropical oceans".

It has been modified and now reads : "The basin zonal average of the mean oxygen level in the lower thermocline layer (500 - 1500m) at 30°S and in the eastern part of the basin (average 20°S – 20°N, 160°W-coast; 500-1500 m) are positively correlated (Pearson correlation coefficient R=0.73) (Fig 2d, Appendix A), suggesting that the oxygen levels in the tropical pacific ocean are partly controlled by extra-tropical oxygen concentrations at intermediate depths and the associated water masses".

There are a number of questions that also arise from the application of Lagrangian diagnostics. First, near line 170 in the text, it wasn't clear whether the Lagrangian analysis included the bolus-velocities from the mesoscale parameterization? Or not?

The paragraph L169 (now L207) reads : "In order to complement the tracer experiment we performed Lagrangian particle releases. Lagrangian particles allow to trace the pathways of water parcels due to the resolved currents, and to track the origin and fate of water parcels. They are not affected by subgrid scale mixing processes". We replaced the last sentence by "They are not affected by subgrid scale diffusive and advective processes".

Second, it wasn't clear from the text if the NEMO01 flow-fields were coarse-grained to the NEMO05 grid before running the trajectory analysis?

This point is stated explicitly :

L173-174 (now  208 :"The NEMO01 circulation fields have been interpolated on the NEMO05 grid in order to allow a comparison of the large scale advective patterns between NEMO01 and NEMO05"

But most importantly, over lines 395-401, it was puzzling that the authors invoke "qualitative mode" rather than "quantitative" mode in designing their Lagrangian experiments if their goal is to evaluate connectivity through source regions.

The experiment is quantitative ("qualitative mode" is not mentioned in the manuscript) as we used a large number of particles to quantify the origin of water. The experiment shows clearly that particles originate from a broader region in NEMO01 compared to NEMO05 (Fig 7, 8). The connectivity between the western / eastern Pacific ocean, but also between the surface / deep ocean is increased in NEMO01.

The choice of 1000m as a release horizon isn't justified. Why was this chosen?

1000 m has been chosen as it is a depth where the equatorial intermediate current system is relatively well developed in high resolution models and basically absent in coarse models (see Fig 5). Another depth horizon in the range 500 – 1500 m (intermediate layer depth) would not change significantly our results. We have now added in the manuscript : "A depth horizon of 1000 m has been chosen as it is a depth where the equatorial intermediate current system is relatively well developed in high resolution models and basically absent in coarse models (see Fig 5). Our results are not sensitive to the choice of another depth horizon in the range of 500 - 1500 m"

How does Fig. 7 help the reader to understand the critical scale and processes needed to represent IWMs in the Equatorial Pacific?

The Figure 7 is a quantitative evaluation of the role of the equatorial intermediate current system on the transport of particles originating from the eastern Pacific Ocean (a) or the western Pacific Ocean (b) after a time scale of 15 years. It shows clearly that the Eastern Tropical Pacific ocean, where the OMZ are located is better ventilated than in NEMO01. 15 years has been chosen as the decadal time scale corresponds to the time scale of response of the OMZ to climate forcings (Deutsch et al., 2010, Duteil et al., 2018)

To reiterate, what is missing here is a clear exposition of scientific objectives, or a clear motivation for the specific choice of models and dye/lagrangian tracer diagnostics.

We agree that our scientific objectives were not defined and motivated clearly enough. In particular we want to communicate more clearly that the objective of this study is to better understand the supply of oxygen in the lower thermocline, at intermediate depth, toward the tropical eastern Pacific Ocean, where the largest OMZ are located.  We add in the introduction (L47) the following paragraph which makes the context / goal of this study clearer.

"Climate models tend to overestimate the volume of the OMZs (Cabre et al., 2015) and do not

agree on the intensity and even sign of oxygen future evolution (Oschlies et al., 2017). In order to

perform robust projections there is a need to better understand the processes at play that are

responsible for the supply of oxygen to the OMZ. We focus here on the Pacific ocean, where large OMZs are located in a depth range from 100 to 900 m (Karstensen et al., 2008; Paulmier and Ruiz-Pino. 2009). Previous modelling studies have shown that the tropical OMZ extension is at least partly controlled by connections with the subtropical ocean (Duteil et al., 2014). In addition, the role of the equatorial undercurrent (Shigemitsu et al., 2017; Duteil et al., 2018; Busecke et al., 2019), of the secondary Southern Subsurface Countercurrent (Montes et al, 2014), of the interior eddy activity (Frenger et al., 2018), have been previously highlighted. These studies focus on the mechanisms at play in the upper oxygen levels (upper 500 m meter). The oxygen content below the core of the OMZ however plays a significant role in setting the upper oxygen levels by diffusive (Duteil and Oschlies, 2009) or vertical advective (Duteil, 2019) processes. Here, we focus specifically on the mechanisms supplying oxygen toward the eastern tropical pacific ocean at intermediate depth (500 – 1500 m), below the OMZ core.

The water masses occupying this intermediate depth layer (500 – 1500 m) (Emery, 2003) subduct at high latitude" (…)

A fundamental limitation of the study is the lack of a water mass set of definitions for analysis with IWMs, without this the manuscript is in my opinion would clearly need to state "we ignore water masses except in a very hand-wavy qualitative sense, and focus instead on aggregated (AAIW, SAMW, etc.) properties over 500-1500m. "

We stated L47-49 (now L80-83) "AAIW, NPIW and the upper part of the PDW are oxygenated water masses occupying the lower thermocline between 500 and 1500 m depth. We will refer to the waters in this depth range as Intermediate Water Masses (IWM) in the following"

Following the suggestions of the reviewers, we now state: "In this study we do not specifically focus on the individual water masses, but rather on the water occupying the intermediate water depth (500 – 1500 m) of the subtropical and tropical ocean. We will refer to the waters in this depth range as intermediate depth water (IDW)".

In my opinion the manuscript would require major revisions to remedy these problems.

---

## Author Response (AR3)

We reproduce the reviewers comments in blue. Our replies are in black.

**Reviewer #1**
I thank the authors for addressing the comments.

I have been in the "oxygen community" for a while so I received the core message and potential importance of this study but I also see the other reviewer's points that it could be confusing to understand the aim in the previous manuscript. I think revising the introduction and conclusion improved this part and addressed the points.

Along with the reply to other reviewers comments, the authors stated the aim and objective more clear and the diagnostics and metrics based on this study will be useful for further evaluating a suite of models (such as CMIP6, and statements included in the conclusion).

We thank the reviewer for her / his positive evaluation.

I have few minor comments on the revised manuscript.

Section 2.1: Sub-section title 2.1 Mean state : I think you should state a bit more in details such as "evaluations of simulated mean states".
This part is now called "2.1 Description of models" (see also General comment from #Reviewer 4)

Section 2.2: Sensitivity simulations sub-section 2.2.1 Forcing of oxygen to observed values in the subtropical regions
- NEMO2-30S30N: I think it is informative to also state that the oxygen boundaries are forced to observed concentrations at the boundaries 30N and 30S for "the full depth (surface to the bottom)" (at least this is how I interpreted).
We now state "NEMO2-30S30N: the oxygen boundaries are forced to observed oxygen concentrations (WOA) at the boundaries 30°N and 30°S in the whole water column"

- NEMO2-30S30N1500m: Since you replied and discussed why you choose the depth interface of 1500m in the Reviewer #3's comment, I think you should include a short statement in this part justifying the choice of the depth interface.
We now state "NEMO2-30S30N1500M: same as NEMOO2-30S30N; in addition oxygen is forced to observed concentrations below 1500m, mimicking a correct oxygen state of the deeper water masses (lower part of the AAIW, upper part of the PDW)"

L248: "restore oxygen": you should replace to "force oxygen" (as you explained in the response and to make the terminology consistent in the manuscript).
Corrected

L350: a "primitive" EICS: could you clarify what you meant by "primitive" EICS?
We replaced "primitive" by "incomplete" : "NEMO2 and NEMO05 display an incomplete EICS as the LLSCs are not represented."

Section 4.3: title "Model resolution and oxygen levels":
I thank for the authors for revising this section and replying why you choose not to include NEMO for the comparison here.
However, I still feel that there is a slight jump from the previous sensitivity sections to this section comparing a suite of models. Perhaps you should consider changing the sub-section title including words "discussion" or "implications" for example (something more informative to smoothly make transition from the previous sections). You might also consider including the NEMO MKE figures and short statements in the Annex A.

The section 4.3 is not strictly necessary to understand the processes at play (described in 4.2). It is however useful, as it provides a validation of the processes at play in another set of models. We therefore transfert section 4.3 in Supplementary materia - Annex CI (see also General Comment 5 from Reviewer #4)

L472: take in account: this should be "take into account".
Corrected

**Reviewer #4**

Duteil et al. examined the role of intermediate depth water (IDW, 500-1500m) in affecting oxygen level in the eastern Pacific Ocean using a set of ocean models. They found the correct oxygen level at extra tropical boundaries (30oS and 30oN) and correct EICS simulation are important factors in simulating oxygen in the eastern Pacific Ocean. These findings could help to explain model bias in the climate models. The work is interesting. However, the numerical experiments are not well designed to convincingly obtain the conclusion and the oxygen analysis based on a certain depth without considering specific water mass are confusing. I suggest a major revision before considering the publication of this manuscript.

We sincerely thank the reviewer for taking the time to make a critical, fair, and in-depth review of our manuscript. Thanks to these comments, we believe that the manuscript has been significantly improved, in particular regarding the confidence in the experiments based on NEMO (major comment 2) and the quantification of the transport in the coarse / high resolution configuration (major comment 3). The wording (general comment 1) and structure (general comment 4 and 5) has also been improved.

General comments
1. IDW is not a specific water mass, but the text makes readers feel like this is a water mass. Lines 83-84 reads "They (models) generally display too shallow and thin IDW". If the IDW refers to a water in a certain depth, it could not be shallow and thin. This would make the readers confused. There are some similar descriptions in the text and these confused descriptions should be corrected.

The following sentences have been corrected :
L83-84 "It is known that current climate models, in particular CMIP5 (Coupled Model Intercomparison Project phase 5) models, have deficiencies in correctly representing the IDW and in particular, the AAIW. They generally display too shallow and thin IDW with a limited equatorward extension compared to observations."
has been replaced by :
"It is known that current climate models, in particular CMIP5 (Coupled Model Intercomparison Project phase 5) models, have deficiencies in correctly representing the IDW. In particular, the AAIW is too shallow and thin, with a limited equatorward extension compared to observations"

L203-205 "...The IDW subducted in mid/high latitudes are highly oxygenated waters. As part of the deficient representation of IDW, the subducted "oxygen tongue" (oxygen values up to 240 mmol.m$^{-3}$) is not reproduced in most of the models part of CMIP5"
has been replaced by :
"The water masses subducted in mid/high latitudes are highly oxygenated waters. The subducted "oxygen tongue" (oxygen values up to 240 mmol.m$^{-3}$) located at IDW level is not reproduced in most of the models part of CMIP5"

L488 "Intermediate Depth Waters (IDW) are subducted in the Southern Ocean and transported equatorward to the tropics by isopycnal processes"
has been replaced by :
"IDW are constituted by waters masses which are subducted in the Southern Ocean and transported equatorward to the tropics by isopycnal processes"

L498 "1. Subducted IDW properties and tropical oxygen"
has been replaced by : "1. Subtropical IDW .."

2. The numerical experiment NEMO2-30S30N and NEMO2-30S30N1500M are not well designed that use a correct oxygen level at 30N and 30S without considering density level. If the water mass is not at the correct depth in the model, the interpolation of oxygen at 30N and 30S may bring bias to the experiment. It would be more convincing if the authors at least could discuss this in the manuscript, such as a simple analysis of oxygen in the water mass to prove the bias is small.

The figure below shows that the density profile is well represented in the NEMO2 experiment

[Figure]

Fig : oxygen levels (colors) and density anomalies (contour) at 30°S, 30N and 100°W in the WOA dataset (a,b,c) and NEMO2-REF experiment (d,e,f)

The deficiency in oxygen in NEMO2-REF is clearly highlighted at 30°S, between 400 and 1500m. In comparison, the density field is well represented in NEMO2-REF. At 500m, density is about 26.6 in both WOA and NEMO2-REF. At 1500 m, the density is 27.6 in WOA and only 27.4 in NEMO2-REF, highlighting some potential water mass formation issue in NEMO2, as in most of models. A section at 100°W shows that isopycnals are almost horizontal at intermediate depth (500 – 1500 m) in WOA and NEMO2 in the subtropical and tropical ocean.

This Figure and the above text are now part of the Annex B

A further suggestion that the authors may use a regional model to do such an experiment instead

of a global ocean model. For example, one is forced by global ocean model results (salt, temp, o2), the other is forced by WOA results (salt temp o2).

The experiments that we perform is actually similar to what the reviewer suggest, as we compare 2 experiments using the same model but different oxygen boundaries conditions. We however do not force T,S at 30°S/30°N : by doing so, the circulation and ocean dynamics in the subtropics / tropics will be very different in both set of experiments and assessing the specific impact of a change in oxygen concentration at 30° very difficult. A solution may be to force velocities (e.g apply velocities of the experiment "Model boundaries" into the experiment "WOA boundaries") as well, but in this case T,S and velocities will not be consistent.

However, we agree that the approach suggested by the reviewer is generally interesting and modifying the open ocean boundaries of a regional model allows us to better understand the regional / large scale connections.

3. The conservative tracer release experiment (S2.2.2) cannot convincingly help readers to reach the conclusion that EICS is important in transporting oxygen from western Pacific to eastern Pacific. It would help to reduce confusion if the authors could give more explanation from result to conclusion. Except for tracer release, I would also suggest analyzing the volume transport or other convincing variables. If possible, remove the part that talking about surface circulation which makes the manuscript unfocused.

The part has been completely rewritten and reads now :

4.2.2  Equatorial IDW circulation

The analysis of the dispersion of Lagrangian particles (see 2.2.3) permits us to understand the origin of the waters circulating in the eastern part of the basin at IDW level. A total of 26515 particles have been released in the area located at 100°W, 10°N-10°S, 500-1500 m. These particles have been integrated backward in time in order to determine their origin and the ventilation of the eastern tropical Pacific ocean (Fig 7).

After 5 years of backward integration we find that the particles originate from a well defined region, which extends from 110°W and 80°W to NEMO05 (Fig 7a). This region extends westward till 150°W, as a result of the stronger currents in NEMO01 (Fig 7b). This larger dispersion and westward origin of the particles is clearly visible after 10, 20 and 50 years of integration. In order to quantify the dispersion of the particles, we define the Intermediate Eastern Pacific Ocean (IETP) as the region 10°N-10°S, 500 – 1500 m, 160°W – coast. The particles originating outside of the IETP in close to 5 % / 50 % of the cases in NEMO05 and 10 % / 60 % of the cases of NEMO01, after a time scale of respectively 10 and 50 years. The Fig 7c shows a lag between NEMO01 and NEMO05 : while 10 % of the particles originate outside the IETP after 10 years in NEMO01 the same quantity is reached only after 20 years in NEMO05, suggesting a stronger transport in NEMO01. However, after the time period of 20 years, the number of particles originating outside the IETP does not grow faster any more in NEMO01 compared to NEMO05. A hypothesis is enhanced recirculation in NEMO01: the same particles may recirculate several times in the equatorial region due to alternating zonal jets in NEMO01.

The transport has been quantified based on this Lagrangian particles release (Fig 8). The volume transport is higher in NEMO01 (up to 0.2 Sv) (Fig 8a) compared to NEMO05 (less than 0.1 Sv at

the equator) (Fig 8b). It also shows recirculating structures and alternating eastern and western transport in NEMO01 (Fig 8c). These recirculating structures are absent in NEMO05 and foster the dispersion of particles as shown above. The mean transport (zonal, meridional and vertical integration) in the region 10°N-10°S, 12E0°E-100°W is [value1] in NEMO01 and [value2] in NEMO05.

[Figure]

Figure 7 : Density (number of particles in a 1°x1° box) distribution of the location of released Lagrangian particles (backward integration in years) in a - NEMO05 and b- NEMO01. The release location is identified as a bold black line and is located at 100°W/10°N-10S/500-1500 m depth. The number of particles have been integrated vertically. The observed mean (500 – 1500 m) oxygen levels (WOA) are displayed in contour. The blue contour represents the Intermediate Eastern

Tropical Pacific basin (IETP). c – percentage of particles originating outside the Intermediate Eastern Tropical Pacific (IETP) basin (160°W, 10°N-10°S, 500-1500 m) in NEMO05 (red) and NEMO01 (black) over time (years)

[Figure]

Figure 8 : mean transport (Sv) in a- NEMO05 and -b NEMO01 derived from the release of particles at 100°W, 10°N-10°S, 500-1500m (backward integration). The mean zonal velocity (ms$^{-1}$) is represented in contour. c- zonally integrated transport (Sv) derived from the release of particles at 100°W, 10°N-10°S, 500-1500m in NEMO05 (red) and NEMO01 (black)

4. It would be clearer if the author could give more introduction to NEMO in the Section 2.1. The NEMO is explicitly used in this work and it need a detailed introduction. Also the two-way nest should be introduced here. More configurations should be written, such as mixing scheme, vertical coordinate, initial conditions.

The paragraph below has been added in section 2.1 and replaces the original text :

The NEMO (Nucleus for European Modelling of the Ocean) model (Madec et al., 2017) has been used throughout this study in different configurations. We first use a coarse resolution version (part 3 of this study). This configuration is known in the literature as ORCA2 (Madec et al., 2017) but we call it NEMO2 in this study for clarity reasons. The resolution is 2°, refined meridionally to 0.5° in the equatorial region. It possesses 31 vertical levels on the vertical (10 levels in the upper 100 m), ranging from 10 m to 500 m at depth. Advection is performed using a third-order scheme. Isopycnal diffusion is represented by a biharmonic scheme along isopycnal surfaces. The parameterisation of Gent and McWilliams (1990) (hereafter GM) has been used to mimick the effect of unresolved mesoscale eddies. The circulation model is coupled to a simple biogeochemical model that comprises 6 compartments (phosphate, phytoplankton, zooplankton, particulate and dissolved organic matter, oxygen). The same configuration has been used in Duteil et al., 2018; Duteil, 2019. The simulation has been forced by climatological forcings based on the Coordinated Reference Experiments (CORE) v2 reanalysis (Normal Year Forcing) (Large and Yeager, 2009) and integrated for 1000 years. Initial fields are provided by the World Ocean Atlas (temperature, salinity, phosphate, oxygen) (Garcia et al., 2019; Locarnini et al., 2019 )

Two other versions of NEMO have been used (part 4 of this study). The configuration ORCA05 (that we call here NEMO05) is characterized by a spatial resolution of 0.5°. It possesses 46 levels on the vertical, ranging from 6 to 250 m at depth (15 levels in the upper 100 m). Advection is performed using a third-order scheme. Isopycnal diffusion is represented by a biharmonic scheme along isopycnal surfaces. Effects of unresolved mesoscale eddies are parameterized following GM. In the configuration TROPAC01 (that we call NEMO01 in the rest of this study), a 0.1° resolution two-way AGRIF (Adaptive Grid Refinement In Fortran) nest (Debreu et al., 2008) nest has been embedded between 30°N and 30°S in the Pacific Ocean into the global NEMO05 grid. Since the model is eddying in the nested region GM is not used. Both configurations are forced by the same interannually varying atmospheric data given by the Coordinated Ocean–Ice Reference Experiments (CORE) v2 reanalysis products over the period 1948–2007 (Large and Yeager, 2009), starting from the same initial conditions. The initial fields for the physical variables are given by the final state of a 60 year integration of NEMO01 (using 1948–2007 interannual forcing and following an initial 80 year climatological spin-up at coarse resolution). The interpretation of differences in the ventilation in the IDW is aided by the use of a passive tracer.

5. Section 4.3 is an extended section of 4.2 with oxygen instead of tracer release. However, this section 4.3 is confused to me. I would suggest rewriting with a focus that directly connect with section 4.2.
The section 4.3 is indeed not strictly necessary to understand the processes at play (described in 4.2). It is however useful, as it provides a validation of the processes at play in another set of models. We therefore transfert section 4.3 in Supplementary material.

Specific comments

1. Line 9, "intermediate depth waters (IDW)", give a specific depth (500-1500m).
The sentence has been completed by "defined here as the 500 – 1500 m water depth"

2. Line 10, "test" --> analyze?
Analyze reads indeed better

3. Line 29, either it should say the euphotic zone or biological processes.
"biological respiration" has been replaced by "biological processes"

4. Lines 40-41, what is upper oxygen level?
We rephrased the sentence. It now reads " These studies focus on the mechanisms at play in the upper 500 m of the water column"

5. Line 49, which water mass is formed in North Pacific at depth 500-1500m? If there is, please describe it in the introduction part. If not, please delete the "North Pacific".
We deleted "North Pacific"

6. Line 106, title in section 2.1 should be model description, model introduction, or similar things.
The title reads now "Models description and experiments"

7. Lines 123-133, I suggest using the official name instead the name given by the authors. It is up to the authors if they prefer a given name in the text. But at least they should tell readers the official names of GFDL1, GFDL025, GFDL01.
The official name of GFDL1 is CM2-1deg, GFDL025 is CM2.5, GFDL01 is CM2.6. These names have been added into the text.

8. Line 145, the authors should discuss or show if the experiment reach an equilibrium status or not.
The role of spinup has been discussed in Annex A.

9. Line 160, "same as NEMOO2-30S30N" should be "same as NEMO2-30S30N"
We corrected the name "NEMO2-30S30N"

10. Line 160. The experiment description of NEMO2-30S30N1500 is not clear. Is it forced from 0-1500m, from bottom to 1500m, or only at 1500 m?
The sentence has been modified "in addition oxygen is forced to observed concentrations below 1500m"

11. Line 173, "In a second set" -- > "in the second set"?
Corrected

12. Lines 176-178. What is the 0.1o two-ways nest? The authors should introduce them in the model description part. In addition, it should be "two-way."
The model description has been updated. See major point above 4

13. Lines 189-190. "They (Lagrangian) are not affected by subgrid scale diffusive and advective processes." Why the Lagrangian is not affected by these two processes? This is confused. May be you want to say these processes are not considered in this work?
This sentence has been deleted. NEMO05 is not eddy resolving, therefore subgrid processes are parameterized by the Gent and McWilliams scheme. NEMO01 is eddy resolving and this subgrid parameterization is therefore not implemented. In order to allow a fair comparison between NEMO01 and NEMO05, we interpolated both models on the same NEMO05 grid, thus smoothing out a large part of the NEMO01 mesoscale activity. We consequently decided not to take in account NEMO05 subgrid processes. In any case, at the considered depth range (500 – 1500 m)

mesoscale activity is weak. We added a sentence "We do not take in account subgrid processes in NEMO05".

UVIC is a very coarse resolution model and is generally characterized by a too diffusive thermocline. However, the high oxygen levels may be due to isopycnal diffusion as well. Too oxygenated water at 30°S may originate from too oxygenated subducted water masses, as vertical mixing has been increased south of 60°S to enhance the overturning (Keller et al., 2015). However the aim of the present ms is not to discuss specifically the mechanisms at play in the UVIC model. We therefore remove this sentence.

We added 3 references. The sentence now reads : ".. consistent with the circulation of IDW with the gyre from the mid/high latitude formation regions towards the northwest in subtropical latitudes (Sloyand and Rintoul. 2001), and followed by a deflection of the waters in the tropics towards the eastern basin (Qu et al., 2004; Zenk et al., 2005). This oxygen peak is missing in all the models analyzed here."
Sloyan, B. M., & Rintoul, S. R. (2001). Circulation, Renewal, and Modification of Antarctic Mode and Intermediate Water. Journal of Physical Oceanography, 31(4), 1005–1030. doi:10.1175/1520-0485(2001)031<1005:cramoa>2.0.co;2
Qu, T., & Lindstrom, E. J. (2004). Northward Intrusion of Antarctic Intermediate Water in the Western Pacific. Journal of Physical Oceanography, 34(9), 2104–2118. doi:10.1175/1520-0485(2004)034<2104:nioaiw>2.0.co;2
Zenk, W., Siedler, G., Ishida, A., Holfort, J., Kashino, Y., Kuroda, Y., … Müller, T. J. (2005). Pathways and variability of the Antarctic Intermediate Water in the western equatorial Pacific Ocean. Progress in Oceanography, 67(1-2), 245–281. doi:10.1016/j.pocean.2005.05.003

We agree

A mean 1997 – 2007 or 2002 – 2007 does not display significant difference. We now use 2002 - 2007 in both cases.

Ok

Right bracket has been added. The whole paragraph has been rephrased (see point 20).

20. Line 294-295. Why is the imbalance apparent in the tropics and why is related to isopycnal diffusion? The authors need to give more explanation here. In addition, what I see about the change of isopycnal diffusion is largest in the sub-tropics instead of tropics.

The whole paragraph has been rewritten :

"Forcing oxygen levels in NEMO2-30S30N at 30°S and 30°N creates an imbalance between

respiration (which remains identical in NEMO2-REF and NEMO2-30S30N) and supply. The oxygen anomaly generated at 30°S propagates equatorward. The positive anomaly originated from the southern boundary recirculates in the equatorial region. Isopycnal diffusion is a major process that transport the oxygen anomaly toward the equator (Fig 3g, Fig 4h), in particular from 30°S to the 5°S and 30°N to 10°N. Total advective transport plays an important role in the transport of the oxygen anomaly as well, especially in the equator region (Fig 4e and 4f) and and in the western boundary (Fig 4f). Meridional advection plays a large role close to the 30° boundaries as the oxygen is transported by the deeper part of the gyres. As the vertical gradient of oxygen decreases (the intermediate ocean being more oxygenated), the vertical supply from the upper ocean decreases in the south (increases in the north) subtropical gyre (Fig 4g). Comparatively the impact on zonal term advection (Fig 4e) is small as the zonal oxygen gradient stays nearly identical in both experiments (the oxygen anomaly is almost longitude independent). The model does not display much increase in zonal recirculation at the equator as well, except in the western part of the basin due to the advection of the oxygen provided by the retroflection of the deep limb of the subtropical gyre. The increase of meridional transport (Fig 4f) is caused by the change in oxygen meridional gradient, mainly caused by isopycnal diffusion processes away from the western boundary"

21. Line 315, Why do you say the equatorward propagation is due to small scale isopycnal processes? I did not see it from the figures. Any citations?
The figure 3g (comparison between NEMO2_REF and NEMO2_30S30N) and 4b (bottom right : Δisopycnal diffusion) shows clearly that the increase in oxygen supply in the region (30°S-5°S) is related with an increase in isopycnal diffusion. Maybe our wording was confusing : "small scale isopycnal processes", "subgrid isopycnal processes" and "isopycnal diffusion" are synonyms (at least in our understanding) as isopycnal diffusion is ultimately based on a parameterization of non explicitly resolved processed.

22. Line 375, where is the release location? I tried to find it in the introduction part but cannot find it.
L179-181 state "In these experiments, we initialized the regions with climatological (WOA) oxygen levels greater than 150 mmol.m$^{-3}$ with a value of 1 (and 0 when oxygen was lower than 150 mmol.m$^{-3}$)".

23. Line 432. Is there any way that could quantify the contribution of ventilation due to EICS? Currently, the result and conclusion are very descriptive. I suggest digging more on it.
We added a quantification of the transport by the EICS (see major point 3)

24. Line 496. Not consistent. Line 496 says equatorward transport is due to isopycnal subgrid scale mixing processes. However, line 315 says the equatorward transport is due to western boundary current and isopycnal process.
We use "subgrid scale mixing" and "isopycnal process" as synonyms (see point 21). The paragraph line 496 has been rewritten for consistency.

The sentence now reads :

"Intermediate Depth Waters (IDW) are subducted in the Southern Ocean and transported equatorward to the tropics by isopycnal processes (Sloyan and Kamenkovich, 2007; Sallee et al., 2013; Meijers, 2014) and the retroflection of the deeper limb of the subtropical gyre (Zenk et al., 2005)

25. Figure 4 and other figures, a name should be given at each panel. Otherwise, it is really hard to know which panel you are talking about in the manuscript. Figure resolution is not high. The authors should provide a high-resolution version.
Each panel has been named in Fig 4 and when necessary to improve readability. .

26. Fig. 1a, this panel shows the circulation at the 500-1500m. Could the author confirm the SEC arrow here? In the surface, the SEC has a different path. If this is true in the 500-1500m, please cite a paper.

We based the trajectory of the SEC on the Fig 1a the Fig 1 of Kawabe et al. (2008). Their Fig 1 and the associated legend is reproduced below :

[Figure]

 "Fig 1 : Map of                                                                                      the South and tropical Pacific Ocean showing the study area enclosed by a square. Gray curves with arrows are the Antarctic Circumpolar Current, the subtropical gyre in the South Pacific, and the anticyclonic gyre current outside the subtropical gyre at a depth of approximately 800 m, referring to Reid (1997)."

Kawabe, M., , Y. Kashino, , and Y. Kuroda, 2008: Variability and linkages of New Guinea coastal undercurrent and lower equatorial intermediate current. *J. Phys. Oceanogr.*, 38, 1780–1793, doi:10.1175/2008JPO3916.1.

Reid, J. L., 1997: On the total geostrophic circulation of the Pacific Ocean: Flow patterns, tracers, and transports. *Prog. Oceanogr.*, 39 , 263–352.

The Fig 1a has been updated based on this figure.

[Figure]

a- schema summarizing the intermediate water masses (IWM) pathway from the subtropics into the equatorial regions. EICS : Equatorial Intermediate Current System. SEC : South Equatorial Current (Kawabe et al., 2008). Dashed line : isopycnal diffusive processes. Observed (World Ocean Atlas) oxygen levels (mmol.m$^{-3}$) in the lower thermocline (mean 500-1500m) are represented in color.

---

## Author Response (AR4)

Reviewer 4

I thank the authors for responding to my previous remarks in the revised manuscript. Largely, the changes that have been made are satisfactory. The revised version is clear and well structured. However, there are some typos in the manuscript, and I would like to suggest the authors really go through the manuscript several times including the text and figure. I still have some minor comment that might help to improve the manuscript. I suggest a minor revision before considering publication of this manuscript.

We thank the reviewer for her/his positive evaluation and reply below to the comments :

1. Where do the authors download the GFDL model results? Please add it in the data and code availability.

The model data has been directly provided by GFDL. The data that we use here will be made available on the GEOMAR data server after eventual acceptance for publication of the present study.

2. Line 251, the models presenting the poorest oxygenated water at 30S are GFDL025 and GFDL1. If I understand correctly, these models should be GFDL025 and UVIC model according to Fig. 2B?

The boundaries of the region used to compute the average oxygen profile in Figure 2B were indicated incorrectly in the figure caption (we apologize for that and thank the reviewer for picking that up). Figure 2B was not consistent with Fig 2C,2D,2E. The correct figure is now displayed. The text L251 is therefore correct.

[Figure]

Figure 2 : a- oxygen levels (mmol.m$^{-3}$) in observations (World Ocean Atlas - WOA) (mean 500 – 1500 m) and models (UVIC, NEMO2, GFDL1, GFDL025, GFDL01). Contours correspond to WOA values. b: average "30°S" (120°E-65°W, 30°S) c : average "tropics" (160°W-coast, 20°N-20°S). d: average "30°S" vs "tropics". e: average "30°S" vs volume of tropical suboxic ocean (oxygen lower than 20 mmol.m$^{-3}$) regions (1e15m3). b-e : UVIC : black, NEMO2 : cyan, GFDL1 : red, GFDL025, green; GFDL01 : blue, WOA: bold line (b,c) and star (d,e).

3. Line 285, NEMO2-30S30N1500M – NEMO2-30S30N should be NEMO2-30S30N1500M – NEMO2-ref. Please double check other typos.

We agree. Thank you for reading our manuscript very carefully – corrected.

4. Y tick label of figs. 3d and 3f is covered and missed. Please double check other figures.

We corrected the figure.

5. Is the oxygen restored at boundary of 30S and 30N or in the region beyond 30S-30N? Because I saw there are large changes between 30S-40S and 30N-40N in Figs. 3c-3f. If the oxygen is only restored at the section of 30S and 30N, it is worth to discuss the poleward influence (at least a simple discussion). This should be clarified in the model experiment introduction part.

The model is forced poleward of 30°, i.e WOA data have been used. The difference in oxygen poleward of 30° is the difference between model and WOA.

L183 now reads "the oxygen boundaries are forced to observed oxygen concentrations (WOA) poleward of 30°N and 30°S, that is in the mid and high latitudes".

"Poleward" has been added L275.

6. The simulation is from 1948-2007 in NEMO2-ref, NEMO2-30N30S, NEMO2-30S30N1500M. Which year results are you using for reanalysis in Fig. 3 and Fig. 4? Is it the whole simulation period? Please add this information to the introduction part or where you start to analyze the model results.

The following sentence has been added to L267 : "The mean state 1997 – 2007 of each experiment is used in the analyses below."

7. Line 300. Since the advection is separated into x and y directions, I think the equation should be in the cartesian coordinate. If I understand correct, Diff_iso should also be separated into x and y directions. Please double check this. I know the separation of diff_iso would not influence the figure 3g and Figure 4 and conclusion (you combine diff_iso_x and diff_iso_y together in the analysis). However, it should be clarified in this equation.

We prefer to stay with Diff_iso as we think that separating Diff_iso in its x and y components in the equation introduces some complexity without clear benefits, as the two terms are merged together in the analyses. In terms of formalism, considering "small scale physics" as a single term is widely used, e.g see below ( Gurvan Madec and NEMO System Team, *Scientific Notes of Climate Modelling Center (27)* – ISSN 1288-1619, Institut Pierre-Simon Laplace (IPSL)).

$$\frac{\partial T}{\partial t} = -\nabla \cdot (T\,\mathbf{U}) + D^T + F^T \tag{2.1d}$$

$$\frac{\partial S}{\partial t} = -\nabla \cdot (S\,\mathbf{U}) + D^S + F^S \tag{2.1e}$$

$$\rho = \rho(T, S, p) \tag{2.1f}$$

where $\nabla$ is the generalised derivative vector operator in $(\mathbf{i}, \mathbf{j}, \mathbf{k})$ directions, $t$ is the time, $z$ is the vertical coordinate, $\rho$ is the *in situ* density given by the equation of state (2.1f), $\rho_o$ is a reference density, $p$ the pressure, $f = 2\mathbf{\Omega} \cdot \mathbf{k}$ is the Coriolis acceleration (where $\mathbf{\Omega}$ is the Earth's angular velocity vector), and $g$ is the gravitational acceleration. $\mathbf{D^U}$, $D^T$ and $D^S$ are the parameterisations of small-scale physics for momentum, temperature and salinity, and $\mathbf{F^U}$, $F^T$ and $F^S$ surface forcing terms. Their nature and formulation are discussed in §2.5 and page §2.1.2.

8. Line 339, I cannot see deep oxygen anomaly is upwelled in the eastern equatorial part of basin from Fig 3g. I would suggest rephrasing the word here. The text should rely on the figure. In addition, a vertical advection like Fig. 4g should be provided for experiment of NEMO2-30S30N1500M (you could plot it in the supporting information). I am curious about the vertical advection difference between nemo2_30S30N_1500m and nemo2_ref (similar to fig 4g).

We corrected the sentence L339 to "In the experiment NEMO2-30S30N1500, in complement to the isopycnal propagation of the subtropical anomaly, the deep (> 1500 m) oxygen anomaly is upwelled in the eastern equatorial (500 – 1500 m) part of the basin (see Fig 3g and Fig S1 )."

[Figure]

NEMO2_30S30N1500M - NEMO2_REF

Figure S1: Difference in oxygen supply processes (mmol.m$^{-3}$.year$^{-1}$ – average 500-1500m) between NEMO2_30S30N1500M and NEMO2_REF : a- zonal advection, b- meridional advection, c- vertical advection, d- isopycnal diffusion. The NEMO2_30S30N1500M – NEMO2_REF oxygen anomaly (mmol.m$^{-3}$) is displayed in contour.

9. Lines 349-350. I would suggest the authors rephrase the conclusion here. In my understanding, the NEMO2-ref shows the equatorward transport of oxygen should be dominated by zonal and meridional advection (Fig. 4a-d). In the experiment of NEMO2-30N30S, the contribution of small-scale isopycnal processes increase very significantly in the region of 30s-5s. This indicates the role of small-scale isopycnal processes might be larger than what we expect if we have a correct oxygen level at 30S. If I understand correctly, please reconsider how to conclude a more scientific results here. Similar re-consideration is suggested in the lines 481-483.

L349 reads now : "Between 30°S and 5°S the oxygen transport occurs mostly by small scale

isopycnal processes while in the band 5°S - 5°N the transport is dominated by large scale advective processes. Increasing oxygen concentration in the gyres largely increases the relative importance of the isopycnal diffusion between 30°S and 5°S"

L487-489 (originally L481-483) :

"The equatorward transport of the anomaly in the subtropics from 30°S to 5°S is largely due to the isopycnal subgrid scale mixing processes away from the western boundaries, as shown by the NEMO2 budget analysis". Note that we discuss here the anomaly.

10. Line 387: 10 cm-1 should be 10 cm/s. Also in Line 386, 5 cm.s-1, should the dot be in the middle (i.e., ·) instead of period? Please follow the requirement of journal.

Corrected : 10 cm/s and 5 cm/s

11. Fig. 8, What's the meaning of negative and positive values? The word in line 457-458 of final version and line 533-535 of track version are different. Please double check.

Negative values correspond to a westward transport while positive values correspond to an eastward transport. This is now stated in the legend of Fig 8. The wording of the final version is the correct one (we apologize for the difference between the final and track version).

12. Line 410. I am still confused about the release location even you answer it previous. This experiment should set the tracer in the initial status, then run the simulation. Is there any release location that continue releasing tracer during the simulation ?

We added to L410 a reference to section 2.2.2, where the location of the tracer release is specified.

See L203 : "In these experiments, we initialized the regions with climatological (WOA) oxygen levels greater than 150 mmol.m$^{-3}$ with a value of 1 (and 0 when oxygen was lower than 150 mmol.m$^{-3}$)". We now state L204 : "The tracer is initialized at the beginning of the experiment and not continuously released."

13. Is there any specific reason that the authors plot the current at 1000 m in Fig. 5? Most of analysis in figs. 3 and 4 is based on the mean between 500-1500m.

We plot the current at 1000 m as to illustrate the complexity of the intermediate current system and minimize vertical averaging (the vertical alternation of the currents direction is clearly visible on Fig 5b)

14. Lines 361-363. A reference or figure should be provided to show the model well reproduce the upper current structure.

This sentence has been deleted as it is redundant with the following sentence : "Previous studies already discussed the upper thermocline current structure in the GFDL models suite (Busecke et al., 2019), NEMO2 and NEMO05 (e.g Izumo, 2005, Lübbecke et al., 2008), and UVIC (Loeptien and Dietze, 2013)"

The sentence was "The southern "shadow zone" is well individualized in NEMO01 compared to NEMO05 as the oxygen levels are high in the equator in NEMO01".

Indeed, the shadow zones are regions characterized by a poor ventilation. In NEMO05, the boundaries of this region are not clearly defined (Fig 6a,c) as it encompasses the equatorial region. Conversely in NEMO01, the region 10°S-20°S is significantly less ventilated compared to its surroundings.

We rephrased this sentence : "The poorly ventilated southern "shadow zone" (Luyten et al., 1983) is well characterized in NEMO01 compared to NEMO05, as its northern boundary is clearly defined by higher oxygen concentrations due to strong equatorial ventilation in NEMO01" and added a reference for the shadow zone :

Luyten, J. R., , J. Pedlosky, , and H. Stommel, 1983: The ventilated thermocline. *J. Phys. Oceanogr.*, **13 ,** 292–309.

16. Line 247, pacific should be Pacific. Double check other similar typos.

The P of Pacific is now in upper case. We corrected the same typo in line 45.

17. In Fig. 4a and 4b, a vector length-scale bar for current vector should be provided? Otherwise, it is hard to know the current size.

The Figure 4 has been updated with a vector length-scale bar.

18. Title in section 5 is suggested to be "discussion and conclusions" or "Summary and implications". It is okay if the author insists on the current one.

We concur with the reviewer's suggestion, the title is now "Summary and implications"

Corrected

20. Line 238 in track version. The region change from 5S-5N to 10S-10N. Is the simulation re-run or a typo in the previous version?

The Lagrangian experiments have been rerun. The release location is now 10°S-10°N, as this meridional band is characteristic of the equatorial jets  (see Fig 1, Fig 5, Fig 8)